# Astronomical Calibration of the Ypresian Time Scale: Implications for Seafloor Spreading Rates and the Chaotic Behaviour of the Solar System?

Thomas Westerhold[1], Ursula Röhl[1], Thomas Frederichs[2], Claudia Agnini[3], Isabella Raffi[4], James C. Zachos[5], Roy H. Wilkens[6]

[1]MARUM – Center for Marine Environmental Sciences, University of Bremen, Leobener Straße, 28359 Bremen, Germany
[2]Faculty 5 Geosciences, University of Bremen, 28359 Bremen, Germany
[3]Dipartimento di Geoscienze, Università degli Studi di Padova, via G. Gradenigo 6, 35131 Padova, Italy
[4]Dipartimento di Ingegneria e Geologia (InGeo) – CeRSGeo, Università degli Studi "G. d'Annunzio" Chieti-Pescara, via dei Vestini 31, 66013 Chieti-Pescara, Italy
[5]Department of Earth and Planetary Sciences, University of California Santa Cruz, 1156 High Street, Santa Cruz, CA 95064, USA
[6]Institute of Geophysics and Planetology, University of Hawaii, Honolulu, HI 96822, USA

*Correspondence to*: Thomas Westerhold (twesterhold@marum.de)

**Abstract.**

To fully understand the global climate dynamics of the warm early Eocene with its reoccurring hyperthermal events, an accurate high-fidelity age model is required. The Ypresian Stage (56 – 47.8 Ma) covers a key interval within the Eocene as it ranges from the warmest marine temperatures in the early Eocene to the long-term cooling trends in the middle Eocene. Despite the recent development of detailed marine isotope records spanning portions of the Ypresian Stage, key records to establish a complete astronomically calibrated age model for the Ypresian are still missing. Here we present new high-resolution X-ray fluorescence (XRF) core scanning Iron intensity, bulk stable isotope, calcareous nannofossil, and magnetostratigraphic data generated on core material from ODP Sites 1258 (Leg 207, Demerara Rise), 1262, 1263, 1265 and 1267 (Leg 208, Walvis Ridge) recovered in the Equatorial and South Atlantic Ocean. By combining new data with published records a 405-kyr eccentricity cyclostratigraphic framework was established, revealing a 300-400 kyr long condensed interval for Magnetochron C22n in the Leg 208 succession. Because the amplitudes are dominated by eccentricity, the XRF data help to identify the most suitable orbital solution for astronomical tuning of the Ypresian. Our new records fit best with the La2010b numerical solution for eccentricity, which was used as a target curve for compiling the Ypresian Astronomical Time Scale (YATS). The consistent positions of the very long eccentricity minima in the geological data and the La2010b solution suggest that the macroscopic feature displaying the chaotic diffusion of the planetary orbits, the transition from libration to circulation in the combination of angles in the precession motion of the orbits of Earth and Mars, occurred ~52 Ma ago. This adds to the geological evidence for the chaotic behaviour of the solar system. Additionally, the new astrochronology and revised magnetostratigraphy provide robust ages and durations for Chrons C21n

to C24n (47-54 Ma) revealing a major change in spreading rates in the interval from 51.0 - 52.5 Ma. This major change in spreading rates is synchronous with a global reorganization of the plate-mantle system and the chaotic diffusion of the planetary orbits. The newly provided YATS also includes new absolute ages for bio- and magnetostratigraphic events/reversals and early Eocene hyperthermal events. Our new biomagnetostratigraphically calibrated stable isotope compilation may act as a reference for further paleoclimate studies of the Ypresian which is of special interest because of the outgoing warming and increasingly cooling phase. Finally, our approach of integrating the complex comprehensive data sets unearths some challenges and uncertainties, but also validates the high potential of all chemostratigraphy, magnetostratigraphy, and biostratigraphy in unprecedented detail being most significant for an accurate chronostratigraphy.

## 1 Introduction

The Ypresian Stage, from 56.0-47.8 Ma, represents the first ~8 million years of the Eocene Epoch (Vandenberghe et al., 2012), an interval characterized by the warmest deep sea temperatures of the Cenozoic era (Zachos et al., 2008), multiple transient global warming events (Cramer et al., 2003; Lourens et al., 2005; Agnini et al., 2009; Galeotti et al., 2010; Leon-Rodriguez and Dickens 2010; Stap et al., 2010; Zachos et al., 2010; Sexton et al., 2011; Slotnick et al., 2012; Littler et al., 2014; Kirtland-Turner et al., 2014; Lauretano et al., 2015; Lauretano et al., 2016) and major faunal turnovers (Thomas and Shackleton, 1996; Gingerich et al., 2003; Clyde et al., 2007). Climatic records from the Ypresian greenhouse are of special interest because they potentially capture the behaviour of Earth's climate system under those $pCO_2$ concentrations likely to be reached in the near future (Zachos et al., 2008; Meinshausen et al., 2011). For the detailed reconstruction of the Eocene greenhouse climate system a complete, precise and highly accurate stratigraphic framework to determine rates of climatic processes and timing of events is required.

Despite the recent progress in construction of astronomically calibrated age models for the Eocene using geochemical records (Cramer et al., 2003; Lourens et al., 2005; Westerhold et al., 2007; Galeotti et al., 2010; Jovane et al., 2010; Westerhold et al., 2012; Westerhold and Röhl 2013; Littler et al., 2014; Westerhold et al., 2014; Westerhold et al., 2015; Lauretano et al., 2016) a fully consistent astrochronology for the Ypresian is not yet available. Two major issues have to be solved to achieve a complete Ypresian Astronomical Time Scale (YATS): 1) the "50 Ma discrepancy" in magnetostratigraphy (Vandenberghe et al., 2012; Westerhold et al., 2015) and 2) the exact number of 405-kyr eccentricity cycles in magnetochron C23 (Lauretano et al., 2016).

The "50 Ma discrepancy" arises from the short duration of magnetochron C23n.2n in Ocean Drilling Program (ODP) Site 1258 data (Suganuma and Ogg 2006; Westerhold and Röhl 2009) that results in a very unlikely abrupt global increase in spreading rate for this chron only at around 50 Ma (Vandenberghe

et al., 2012). Assuming a 400 kyr longer duration for Chron C22r in the same record would result in lower spreading rates than calculated on the basis of the standard Geomagnetic Polarity Time Scale (GPTS) using the synthetic magnetic anomaly profile of the South Atlantic (Cande and Kent 1992; Cande and Kent 1995; CK95). But the Site 1258 magnetostratigraphy is currently the only available record covering

the entire Ypresian. Paleomagnetic interpretation for sediments from Site 1258 is difficult because the core was retrieved by rotary drilling. Minicores from Site 1258 gave a relatively good magneto-stratigraphy but it is based on a rather subjective method of polarity interpretation (Suganuma and Ogg, 2006). After revision of the composite record of Site 1258 the interpretation was improved (Westerhold and Röhl 2009), but the interpretation for Chrons C22r, C23n and C23r remained ambiguous.

Eccentricity-modulated precession cycles in X-ray fluorescence (XRF) core scanning Iron (Fe) intensities from Site 1258 and Leg 208 sites helped to set up the first cyclostratigraphic age models for the Ypresian (Westerhold et al., 2007; Westerhold and Röhl 2009). High-resolution bulk and benthic stable isotope data for the early (Cramer et al., 2003; Zachos et al., 2010; Stap et al., 2010; Littler et al., 2014; Lauretano et al., 2015; 2016), middle (Kirtland-Turner et al., 2014) and late Ypresian (Sexton et al., 2011) showed

exceptionally strong 100 and 405-kyr eccentricity cycle variations that partly were utilized for age model construction. In order to test the 50 Ma discrepancy and the astronomical age model at ODP Site 1258, benthic high resolution stable isotope records spanning 54 to 49 Ma were compiled from ODP Site 1262 (Littler et al., 2014; Lauretano et al., 2015) and Site 1263 (Stap et al., 2010; Lauretano et al., 2015; Lauretano et al., 2016), and astronomically tuned to the La2010d (Laskar et al., 2011a) orbital solution

(Lauretano et al., 2016). In theory, the dominant eccentricity related cyclicity in this interval should enable development of a robust astrochronology. However, a period of low amplitude variability and a major shift in benthic $\delta^{13}C$ data in Chron C23n and upper C23r complicate the identification of the correct number of 405-kyr cycles. Lauretano et al (2016), assumed constant sedimentation rates and including bulk stable isotope data from Site 1258, opted for an age model with two 405-kyr cycles in this complex

interval which provided absolute age estimates for early Eocene hyperthermal events, but did not solve the "50 Ma discrepancy" because the duration for C23n.2n is much too short (295 kyr) compared to CK95 (696 kyr).

To establish a complete Ypresian Astronomical Time Scale (YATS) with consistent GPTS ages a new deep-sea magneto-cyclostratigraphic record is needed to test the ODP Site 1258 magnetostratigraphy. In

particular, the durations of Chrons C22 and C23 have to be evaluated. Here we present a new complete magnetostratigraphy spanning Chron C21n to C24n by integrating records from ODP Leg 208 Sites 1262, 1263, 1265 and 1267. New XRF core scanning data and core images are used for ultra-high resolution correlation between the Leg 208 sites. New calcareous nannofossil data from Site 1258, 1263 and 1265 are presented and combined with published datums for direct calibration to magnetostratigraphy and

revision of age datums. After integration of Leg 208 and ODP Leg 207 Site 1258 isotope data with new XRF data and core images a cyclostratigraphic framework has been compiled that subsequently was

astronomically tuned to the La2010 (Laskar et al., 2011a) orbital solution. This study provides the first complete and consistent bio-chemo-magnetostratigraphy for the Ypresian Stage. Our new data sets combining two regional records of ODP Leg 207 and 208 reveal unprecedented insight into possible challenges and uncertainties, but also demonstrate the high potential of detailed chemostratigraphy, magnetostratigraphy, and biostratigraphy that will be the prerequesite for the major field of chronostratigraphy.

## 2 Material and Methods

New data for this study were generated from sediment cores retrieved during ODP Leg 207 (Site 1258, Erbacher et al., 2004) and Leg 208 (Sites 1262, 1263, 1265, 1267, Zachos et al., 2004). The carbonate-rich sequences drilled at the equatorial Atlantic Demerara Rise (Leg 207) and the southeast Atlantic Walvis Ridge (Leg 208, Fig. 1) were recovered from multiple holes combined to composite records. For Sites 1262 and 1265 the shipboard composite depth was used (mcd – meters composite depth) for analysis (Zachos et al., 2004). For Sites 1258, 1263, and 1267 the revised composite record was applied (rmcd – revised composite depth; Westerhold et al., 2007; Westerhold and Röhl 2009; Westerhold et al., 2015) for further analysis.

### 2.1 XRF core scanner data

To obtain a complete record for the Ypresian, published data had to be combined with the new XRF Fe core scanning data. The newly acquired data presented here have been measured on the three XRF core scanners at MARUM – Center for Marine Environmental Sciences, University Bremen, with various hardware generations and under different settings (for details see supplement). Site 1262 XRF data were obtained from 92.74-104.57 and 111.10-112.40 mcd, and combined with data from Westerhold et al. (2007) and (2012). Site 1263 data reported here are from 168.08-289.00 rmcd and combined with data from Westerhold et al. (2007). Site 1265 XRF data were measured from 228.10-275.00 mcd and merged with data from Westerhold et al. (2007). Site 1267 XRF new data were generated from 153.12-236.53 rmcd. Settings for XRF core scanning of Site 1258 are given in Westerhold and Röhl (2009). Here we report the data from 134.61 to 212.49 rmcd for Site 1258. 18,000 new XRF core scanning data points (see supplement) combined with published data result in a total of more than 30,000 data points covering the latest Paleocene and Ypresian. This enormous dataset is the prerequisite to correlate the five drill sites in detail.

### 2.2 Bulk stable isotope data

To close a stratigraphic gap in the stable isotope record for Site 1263, a section (230.50 to 239.00 rmcd) of a nearby Site 1265 was selected. A total of 369 sediment samples were collected, freeze-dried and powdered (Table S8, supplement). The $\delta^{13}C$ and $\delta^{18}O$ values were measured at MARUM – Center for

Marine Environmental Sciences, University of Bremen, and are reported relative to the Vienna Pee Dee Belemnite (VPDB) international standard, determined via adjustment to calibrated in-house standards and NBS-19. Analyses at MARUM were carried out on a Finnigan MAT 251 mass spectrometer equipped with an automated carbonate preparation line (Kiel III). The carbonate was reacted with orthophosphoric acid at 75 °C. Analytical precision based on replicate analyses of an in-house standard (Solnhofen Limestone) averaged 0.03‰ (1sigma) for $\delta^{13}C$ and 0.06‰ (1sigma) for $\delta^{18}O$ for samples measured in 2014, and 0.04‰ for $\delta^{13}C$ and 0.04‰ for $\delta^{18}O$ for samples run in 2015.

Published stable isotope data also used in this study have been compiled from Lourens et al., 2005; Zachos et al., 2005; McCarren et al., 2008; Stap et al., 2009; Stap et al., 2010; Zachos et al., 2010; Sexton et al., 2011; Westerhold et al., 2012; Kirtland-Turner et al., 2014; Littler et al., 2014; Lauretano et al., 2015; Westerhold et al., 2015; Lauretano et al., 2016. All data are provided in the supplement (and PANGAEA) relative to the Site 1263 depth and with the tuned age added.

## 2.3 Paleomagnetic data Leg 208

Natural remanent magnetization (NRM) was measured on 400 discrete cube samples (dimensions 2 cm x 2cm x 2 cm) to document magnetic polarity boundaries C20r to C24r at ODP Sites 1262 (64), 1263 (128), 1265 (115) and 1267 (89). Discrete samples were analyzed at the Faculty 5 Geosciences, University of Bremen. Paleomagnetic directions and magnetization intensities were measured on a cryogenic magnetometer (2G Enterprises model 755 HR). NRM was measured on each sample before being subjected to a systematic alternating field demagnetization treatment involving steps of 7.5, 10, 15, 20, 25, 30, 40 and 60 mT. Intensities of orthogonal magnetic components of the remanent magnetization were measured after each step. Raw inclination, declination, and intensity data for each measurement step are provided in tables S17-S20, the calculated characteristic remanent magnetization in tables S21-S25 and the magnetostratigraphic interpretations including published data are recorded in table S26-S30 for each of the five sites.

## 2.4 Calcareous nannofossils at ODP Sites 1258, 1263 and 1265

Smear slides were processed following the standard procedures described in Bown (1998) in order to investigate calcareous nannofossil assemblages. High-resolution semi-quantitative counting methods, which consist of counting the number of forms ascribed to the same taxon detected on or normalized to a prefixed area (i.e., 1 mm$^2$; Backman and Shackleton, 1983; Agnini et al., 2014), were performed. These counts provide very detailed abundance patterns, which permit to precisely identify the position of each biohorizon, especially in those cases in which either the marker species displays an overall uneven distribution or a marker species is rare and discontinuous at the base and top of its stratigraphic range. On this basis and according to Agnini et al. (2014), six types of biohorizons are adopted, these are: Base rare (Br), Base (B), Base common and continuous (Bc), Top (T), Top common and continuous (Tc) and

crossover (X). In two cases we used the increase/decrease in abundance of taxa (i.e., *Fasciculithus* spp., *Zygrhablithus bijugatus*) to define additional biohorizons. Taxonomic concepts adopted in this study are those of Perch-Nielsen, (1985) and Agnini et al. (2007).

Calcareous nannofossil biostratigraphic analyses were newly performed or refined for the early Eocene
of ODP Sites 1258, 1263 and 1265. The biohorizons used in different low to middle latitude zonations (Martini, 1971; Okada and Bukry, 1980; Agnini et al., 2014), as well as additional biohorizons, were recognized. A set of 168 samples was analysed from the Demerara Rise (equatorial Atlantic) which allows for the identification of forty biohorizons. At Walvis Ridge (SE Atlantic), a total number of 181 samples was studied from ODP Sites 1263 (77) and 1265 (104), which permits to detect twenty-seven biohorizons.
New data were integrated with ship data or other published results to obtain a more complete and reliable dataset. Tables of the calcareous nannofossil biohorizons are given in table S31-35.

## 3 Results

All data generated within and available data compiled for this study are combined in the data set file and available open access online at http://doi.pangaea.de/10.1594/PANGAEA.871246.

**3.1 XRF core scanning results**

Fe intensity data reveal the cyclic pattern commonly observed for the interval from PETM to ETM-2 (Westerhold et al., 2007, Littler et al., 2014) with higher values in darker, more clay rich layers. A decrease in carbonate content around 51 Ma for all sites (Zachos et al., 2004) is reflected by overall higher Fe intensities. Multiple distinct peaks in Fe data for Chrons C22 and C23n correspond to strikingly
bundled sets of clay rich dark intervals for all sites. At Site 1262 the XRF Fe record ends with the shoaling of the CCD above the site in Chron C21r around 93 mcd (Zachos et al., 2004). Generally, the records from Sites 1262 and 1267 are of lower resolution than those from Sites 1263 and 1265 due to the regional decline in carbonate accumulation rates with increasing water depth. Site 1263 shows the most persistent high resolution XRF Fe intensity signal. A gap in the 1263 record from 229.15 to 233.68 rmcd caused by
drilling disturbance (Zachos et al., 2004) can successfully be bridged by the records from Sites 1265 and 1267. The high resolution XRF Fe data show consistent patterns that are required to do a detailed site to site correlation and integration of Leg 208 Sites 1262, 1263, 1265 and 1267. New XRF Fe intensity data for Site 1258 between PETM and ETM-2 reveal the same eccentricity modulated precession cycles as observed from Leg 208 sites.
XRF core scanning Fe intensity data from four Leg 208 sites are shown in supplementary figure S1 and given in tables S1 to S7. All data plotted versus Site 1263 depth are given in Fig. 2 from ETM-2 to Chron C20r. Data for the interval from the PETM to ETM-2 are plotted in the supplementary figure S7. Due to the large and very detailed data set and the fact that most of the data from PETM to ETM-2 have been

published previously, priority for figures in the main manuscript is on the interval from ETM-2 to Chron C20r.

## 3.2 Bulk stable isotope results

To obtain a complete bulk stable isotope record for Leg 208 sites the gap in the Site 1263 record was bridged by incorporating bulk data from Site 1265 (Fig. S1a). Bulk stable isotope data from Site 1265 show cyclic variations between 1.6 and 2.2 ‰ and match with overlapping data from 1263 (Westerhold et al., 2015, Fig. 2, 217 to 227 rmcd 1263). As observed previously (Zachos et al., 2010; Littler et al., 2014), lighter bulk $\delta^{13}$C data coincide with dark clay-rich intervals with relatively higher XRF Fe intensities in all Leg 208 sites. All bulk and benthic data compiled for this study are presented in table S8 to S16.

## 3.3 Core image processing and Site-to-Site correlation

To correlate and integrate Leg 208 and Site 1258 records the new software tool CODD (Code for Ocean Drilling Data, Wilkens et al., 2017.) was utilized. This tool greatly facilities handling of large and complex data sets and allows to use core images for scientific analysis. For all sites in the study, core images and all available data were assembled by holes. Then the composite records were cross-checked and assembled. In order to be able to use data generated outside of the splice all cores where mapped onto the splice by differential stretching and squeezing (Tab. S36-S40). Site 1263 was chosen as the reference site to correlate all other sites to because it has the highest sedimentation rates and the most detailed stable isotope data (Lauretano et al., 2015; 2016, McCarren et al., 2008; Stap et al., 2010; Westerhold et al., 2015). XRF Fe and core images primarily guided the correlation between sites (Fig. 2). Stable isotope data are used to assess the correlation between 1258 and 1263 because the XRF Fe data of 1258 are dominated by precession cycles and thus difficult to directly correlate to the eccentricity dominated cycles in 1263 (Westerhold and Röhl 2009). Figure 2 shows the outstanding match between the sites just by visual comparison of the core images. Existing correlations between Leg 208 sites (Röhl et al., 2007; Westerhold et al., 2007; Lauretano et al., 2015) were updated and adjusted if needed. Correlation tie points are provided in table S41-S44. The recent correlation between 1262 and 1263 as well as 1263 and 1258 by Lauretano et al., 2015 and Lauretano et al., 2016 based on stable isotope data were further refined as well. The primary modifications between 1262 and 1263 were made in a short interval (265 to 267 rmcd) of 1263. The detailed comparison with Site 1265 reveals a gap of about 3 precession cycles in XRF Fe at 284.40 rmcd at Site 1263 due to a core break (Fig. S6). No additional mismatches were recognized suggesting that the combined Leg 208 records represent the complete stratigraphic sequence for Walvis Ridge. However, correlation of 1258 to 1263 shows overall good agreement except for the interval from 230 to 235 rmcd 1263. Fine scale comparison of the benthic 1258 and bulk 1265 stable $\delta^{13}$C data show that Leg 208 sediments encompass a regional condensed interval at ~229 to 230 rmcd 1263 (Fig. S8). The

missing stratigraphic interval spans 300 to 400 kyrs as depicted from the 1258 benthic record, and thus needs to be corrected for in the 1263 astronomically tuned age model. This demonstrates the benefits of utilizing multiple records from different regions with robust composite records to establish a highly accurate stratigraphic framework based on orbitally tuning for any given interval.

## 3.4 Magnetostratigraphic results and interpretation

Vector analysis according to the method by Kirschvink (1980) without anchoring to the origin of the orthogonal projections was applied to the results of the AF demagnetization of NRM to determine the characteristic remanent magnetization (ChRM). The maximum angular deviation (MAD) values were computed reflecting the quality of individual magnetic component directions. Most of the MAD values are below 10 (Fig. S4, Table S17-S30). Figure S5 displays the demagnetization characteristics of a sample with reversed polarity from C22r and a sample with normal polarity from C22n, respectively. As an example of samples with demagnetization behaviour with larger scatter (larger MAD), data from a sample at the C22n/C22r reversal are plotted in Fig. S5. The larger MADs that a few samples show are not simply related to the intensity of their remanent magnetization. The median destructive field (MDF) of the NRM demagnetization is comparably low for most of the samples. It ranges from 2.6 to 24 mT (mean 6.1 ± 3.8 mT), indicating a magnetically soft overprint in many samples. The interpretation of the ChRM in terms of magnetic polarity is focused on the inclination data, which provide a reliable magnetostratigraphy for most intervals.

Recognition of calcareous nannofossil events allow for the magnetic chrons to be clearly identified from C20r to C24r (Fig. 2 and S4, Tables S26-S30). Raw inclination, declination, and intensity data for each measurement step for Leg 208 sites are given in Table S17-S20. Magnetostratigraphic interpretation is given in Table S21-S25. Processed paleomagnetic data from Leg 208 sites, the basis for the magnetostratigraphic interpretation, are provided in Table S26-30 for each site. The assignment of error bars, as with all magnetostratigraphic data, is a subjective endeavor. The error bar for Leg 208 data marks the interval where the inclination shifts from clearly reversed to clearly normal polarity or vice versa. Poorer sample resolution and/or ambiguous or transitional inclination values across a reversal thus will increase the error bar. We did not apply an inclination threshold value to mark a shift in polarity because the reversals occur in different seafloor depth at all sites. Drilling depth and compaction difference between sites might have affected the inclination at each site differently. A much sharper-defined length of error bars could be derived from higher-resolution data (e.g., by analyzing u-channels) which is beyond the scope of this study.

Having multiple magnetostratigraphic records from the same region combined with the high-resolution correlation established allows the quality of the paleomagnetic data to be evaluated and inconsistencies identified. This, again, is crucial for resolving the "50 Ma discrepancy" because a single magnetostratigraphic record from one succession could contain significant unresolved errors. Plotting all

Leg 208 ChRM data and the published Site 1258 magnetostratigraphic interpretation against Site 1263 depth immediately shows how consistent but also "dynamic" the magnetostratigraphy of each site can be (Fig. 2). For example, Chron C22n is clearly too short at Site 1265 which could be related to the condensed interval in this part. Sites 1262 and 1267, however, are consistent with the Site 1258 Chron C22n

thickness. At Site 1263 the top of Chron C22n is compromised by drilling disturbance and the base of Chron C22n is spread over a larger interval than at the other Leg 208 sites. Chron C21n is well captured in Sites 1258, 1263 and 1267, and the base also in 1265. The top of Chron C23n is consistent between Sites 1258, 1262, 1263 and 1267. The signal from Site 1265 is a bit noisy and a clear identification for the top of Chron C23n is difficult, probably the normal interval labelled as C23n could only be C23n.2n.

The bottom of Chron C23n is consistent within error in Sites 1262, 1265 and 1267, with 1262 giving the best signal. The ChRM of Site 1263 does not provide an interpretable signal below 260 rmcd preventing the identification of the base of C23n and the entire Chron C24n. Clearly, comparison to Site 1258 reveals that Chron C23n is too short in 1258, probably due to the position of the base Chron C23n. Chron C24n can be identified in Leg 208 but has relatively larger error bars than the other chrons. The top of Chron

C24n spreads out from ~270-273 rmcd 1263 considering all sites. Taking the overlaps of the error bars into account the best position for the top and bottom of Chron C24n is taken from Site 1262. More difficult to determine is the exact position of the reversals in Chrons C24n, C24n.1r and C24n.2r. No data are available for Sites 1263 and 1265 in this interval. Resolution at Site 1262 is too low to resolve the short reversed chrons. Sites 1267 and 1258 do not give consistent results either. For the moment the Site 1258

positions are used for the combined magnetostratigraphy but will need future revision. Based on the integration of all data and evaluation of errors a best fit combined magnetostratigraphy for the Ypresian was constructed and is given in table S45. The results of the combined ChRM data on the high-resolution correlation suggest that a magnetostratigraphic interpretation from a single site might contain significant errors that need to be taken into account. Thus a magnetostratigraphic interpretation from a single site or

location can lead to major discrepancies when used as a template for orbital tuning. The new multi-site data already resolve the "50 Ma discrepancy" showing that Chron C23n is too short at Site 1258 causing spreading rates that are too high for the South Atlantic. All of the above uncertainties have been considered while doing the time series analysis and subsequent astronomical calibration for the Ypresian.

**3.5 Calcareous nannofossil events in Sites 1258, 1263 and 1265**

High-resolution correlation between the sites allows us to investigate how reliably calcareous nannofossil datums can be determined, especially over the depth transect of Leg 208. Therefore, key taxa were targeted to be identified at sites 1263 and 1265 to be compared to the high-resolution work at 1262 (Agnini et al. 2007) and low resolution ship board data at Site 1267. Biostratigraphic datums are transformed in biochronological data using the integrated bio-magneto-astrocyclostratigraphic age model developed in

this study (Table S31-S35). Age estimations of calcareous nannofossil biohorizons are generally

consistent through the Walvis Ridge sites and in agreement with recently published bichronological data (see Agnini et al., 2014 for review). Most of these biohorizons, in particular almost all the bioevents used in previous and more recent calcareous nannofossil biozonations were proved to represent reliable data and powerful tools for highly-resolved correlations. A total number of eighteen biohorizons across the study interval (i.e., decrease in diversity of *Fasciculithus spp.*, B of *Rhomboaster spp.*, the crossover (X) between *Fasciculithus spp.* and *Zyghrablithus bijugatus*, the T of *Fasciculithus spp.*, the Base od *D. diastypus*, the T of *Tribrachiatus orthostylus*, the T of *Tribrachiatus contortus*, the Tc of *Discoaster multiradiatus*, the B of *Sphenolithus radians*, the B of *Girgisia gammation*, the T of *D. multiradiatus*, the Br and B of *Discoaster lodoensis*, the B *Chiphragmalithus spp.* (circular), the T of *T. orthostylus*, the T of *D. lodoensis*, B of *Chiphragmalithus calathus* and the B of *Nannotetrina spp.*) were calibrated and the age estimations are impressively consistent throughout Walvis Ridge sites. Two exceptions are the B of *Discoaster sublodoensis* for which the estimates calibrated at different sites show a high degree of uncertainty and the B of *Blackites inflatus*, which was not possible to identify because of the absence of this taxon at Walvis Ridge. Other biohorizons, as for instance the B of *Coccolithus crassus*, are not tested at Walvis Ridge but seems to be promising if data from Demerara Rise and Tethyan data are compared (Agnini et al., 2014).

Data from ODP Site 1258 were produced to biostratigraphycally frame the study succession but these data are also used to investigate the degree of reliability of calcareous nannofossil data over wide areas. In general, the stratigraphic positions as well as the ranking and spacing of the biohorizons detected at this site are in fair agreement with data from Walvis Ridge. The two bichronological datasets presented for Walvis Ridge and Demerara Rise sites showed that the ages calculated for some biohorizons, the B of *Rhomboaster spp.*, the T of *Fasciculithus spp.*, the B of *T. orthostylus*, the Tc and T of D. *multiradiatus*, the B of *D. lodoensis*, the T of *T. tribrachiatus* and the B of *Nannotetrina spp.*, are in fact quite close. However, some other results (i.e., the B of *D. diastypus* and B of *T. contortus*) are certainly anticipated with respect to the results obtained at Walvis Ridge. These discrepancies are of particular relevance in the mid-upper part of Chron C24r though a general slight offset is observable between the two areas. In accordance with the age models developed for the study sites, the anticipation of the first occurrences of some taxa could be explained as related to the warm water preferences of the taxa considered (e.g., *Discoaster* and *Tribrachiatus*), but more data are needed to confirm if the diachroneity recorded at Demerara Rise is a general feature of the equatorial latitudes or rather, and more likely, something controlled by local conditions.

**4 Astronomical Calibration of the Ypresian**

Time series analysis of early Eocene records that are used here already showed that the dominant cyclicity in multiple proxy records is related to eccentricity. The interval from PETM to ETM-2 is dominated by eccentricity modulated precession cycles that are an impressive recorder of earth orbital variations

through time and the climatic response to it (Lourens et al., 2005; Zachos et al., 2010; Littler et al., 2014). Data from this interval not only allowed to construct high precision cyclostratigraphies (Lourens et al., 2005; Westerhold et al., 2007) but also to test these astrochronologies as well as the theoretical astronomical solutions (Westerhold et al., 2007; Westerhold et al., 2012; Meyers 2015). As observed in a compilation of late Paleocene to early Eocene stable isotope data, the $\delta^{13}C$ variations in both bulk and benthic records from this time show the clear imprint of eccentricity variations with lighter values occurring in eccentricity maxima (Cramer et al., 2003; Lourens et al., 2005; Zachos et al., 2010; Littler et al., 2014). Hence, bulk and benthic stable isotope data helped to develop astrochronologies from the PETM up to ~49 Ma spanning Chron C22r to C24r at Leg 208 sites (Zachos et al., 2010; Littler et al., 2014; Laurentano et al., 2015; 2016). For Site 1258 a first cyclostratigraphic age model based on XRF core scanning Fe intensity was made from ETM-2 to the base of Chron C21n (~47 Ma) leading to revised estimates for the reversal ages from Chron C24n to C21n (Westerhold and Röhl 2009). Because of higher sedimentation rates than observed at Leg 208 sites, cyclicity in the Site 1258 XRF Fe data is mainly precession related with less pronounced modulation by eccentricity. Relatively higher sedimentation rates on the order of 3 to 5 cm / kyr lead to pronounced precession cycle recordings, whereas slower sedimentation rates tend to amplify the modulation of precession cycles, thus eccentricity. Compared to sites with lower sedimentation rate of 1 to 2 cm /kyr, the modulation of eccentricity is less pronounced in the XRF data of Site 1258. Both high-resolution bulk and benthic isotope records from 1258 revealed that multiple Eocene hyperthermal events exist, presumably forced by eccentricity paced threshold passing (Sexton et al., 2011; Kirtland-Turner et al., 2014). Recently, the eccentricity driven variations in benthic $\delta^{13}C$ of sites 1262 and 1263, combined with the bulk $\delta^{13}C$ data from 1258, were used to construct an astrochronology by tuning the 405 and 100 kyr eccentricity components to the Laskar 2010d orbital solution (Laskar et al., 2011a; Laurentano et al., 2015, 2016). Due to a major shift in $\delta^{13}C$ at ~260 rmcd 1263 two tuning options identifying two or three 405 kyr cycles in this interval were proposed. Integration of Site 1258 bulk isotope data and the best fit to the La2010d solution arguably lead to a preferred age model with two 405-kyr cycles in the above mentioned interval at Site 1263 (Laurentano et al., 2016).

Based on the previous effort we construct a new consistent age model now spanning the entire Ypresian for sites from Leg 208 and Site 1258 by combining a wealth of available information with new high-resolution data also making extensive use of the previously untouched spliced core images. Published benthic and bulk stable isotope data were combined for Leg 208 and Site 1258 (Fig. 2 and S7), plotted on the 1263 rmcd and detrended for long term trends (Fig. S9). Data have been linearly interpolated at 2-cm spacing and then were smoothed applying the IGOR Pro smooth operation using binomial (Gaussian) smoothing and 30001 points in the smoothing window. Benthic data from Site 1263 located in the disturbed drilling interval were removed from the combined record. The methods for time series analysis are those of Westerhold et al., (2015). Presence of the short and long eccentricity cycle in the isotope data is well documented (Zachos et al., 2010; Littler et al., 2014; Laurentano et al., 2015, 2016). Strong

eccentricity related cyclicity is clearly present in the evolutive wavelet power spectra of both isotope and XRF Fe data (Fig. S10) for the entire Ypresian applying the magnetostratigraphic interpretation and using either the CK95 (Cande and Kent 1995) or the GPTS2012 (Vandenberge et al., 2012) ages for reversals. We extracted the 405 and 100 kyr component of the data as detected in the evolutive spectra and plotted the filter over the data to investigate where the signal originates. The first order tuning was done identifying the 405-kyr cycle in all data consistently. The advantage of having the high-resolution XRF Fe data is the ability to detect distinct modulation of the amplitude in the 100 kyr period related to the 405 and 2.4 myr eccentricity cycle modulations. In supplementary figure S11, for example, around 242 and 265 rmcd 1263 the XRF Fe data from 1263 and 1267 show very low amplitude variations separated by four 405-kyr cycles. If the amplitude modulation in the data is mainly driven by eccentricity (Zachos et al., 2010; Littler et al., 2014) then these intervals represent the 2.4 myr eccentricity nodes with minor amplitude variations on the 100 kyr level. Identification of the 2.4 myr minima is very important because they function as major tie points for orbital tuning and test for consistency with astronomical solutions (Westerhold et al., 2012; 2015; Zeeden et al., 2013). The starting point for the stable 405 kyr cyclostratigraphy is eccentricity cycle 119 at 48.0 Ma which also represents a 2.4 myr eccentricity minimum (Fig. S11). Records presented here reconnect to astrochronologies that cover the Eocene cyclostratigraphic gap (Westerhold et al., 2015) from Site 1263.

**4.1 How many 405-kyr cycles represent Chron C23?**

Two enigmas had to be solved before a final stable 405-kyr cyclostratigraphy was set up. First, the question of whether two or three 405-kyr cycles are present at Site 1263 in the critical interval from 254-265 rmcd 1263. And second, which orbital solution is appropriate for more detailed orbital tuning on the short eccentricity level. The first issue is complicated by two $\delta^{13}$C shifts from 257-260 and at ~262 rmcd 1263 (Fig. 2). Between the shifts benthic $\delta^{13}$C data at 1263 do not show a clear eccentricity related cyclicity. Instead they reveal higher frequency cycles consistent with higher frequency cycles in XRF Fe data in 1258 and 1263. One argument to favour the two 405-kyr cycle model is that it is consistent with uniform sedimentation rate above and below and across this interval (Lauretano et al., 2016). While a reasonable assumption in the absence of other constraints, sedimentation rates at Leg 208 sites do change across the ETM-2 event (see Littler et al., 2014 Fig. 7 therein); decreasing from 1.5 to 0.7 cm/kyr at Site 1262. Using only the new magnetostratigraphy and the CK95 GPTS ages, sedimentation rates at Site 1263 drop from 2.6 to 2.0 cm/kyr across the C24n.3n/C24r reversal close to ETM-2 and from 1.75 to 1.1 cm/kyr across the C23n.2n/C23r reversal in the interval between the two $\delta^{13}$C shifts (Fig. S13). A decrease in sedimentation rates at both 1258 and 1263 is very likely to be located in C23r. Time series analysis provides some evidence that cycle thicknesses change between 265-254 rmcd 1263 (Fig. S11). Particularly relevant to this question is that previous astrochronologies for 1258 and 1263 (Westerhold and Röhl, 2009; Lauretano et al., 2016) were based on an under estimate of the duration Chron C23n from

1258 which led to the "50 Ma discrepancy", thus making it difficult to determine the correct number of 405 kyr cycles in C23. Comparison to GPTS2012 (Vandenberge et al., 2012) is compromised by the errors in the radioisotopic calibration points used in C22n (48.96 Ma ± 0.33) and C24n (52.93 Ma ± 0.23). Our new magnetostratigraphy is now more consistent with the width ratio observed in seafloor anomaly profiles from different ocean basins (Cande and Kent, 1995) that was used in GPTS2012.

The eccentricity modulated precession cycles at ODP Site 1258 can help to test the effects of different numbers of 405-kyr cycles in the interval from 68 to 95 rmcd spanning Chron C23 (Fig. S14). The thicknesses of high frequency cycles in 1258 change from ~45 cm/cycle to ~32 cm/cycle across 77-82 rmcd in 1258 (Westerhold and Röhl 2009, see Fig. S4 therein). Assuming that the average duration of the cycles is 21 kyr, the compression in cycle length translates into a decrease in sedimentation rate from 2.1 to 1.5 cm/kyr, as also seen in the new magnetostratigraphy. Applying the Lauretano et al., (2016) age models show durations for the cycles of 16 - 24 kyr (two 405-kyr cycles) and 23 - 35 kyr (three 405-kyr cycles) in this interval (Fig. S14). Due to the constant sedimentation rates in their model between 68 and 92 rmcd, the shift in cycle thickness is transformed into overall shortening of the precession cycles which is rather unrealistic. The option with two 405-kyr cycles seems to best fit the overall duration of 21 kyr for precession cycles and thus was chosen as the preferred age model by Lauretano et al. (2016) for 1258 and 1263. We have developed a new 405-kyr age model (Tab. S46) based on the time series analysis of multiple high-resolution records that add new tie-points between the interval from 68-92 rmcd previously not covered. Our new model proposed three rather than two 405-kyr cycles in this interval. As seen in the 1258 XRF Fe data the cycle thickness of the precession related cycles for the entire interval now show a duration of 21-23 kyr (Fig. S14B). This basic age model also reveals that the long cycles of ~50 cm length around 80 rmcd represent 41 kyr obliquity cycles.

The principal terms for the Precession of the Earth are given by the combination of the fundamental secular frequencies (g, s) of the Solar System and the precession frequency p (Laskar, 1993; Hinnov 2000; Laskar et al., 2004). Two periods are dominating Precession: the ~23 kyr period related to Jupiter (p+g5) and Venus (p+g2), and the ~19 kyr period related to Mars (p+g4) and Earth (p+g3) (Laskar et al., 1999). 50 million years ago, due to the evolution of the precession frequency p, the periods have been estimated to be ~22.5 and ~18.6 kyr (Laskar et al., 2004). In the 2.4 myr eccentricity minima, caused by the resonance between Earth and Mars (g3-g4), the ~19 kyr period is weak or absent and only the ~23 kyr period is present (Laskar et al., 2004). It has been suggested that the amplitude modulation of the XRF Fe signal of Site 1258 spanning 80-87 rmcd could represent a 2.4 myr eccentricity minimum (Westerhold and Röhl 2009). If this is correct, the dominating period for precession cycles recorded in the XRF Fe data for the interval from 68-92 rmcd should be towards 23 kyr rather than 19 kyr rejecting the two 405-kyr cycle model (Lauretano et al., 2016). Because of a change in phasing between XRF Fe and bulk stable isotope data at 87 rmcd cyclostratigraphy gets more complex at Site 1258. In contrast, Leg 208 sites show a consistent phase relation with decreased $\delta^{13}C$ values corresponding to higher Fe intensities. Both XRF

Fe from Leg 208 and the combined carbon stable isotopes show three 405-kyr cycles in the 265-254 rmcd 1263 interval (S11), and thus we propose a three cycle model (Fig. S12).

## 4.2 Which orbital solution applies best for astronomical tuning?

### 4.2.1 Visual evaluation and determination

After a stable 405-kyr cyclostratigraphic framework is established the orbital age model can be refined by tuning the carbon isotope data to an orbital solution on the short eccentricity level. Uncertainties in the ephemeris used to construct the orbital solutions currently limits their accuracy to roughly 48-50 Ma (Laskar et al., 2011a,b; Westerhold et al., 2015). Going beyond 50 Ma the modulation pattern of short eccentricity recorded in the geological data can help to find the correct orbital solution (Laskar et al.,

2004; 2011a,b). In particular, knowledge of the exact positions of the very long eccentricity minima, primary anchor for accurate orbital tuning (Shackleton and Crowhurst, 1997; Westerhold et al., 2007; Zeeden et al., 2014), can help to constrain astronomical solutions (Laskar et al., 2004; Westerhold et al., 2012). Beyond an age of 50 Ma the position of the very long eccentricity nodes in available orbital solutions (La2004 – Laskar et al., 2004; La2010 – Laskar et al., 2011a; La2011 – Laskar et al., 2011b)

are much more uncertain. Tuning to the La2010 or La2011 solution on the 100-kyr level is possible on the basis of a 405-kyr cyclostratigraphy, but should be evaluated with great care. Before a bulk stable isotope record for 1258 and a benthic stable isotope record for Site 1263 were generated, only Site 1258 XRF Fe data provided a record sufficient to attempt astronomical tuning of the early Eocene (Westerhold and Röhl 2009). We here use a larger and more diverse dataset from multiple sites on a stable 405-kyr

cyclostratigraphy to test which orbital solution is the most appropriate one for detailed tuning.

The compiled records from the Ypresian, in particular the XRF core scanning Fe intensity data, show prominent minima in eccentricity related modulation of the data in the intervals 212-220, 240-245, 260-267, 277-285 and 297-307 rmcd 1263 (Fig 3, S7). Starting with the first common node in the 2.4 myr cycle of La2010 and La2011 at 405 kyr cycles 118 to 119 (47.5-48.0 Ma) we go back in time and compare

the position of the very long eccentricity cycle minima to the data amplitude minima of the Ypresian records. Correlating the modulation minima at 212-220 rmcd 1263 to the node at 405 kyr cycles 118 to 119 anchors the records to the astronomically tuned middle to late Eocene time scale (Westerhold et al., 2015). With the application of the stable 405-kyr framework introduced above the preceding data modulation minima at 240-245, 260-267, 277-285 and 297-307 rmcd 1263 require very long eccentricity

minima at 405-kyr cycles 123-124, 128-129, 132, and 135-136 (Fig. 3). The first three minima can be observed in the different orbital solutions suggesting that basically all the solutions could be used back to 52 Ma as target curves for tuning. Beyond 52 Ma only the La2010b and La2010c solution show a minimum at 405-kyr cycle 132. Going further back in time to 56-57 Ma, the minimum before ETM-2 (Lourens et al., 2005; Westerhold et al., 2007; Meyers 2015) and the minimum before the PETM

(Westerhold et al., 2007; Zachos et al., 2010; Littler et al., 2014) in the data even match very long

eccentricity minima in La2010b and La2010c in 405-kyr cycle 135-136 (54.3-54.9 Ma) and 140 (56.4 Ma).

### 4.2.2 Statistical evaluation and determination

Extraction of the amplitude modulation (AM) using statistical methods like those implemented in the Astrochron package (Meyers 2014) or the ENVELOPE (Schulz et al. 1999) routine are important for independently testing the visual recognition of cycle patterns (Hinnov 2013, Hilgen et al. 2014). AM analysis on XRF core data using the ENVELOPE routine was applied at ODP Sites 1262 (52-60 Ma) and ODP Site 1258 (47-54 Ma) records in Westerhold et al. (2012) in order to search for the very long eccentricity minima. Meyers (2015) used a* values from ODP Site 1262 between PETM and ETM2 to test the existing astrochronologies (Lourens et al. 2005, Westerhold et al. 2007). Both methods (Astrochron, ENVELOPE) thus provide sound statistical testing of chronologies at ODP Site 1258 and for Leg 208 sites.

Following the approach of Zeeden et al. (2015) we extracted the short eccentricity cycle (100-kyr) and applied a broad bandpass filter (0.004 to 0.016 cycles/kyr; 250-62.5 kyr/cycle; Tukey window) and subsequently made a Hilbert transform to extract the AM using the Astrochron software package (Meyers 2015) for Sites 1258 and 1263 data. We applied the 405-kyr age model as a basic age model (Table S46). The resulting 405-kyr AM of the XRF Fe intensity data are plotted against the La2004, La2010, and La2011 orbital solutions (Fig. 4). The AM of the orbital solutions were extracted as described in Westerhold et al. (2012). For Site 1262 we plotted the 405-kyr AM of XRF Fe intensity data using the Option2 age model of Westerhold et al. (2012), which is almost identical to the updated 405-kyr age model presented here for the 53 to 58 Ma interval. We followed this procedure to demonstrate that similar results can be obtained with different approaches (Astrochron vs ENVELOPE).

The position of the very long eccentricity minima in the AM of XRF Fe intensity data in the interval from 46 to 59 Ma (blue bars in Fig. 4) best fits with minima in the La2010b and La2010c orbital solutions. In contrast, the minima do not match minima in the La2004 solution suggesting that this solution is not appropriate for testing geological data in this period of time. The La2010a-d and La2011 solutions fit to geological data back to 50 Ma. Beyond 50 Ma these solutions diverge (as discussed in Westerhold et al. 2012). Only the La2010b and La2010c solutions exhibit very long eccentricity minima at ~53.3 and ~54.5 Ma. The minimum at ~54.5 Ma is a very prominent feature in the data of the Leg 208 sites that has been intensively discussed (Lourens et al. 2005, Westerhold et al. 2007, Meyers 2015). The minimum at ~53.3 Ma is also detectable using the statistical methods but can be much better seen by visual inspection of the data (Fig. 2b, 3, and 5).

Quantitative evidence supporting the correct eccentricity node identification can also be derived from the emergence of obliquity cycles in the data at the nodes. Obliquity is not present in the Paleocene and early Eocene (Littler et al. 2014, Zeebe et al. 2017) of the investigated records. But Site 1258 Fe intensity data show some obliquity related cycles at around 80 rmcd (also see supplement Figure S14), and from 55 to

60 rmcd corresponding to the end of the very long eccentricity node at 52 Ma and the beginning of the very long eccentricity node at 50 Ma. At another potential node (48 Ma) Site 1258 Fe data do not clearly exhibit obliquity cycles, but low amplitude modulations of precession related cyclicity (Westerhold and Röhl (2009), see Figure 9 therein). Considering all these observations they provide some independent evidence for the existence of eccentricity nodes at 50 and 52 Ma. The nodes at ~53.3 and ~54.5 Ma show no prominent obliquity cycles in the Fe records as already discussed above and in Littler et al. (2014). Based on our observations and the statistical analysis (Hinnov 2013, Hilgen et al. 2014) we decided to fine-tune the records to the La2010b orbital solution (Fig. 5). It has to be noted here that there is hardly any difference between the La2010b and La2010c solution in the Ypresian from 46-56 Ma, therefore it does not matter if La2010b or La2010c is chosen as a target curve. Consequences of the match between orbital solutions and the geological data as well as the implications of the new age model for magnetostratigraphy needs to be discussed. The tie points for the tuned short eccentricity age model are given in table S47.

**4.2.3 Potential distortion by non-linear response of the climate system**

Non-linear response of climate is critical in the Ypresian. Multiple carbon cycle perturbations are documented in the stable isotope records (CIEs). These hyperthermal events are likely caused by massive releases of carbon to the ocean-atmosphere system including the dissolution of carbonates at the seafloor. Because the extent of the CIE's are scaled to the amount of carbon injected (Pagani et al., 2006) and the residence time of carbon is on the order of 100 kyr (Broecker and Peng, 1982), the events will influence the amplitude of the bulk and benthic stable carbon isotope data and thus any AM analysis of early Eocene records. On top of this, the added carbon leads also to dissolution of carbonates at the seafloor increasing the relative amount of non-carbonate material in the sediment (as detected by higher XRF Fe values). This will influence the statistical and visual recognition of cyclicity as discussed above. Modeling suggests that hyperthermals, except for the PETM, could be paced by eccentricity forcing of the carbon cycle with the amplitudes of the events being partly driven by the eccentricity amplitude itself (Kirtland-Turner et al. 2014). The carbon isotope data (Fig. 5) do show a good correspondence to the short eccentricity AM. In particular, the very long eccentricity minima are expressed as an interval of very low AM in the benthic carbon isotope data. Almost all hypothermal events occur outside the very long eccentricity minima. Only very minor excursions at C21r5, C22r5, and C23n.2nH1 coincide with these nodes, but with comparatively reduced CIE than hyperthermals suggesting these might not be hyperthermals in the end. Hyperthermal layers are very well documented in the XRF data by prominent peaks due to dissolution of carbonate and as larger CIEs are characterized by higher XRF Fe peaks. This tends to exaggerate the AM in the statistical analysis (Fig. 4). Hyperthermal events could be interpreted as amplifiers of the eccentricity amplitude with a bias toward higher amplitudes. Because the focus in identifying the best fit astronomical solution lies on the very long eccentricity minima, this ensures that the distortion by hyperthermal events is not significantly altering the results of our study.

# 5 Discussion

We have established the first complete astrochronology for the entire Ypresian Stage (YATS) compiling, integrating and synthesizing geochemical, bio- and magnetostratigraphic records at unprecedented precision. The result is a complex stratigraphy that can function as a reference allowing synchronization of paleoclimate records essential to understand cause and consequences of events in the early Eocene.

## 5.1 Geological evidence for chaotic behaviour of the solar system in the Ypresian?

Just recently the first geologic evidence confirming the chaotic behaviour of the Solar System through the identification of a chaotic resonance transition during the Coniacian (~85-87 Ma) was reported (Ma et al. 2017). Similarly, the new records presented here appear to provide additional observational confirmation of the past chaotic evolution of the Solar System, but as recently as ~52 Ma ago. Long term simulations of orbital motions to study the stability of the solar system propose a chaotic rather than quasiperiodic pattern of motion in the solar system (Laskar, 1989; Laskar et al., 2004; Laskar et al., 2011a). A macroscopic feature displaying the chaotic diffusion of the planetary orbits indentifiable in geological records is the transition from libration to circulation in the resonant argument related to $\Theta =$ (s4 - s3) - 2(g4 - g3), the combination of angles in the precession motion of the orbits of Earth and Mars (Laskar et al., 2004; 2011a; Pälike et al., 2004; see Westerhold et al., 2015 chapter 5.3 for detailed discussion). One needs to extract the AM of both obliquity and precession in a geological dataset in order to detect the transition from libration to circulation (Laskar 1999), which is almost impossible requiring a record that is both influenced (or driven) by high latitude and low latitude processes (Laskar et al (2011). Obliquity AM could be extracted from benthic $\delta_{18}O$ records, for example, if deep-sea temperature variations are continuously affected by obliquity. This is not the case for the Paleocene and early Eocene (Littler et al. 2014, Zeebe et al. 2017), thus investigation of the AM of obliquity is difficult with the currently available records. Laskar et al. (2011) recommended to search for a modulation of the g4 − g3 period, the ~2.4 myr eccentricity modulation. The transition from libration to circulation should be visible by a switch from a ~2.4 myr period to a ~1.2 myr period in the modulation of eccentricity and climatic precession (Laskar 1999, Laskar et al. 2004, Pälike et al. 2004, Laskar et al. 2011, Ma et al. 2017). Importantly, this could be transient with a switch back to ~2.4 myr shortly after (Laskar 1999).

With the new cyclostratigraphy based on the stable 405-kyr eccentricity cycle for the Ypresian, we can test if this switch is present in the observations. Previously, the identification of the very long eccentricity cycle in geological data for the Ypresian used the XRF Fe intensities from Sites 1258 and 1262 only (Westerhold et al., 2007; Westerhold and Röhl 2009). The multiple proxy data now provide a much clearer picture, as described in chapter 4.2, and show a very good match between geological data and the La2010b and La2010c numerical orbital solutions (Laskar et al., 2011a) for the Ypresian. The important feature shared between data and models share is the position of the very long eccentricity minima expressed as areas of low amplitude modulation in the data itself (Fig. 3, 4, 5). In the La2010b and La2010c solutions

the transition from libration to circulation occurs at ~52 Ma (Laskar et al., 2011a). Comparing orbital solutions with geological data indicates that the transition from libration to circulation occurred between 52 and 55 Ma. The AM minima in the data from 47 to 52 Ma are 2 to 2.4 myr spaced (Fig. 4, 5), from 52 to 55 Ma they are roughly 1.2 myr spaced, and after 55 Ma the spacing is ~2.4 myr.

None of the available orbital solutions perfectly fit to the geological data. However, it is important that we isolated the transition in the data, which is also present in the La2010b and La2010c solutions. The short eccentricity cycle pattern both in the solutions and the geological data will not match perfectly beyond 50 Ma where the uncertainty in the solutions increase (as discussed in Westerhold et al. 2012). Still the geological data and La2010b/c solutions are very similar from 53.5 Ma to the PETM. In the

interval from 51 to 52 Ma, the most difficult part to tune in the Ypresian, multiple hyperthermal events and the shift in carbon isotope data make a direct comparison much more difficult. It has to be noted here that the eccentricity solutions from La2010b/c might not be completely reliable in this interval. Despite the uncertainties we provide a tuned age model to La2010b/c because the match in the 53.5 Ma to the PETM interval is good enough to do so. If in doubt, the provided 405-kyr age model can still be used.

The point in time when the transition occurs in the numerical solutions is sensitive to the initial conditions of the planetary ephemeris used for back-calculation of the planetary motions. The initial conditions depend on the accuracy of the observational data used to make a least-square fit of the model to the data. The La2010b-c solutions used the INPOP08 Ephemeris (Fienga et al., 2009; Laskar et al., 2011a). In contrast, the La2010d solution used the INPOP06 Ephemeris (Fienga et al., 2008) and La2011 solution

the INPOP10a ephemeris (Fienga et al., 2011). The very similar long-term behaviour of INPOP06 and INPOP10a lead to the conclusion that these ephemerides are more stable than INPOP08 (Laskar et al., 2011b; Westerhold et al., 2012). Although the INPOP10a ephemeris is considered to be more accurate than INPOP08 (Fienga et al., 2011) the geological data provide evidence that the latter is closer to reality. Identifying the transition from libration to circulation at ~52 Ma in sediment archives is of great

importance, not only because it supports the theory on the chaotic nature of the solar system (Laskar, 1989), but it provides a benchmark to set the conditions for the gravitational model of the solar system (Laskar et al., 2004). It has to be noted that from an astronomer's point of view the La2010b and La2010c solution are considered less reliable because they used a less stable ephemeris (Fienga et al. 2011, Laskar et al. 2011). The same is true for the Laskar et al. (2004) solution but the Niobrara data suggest a better

fit to geological data than the nominal models of La2010d and La2011 (Ma et al. 2017). Clearly the La2004 solution is not consistent with the geological data from 46 to 58 Ma (Fig. 4), with implications for arguments of Ma et al. (2017). In fact, all eccentricity AM from the different theoretical astronomical solutions show a different and to some extent unusual behaviour between 52-54 Ma (Fig. S15). Our new findings should motivate efforts to further explore the differences in the ephemerides.

Our results support the application of La2010b or La2010c solution for eccentricity to construct astronomical age models back to 60 Ma. Beyond 60 Ma an accurate solution for eccentricity is not

possible at the moment (Laskar et al., 2011b), but the stable 405-kyr cycles will still provide a good target to establish astrochronologies (Laskar et al., 2004; 2011a,b). Here again, the geological data should be examined to find the very long eccentricity minima in very early Cenozoic and Mesozoic strata (Meyers 2015) and provide a landmark for developing more precise orbital solutions.

## 5.2 Solving the 50 Ma discrepancy in sea-floor spreading rates

Combining the new astrochronology with the revised magnetostratigraphy for the Ypresian allows us to consider the significance of the abrupt global increase in spreading rates in Chron C23n.2n, which is also known as the "50 Ma discrepancy" in the Paleogene Time Scale (Vandenberghe et al., 2012). The unusual peak in spreading rates in the South Atlantic (Fig. 6a) is independent of the age model used for the magnetostratigraphic interpretation of Site 1258 (Westerhold and Röhl, 2009; Westerhold et al., 2012; Lauretano et al., 2016). The new multi-site magnetostratigraphic data from Leg 208 sites reveals that Chron C23n is too short in the 1258 magnetostratigraphic interpretation (Suganuma and Ogg, 2006) and the likely reason for the computed peak in spreading rates. Application of the tuned age model to the integrated magnetostratigraphy results in a moderate but distinct jump in spreading rates at the C23n.2n/C23r reversal from 12 to 19 km/myr. The exact age of the increase cannot be precisely located using the reversal pattern only, but it probably occurred somewhere in C23r or C23n.2n between 51.0 and 52.5 Ma. Interestingly, this timing is synchronous with a major reorganization of the plate-mantle system (Whittaker et al., 2007), the subduction initiation of the Izu-Bonin-Mariana arc (Ishizuka et al., 2011) and the bend in the Hawaii-Emperor seamount chain (O'Connor et al., 2013). Changes in spreading rates in the interval from 51.0 - 52.5 Ma thus seem to be a global phenomenon pointing to a major common driving mechanism.

Astronomical calibration of the refined magnetostratigraphy in the marine records resulted in an improved Geomagnetic Polarity Time Scale (GPTS) for the Ypresian (Fig. 6b, Tab. 1). Duration of polarity zones are now consistent with the reversal thickness relationships observed in the South Atlantic and within error for the mean width of magnetic anomalies as published in table 4 of Cande and Kent (1992) (Tab. 2 and S46). The improved magnetostratigraphy shows a 376 kyr shorter duration for C22r, a 335 kyr longer duration for C23n.2n, and a 283 kyr shorter duration of C24n compared to previous marine records (Westerhold et al., 2015). The 50 Ma discrepancy in sea-floor spreading rates is now eliminated. It was clearly the effect of the difficult and incomplete identification of Chron C23n in Site 1258. Moreover, the durations are consistent within error to the GPTS2012 (Vandenberge et al., 2012) except for Chron C20r which is difficult to access due to the relatively large error for the radio-isotopic ages of the Mission Valley and the Montanari ash (Fig. 6) (for discussion see Westerhold et al., 2015). At this point more precise estimates of the mean width of magnetic anomalies and their error is required to be able to evaluate and improve the GPTS in the late Eocene. However, comparison to the GPTS models from terrestrial successions corroborates the finding (Westerhold et al., 2015) that the model of Tsukui and Clyde (2012)

more closely resembles the marine GPTS than the model of Smith et al., (2010; 2014) (Fig. 6). Issues in the correlation of the Layered tuff, Sixth tuff and Main tuff to local magnetostratigraphic records in the terrestrial records from the Green River Formation need to be resolved (see Tsukui and Clyde, 2012) to understand the current discrepancies in various terrestrial and marine GPTS models.

## 5.3 Defining the Age of the Top and Bottom of the Ypresian Stage

The GSSP of the Ypresian, that also marks the Paleocene/Eocene boundary, is defined at the base of the onset of the Paleocene-Eocene Thermal Maximum (PETM) carbon isotope excursion (CIE) (Aubry et al., 2007) about two thirds of the way down in magnetochron C24r (Westerhold et al., 2007) at base of Zone CNE1 where the Top of the calcareous nannofossil *Fasciculithus richardii* group and the Base of Calcareous nannofossil excursion taxa (CNET) occur (Westerhold et al., 2007; Agnini et al., 2014; Westerhold et al., 2015). The age and position of the onset of the PETM is confirmed by our study and thus needs no further discussion.

The Top of the Ypresian Stage (or Base of the Lutetian) is defined at the lowest occurrence of the calcareous nannofossil *Blackites inflatus* (CP12a/b boundary; Okada and Bukry, 1980) in the Gorrondatxe sea-cliff section in the Basque Country, northern Spain (Molina et al., 2011). The lowest occurrence of *B. inflatus* is reported ~819 kyr (39 precession cycles) after the base of Chron C21r (Bernaola et al., 2006; Payros et al., 2009), or 60% up in Chron C21r (C21r.6), leading to an age of ~47.8 Ma using GPTS2012 (Vandenberge et al., 2012). The new YATS absolute ages for the C21n/C21r (47.834 Ma) and C21r/C22n (48.994 Ma) boundaries results in a 500 kyr older Top of the Ypresian (= Base of *B. inflatus*, C21r.6 will be 48.3 Ma). However, when data from the Gorrondatxe reference section are compared with those from Site 1258 and SE Newfoundland Ridge (Norris et al., 2014) a large discrepancy emerges in the position of the Base of *B. inflatus* which is located near the bottom of Chron C21n outside the GSSP section (Fig. 5) and has a tuned age of 47.686 ± 0.065 Ma. According to Agnini et al., (2014), *B. inflatus* is more abundant in shallow-water to hemipelagic settings and rarely observed in pelagic settings and, even importantly, few data exist that provide a precise tie to magnetostratigraphy. Diachrony for some nannofossil events is well documented in the high-resolution astronomically calibrated bio-magnetostratigraphy for Leg 208 sites and Site 1258 presented here. But an offset of more than 600 kyr (48.3 - 47.686 Ma) between the IODP Atlantic sites and the Gorrondatxe section, resulting from the inconsistent position in C21n rather than C21r.6, suggests that either the Base of *B. inflatus* might not be the perfect global marker for the Ypresian-Lutetian boundary or the quality of magnetostratigraphic data in the Spanish section is not appropriate. Additional work on deep marine and hemipelagic records with good magnetostratigraphy is required to test the consistency of the position of this biohorizon and the relative ranking and spacing with the Base of *Nannotetrina cristata* that is reported to occur very close to the Base of *B. inflatus* and Chron C21n (Agnini et al, 2014; Norris et al., 2014).

## 5.4 Hyperthermal Events

The Ypresian Stage is of special interest to recent scientific work because it allows to study climate dynamics and feedbacks in a warm world (Zachos et al., 2008). In particular, the occurrence of multiple transient global warming events (hyperthermals) could help tremendously to understand the response of the climate system to a massive release of carbon to the ocean-atmosphere system (Dickens et al., 2003; Zachos et al., 2008; Lunt et al., 2011; Kirtland-Turner et al., 2014). A paired negative excursion in the carbon and oxygen isotope composition of bulk sediment and benthic foraminifera associated with a more clay rich layer, indicating dissolution of carbonate, are the characteristics of the deep-marine hyperthermal events (Zachos et al., 2005; Lourens et al., 2005; Leon-Rodriguez and Dickens, 2010). The early Eocene hyperthermals are paced by Earth's orbital eccentricity, except for the PETM (Cramer et al., 2003; Zachos et al., 2010; Sexton et al., 2011; Littler et al., 2014; Lauretano et al., 2015; 2016; Laurin et al., 2016). After the discovery of a large number of early Eocene hyperthermal events, a magnetochron based naming scheme was introduced (Sexton et al., 2011; Kirtland-Turner et al., 2014). Because the scheme used the inaccurate magnetostratigraphy of Site 1258 for Chrons C21 to C24, we updated the naming scheme of Kirtland-Turner et al., (2014). We also maintain the labelling system of Cramer et al., (2003) which was extended by Lauretano et al., (2016) for consistency. In table 3 we provide an overview of the naming schemes, astronomical age and position with respect to magnetostratigraphy.

The most important characteristics used to identify hyperthermal events is a paired excursion in $\delta^{13}C$ and $\delta^{18}O$ isotope data of preferably benthic foraminifera as in Sexton et al., (2011). However, some of the events defined in Sexton et al., (2011) fail this criterion but still have been considered as hyperthermals if the excursions are more negative than $1\sigma$ beyond the mean of its 2-Myr running average (Kirtland-Turner et al., 2014). This kind of test was applied to both benthic and bulk stable isotope data. It can be questioned if the bulk stable isotope data alone are sufficient to identify a hyperthermal event as several other factors can influence bulk sediment composition (e.g., production and preservation). To definitely identify hyperthermal events in the entire Ypresian paired high-resolution benthic stable $\delta^{13}C$ and $\delta^{18}O$ are needed. These records are available from PETM to the C24n.2rH1 event (Littler et al., 2014) and from the C22rH5 to C21rH5 event (Sexton et al., 2011). Unfortunately, the published record from Site 1263 (Lauretano et al., 2016) which spans from ETM-2 (H1) to the C22nH2 (W) event only provides benthic $\delta^{13}C$ data. To definitely identify hyperthermals in the Ypresian more paired benthic stable isotope data are needed. Hence, we refrain from changing or even making up a new naming scheme. We synthesized the available schemes and here adopted the naming in table 3 as well as providing their astronomical ages from the new astrochronology (Fig. 5).

## 6 Conclusions

A new complex cyclostratigraphy and refined bio-chemo-magnetostratigraphy has been developed for key ODP records spanning the entire Ypresian Stage from 56 to 47 Ma. Detailed correlation of ODP Sites 1258, 1262, 1263, 1265, and 1267 using the new CODD macros software tool revealed a 3-400 kyr condensed interval in the Leg 208 sites during Chron C22n. New characteristic remanent magnetization (ChRM) data from four Leg 208 sites show an overall consistent magnetostratigraphy refining the Ypreasian Geomagnetic Polarity Time Scale. Multi-site ChRM data correlated on a cm to dm scale suggest that a magnetostratigraphic record from a single site might contain significant errors due to coring disturbance. Cyclic variations in synthesized XRF core scanning and stable isotope data as well as lithological changes apparent in core images have been successfully used to refine previous astrochronologies and construct the first complete Ypresian Astronomical Time Scale (YATS). In absence of independent, high-precision time control like radio-isotopic dates at Leg 207 and 208 sediments, our study clearly validates that it is crucial to combine multiple records from multiple regions to help safeguard against incompleteness that is otherwise difficult to assess both qualitatively and quantitatively. The YATS not only provides updated absolute ages for bio- and magnetostratigraphy but also a comprehensive list of the early Eocene hyperthermal events. The new astronomically calibrated Ypresian GPTS resolves the "50 Ma discrepancy" which was primarily caused by the imprecise magnetostratigraphy of Site 1258. Comparing the eccentricity related cyclic pattern in XRF core scanning and stable carbon isotope data to numerical orbital solutions suggests that the transition from libration to circulation as predicted by the La2010b solution occurred ~52 Ma ago. This adds to the geological evidence for the chaotic nature in the evolution of the solar system.

### Acknowledgments

We thank Henning Kuhnert and is team for stable isotope analyses at MARUM, Alex Wülbers and Walter Hale at the IODP Bremen Core Repository (BCR) for core handling, and Vera Lukies (MARUM) for assistance with XRF core scanning. We thank Frits Hilgen, Stephen R. Meyers and an anonymous referee for their constructive critical reviews. This research used samples and data provided by the International Ocean Discovery Program (IODP). IODP is sponsored by the US National Science Foundation (NSF) and participating countries. Financial support for this research was provided by the Deutsche Forschungsgemeinschaft (DFG) and the National Science Foundation (NSF). The data reported in this paper are tabulated in the Supplement and archived in the Pangaea (www.pangaea.de) database.

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

## Figures

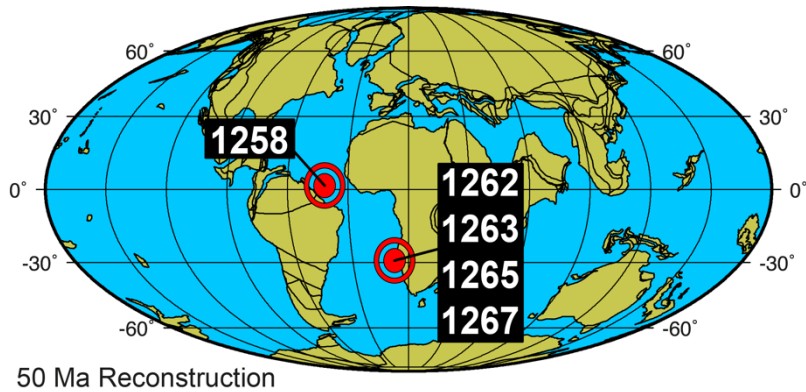

Ma Reconstruction

**Figure 1:** Location of ODP Site1258 and Leg 208 sites on a 50 Ma paleogeographic reconstruction in Mollweide projection (from http://www.odsn.de).

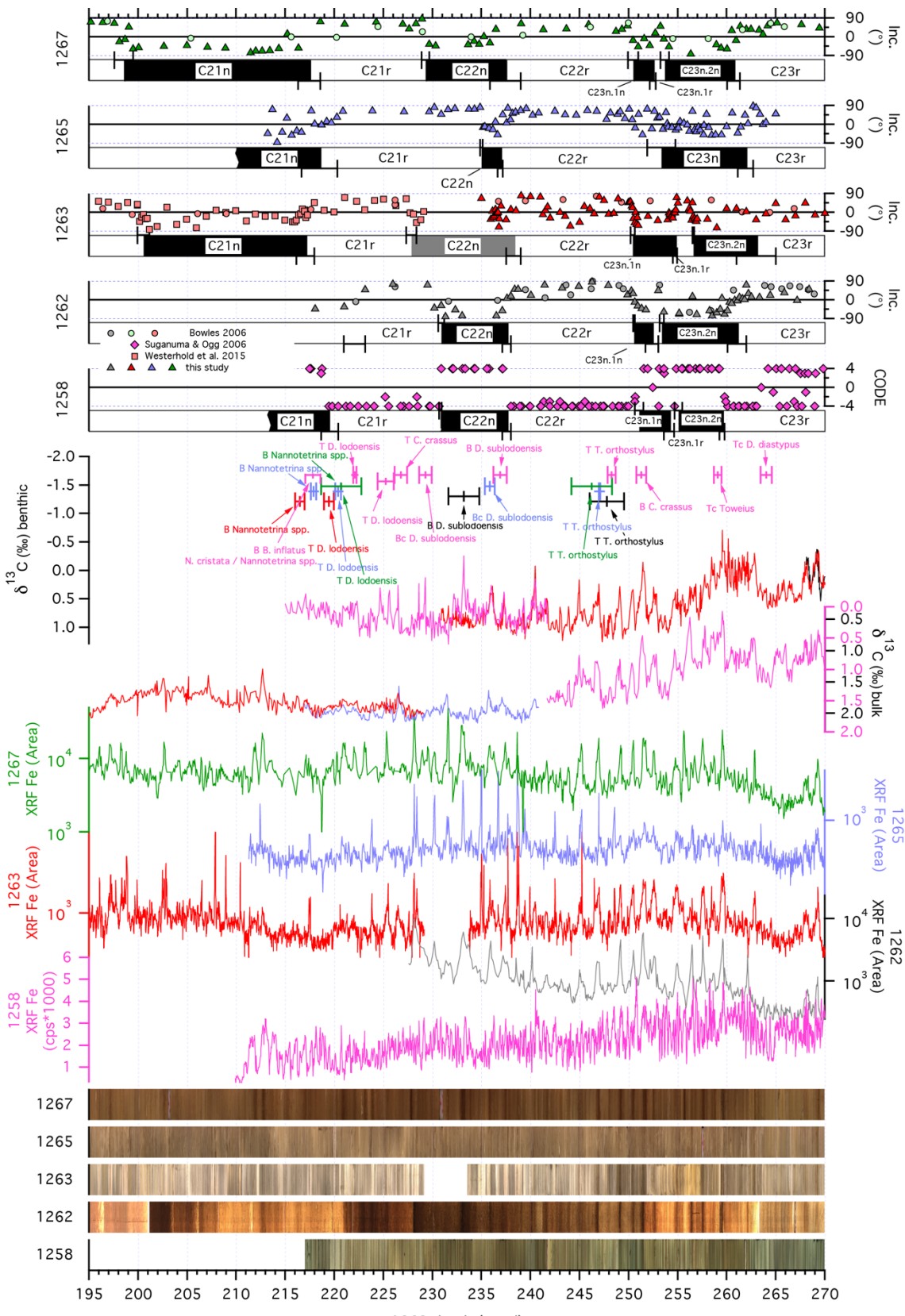

**Figure 2a:** Overview of bio- and magnetostratigraphic data, XRF core scanning Fe intensity data and core images from ODP Site 1258, 1262, 1263, 1265 and 1267 from 195-270 revised meters composite depth of Site 1263. Upper five panels show the inclination from charateristic remanent magnetization investigations from this and previous studies with uncertain polarity interpretation in gray. The position of calcareous nannofossil events including the depth error is plotted for all sites followed by the compiled benthic and bulk **δ13C** data and XRF core scanning Fe intensities. For detailed source of data see manuscript text. Purple – 1258, Black and Grey – 1262; Red – 1263; Blue – 1265, Green – 1267.

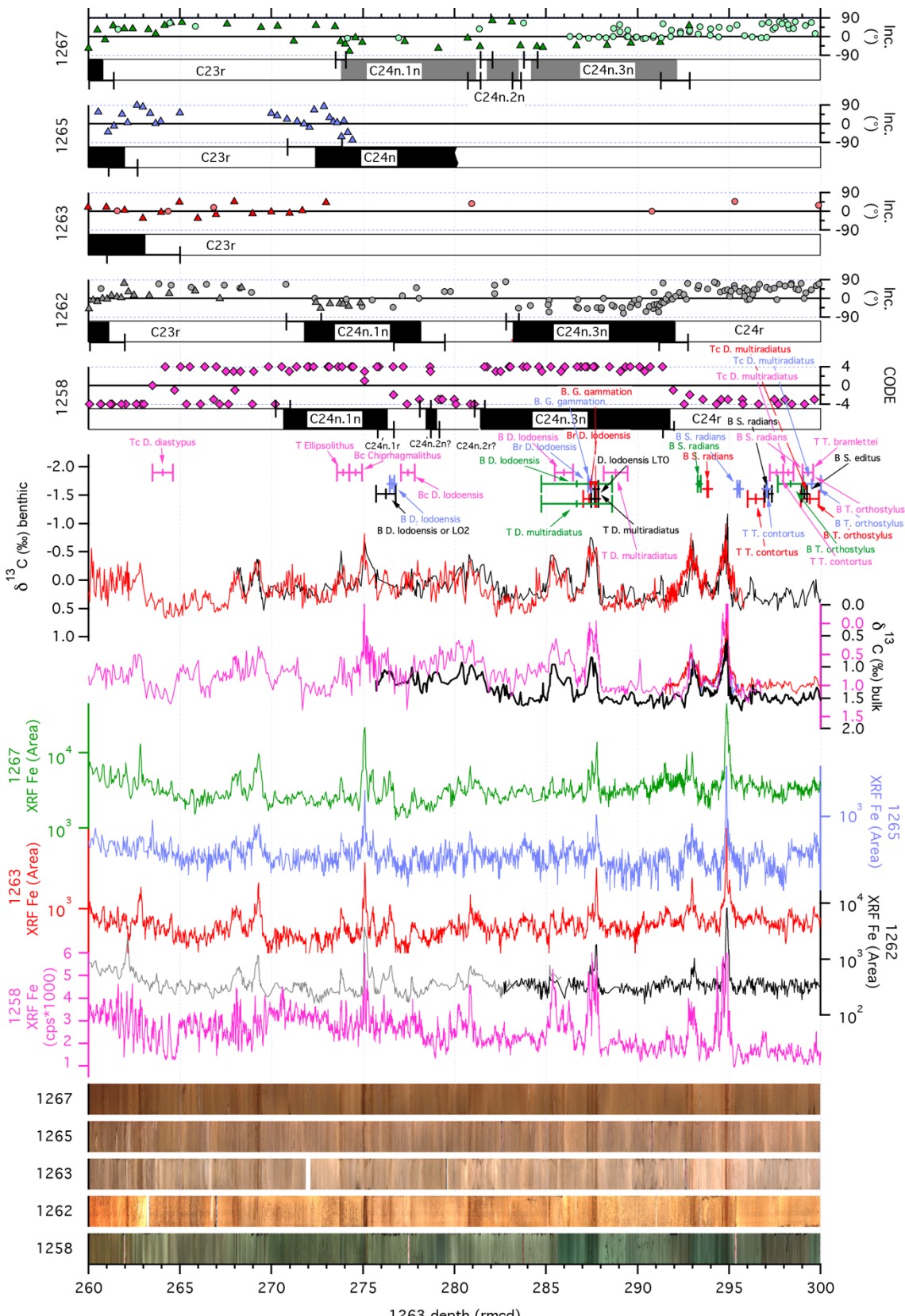

**Figure 2b:** Overview of bio- and magnetostratigraphic data, XRF core scanning Fe intensity data and core images from ODP Site 1258, 1262, 1263, 1265 and 1267 from 260-300 revised meters composite depth of Site 1263. See Figure 2a. Note: Overview from 300-340 rmcd 1263 is in Figure S7 of the supplement.

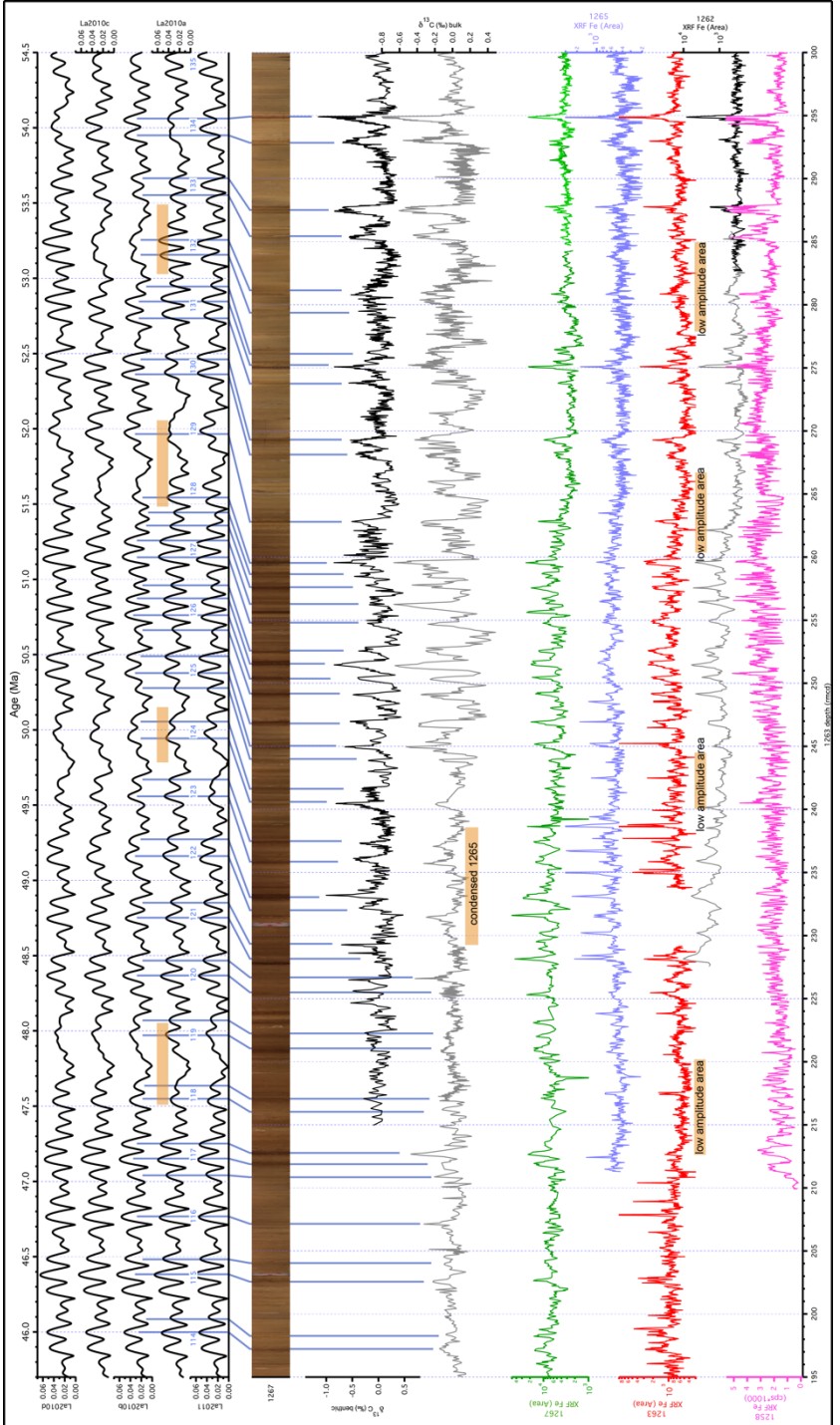

**Figure 3:** Correlation ties for astronomical calibration from 54.5-46 Ma. Top panel shows the numerical solutions La2011 (Laskar et al., 2011) and La2010a-d (Laskar et al., 2011a) including the 405-kyr cycle number counted backwards from today (Wade and Pälike, 2004). Below the Site 1267 core image, which shows the best expressed dark layer pattern of all sites, we plotted the detrended benthic (black) and bulk (grey) combined **δ¹³C** data as well as XRF Fe intensity records on rmcd 1263. Areas of low amplitude modulation in the data are marked in the XRF data for comparison. The blue lines show the tie points for the astronomically tuned age model correlating La2010b eccentricity maxima to lighter (more negative) **δ¹³C** values. Note the condensed interval in the bulk Site 1265 **δ¹³C** data, which cannot be tuned to the orbital solutions. For detailed discussion see text.

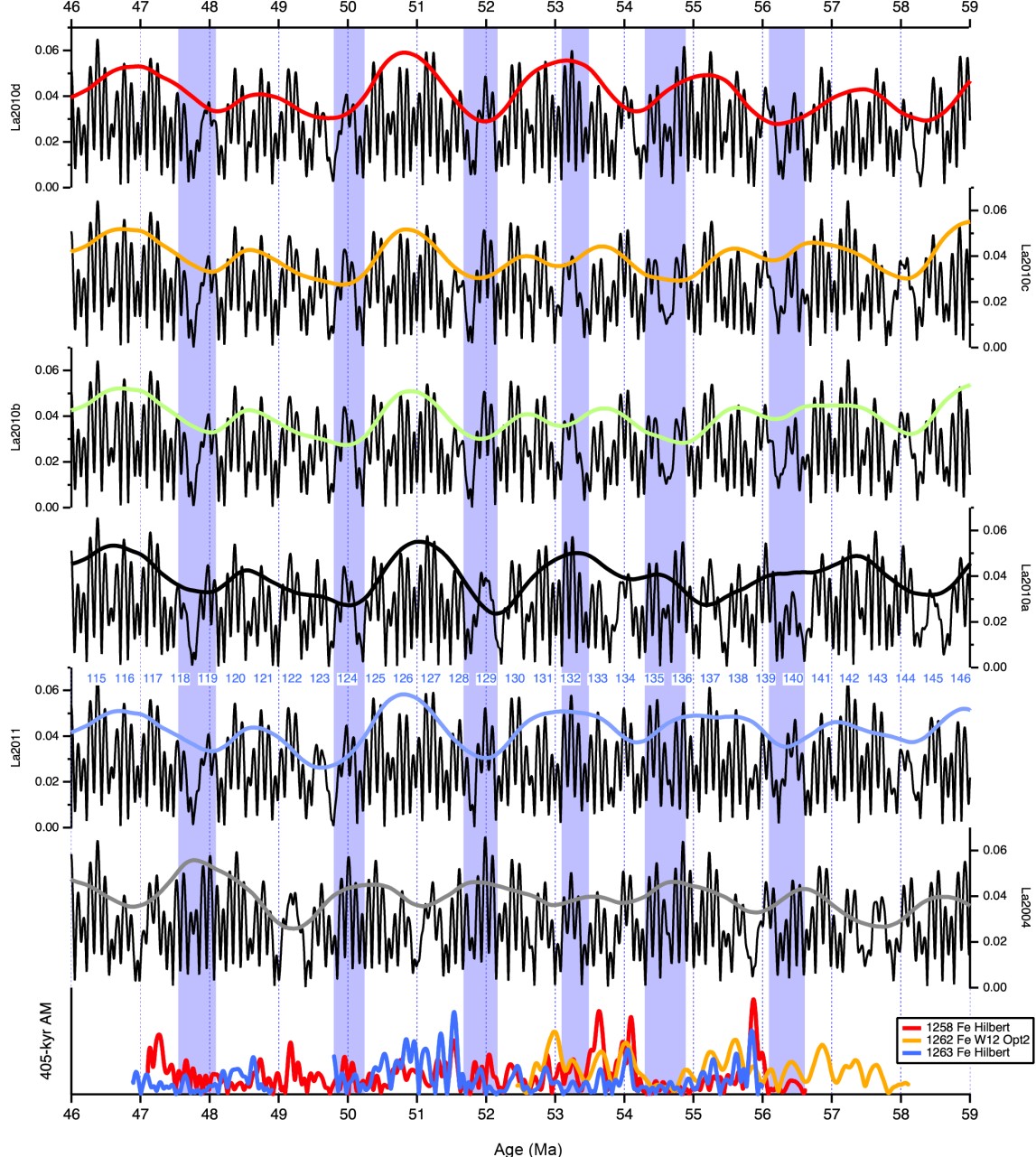

**Figure 4**: Comparison of the amplitude modulation (AM) of the short eccentricity cycle between the La2004, La2010, and La2011 orbital solutions and Fe intensity data from ODP Sites 1258 (red), 1262 (orange) and 1263 (blue). For the orbital solutions we also plotted the 405-kyr AM. The short eccentricity AM of Sites 1258, 1262 and 1263 Fe intensity data are plotted on the 405-kyr scale model. The very long eccentricity minima are highlighted by light blue bars in the orbital solutions and the Fe intensity data. Statistical and visual recognition of cycle pattern suggest that the La2010b and La2010c solutions are most consistent with the geological data.

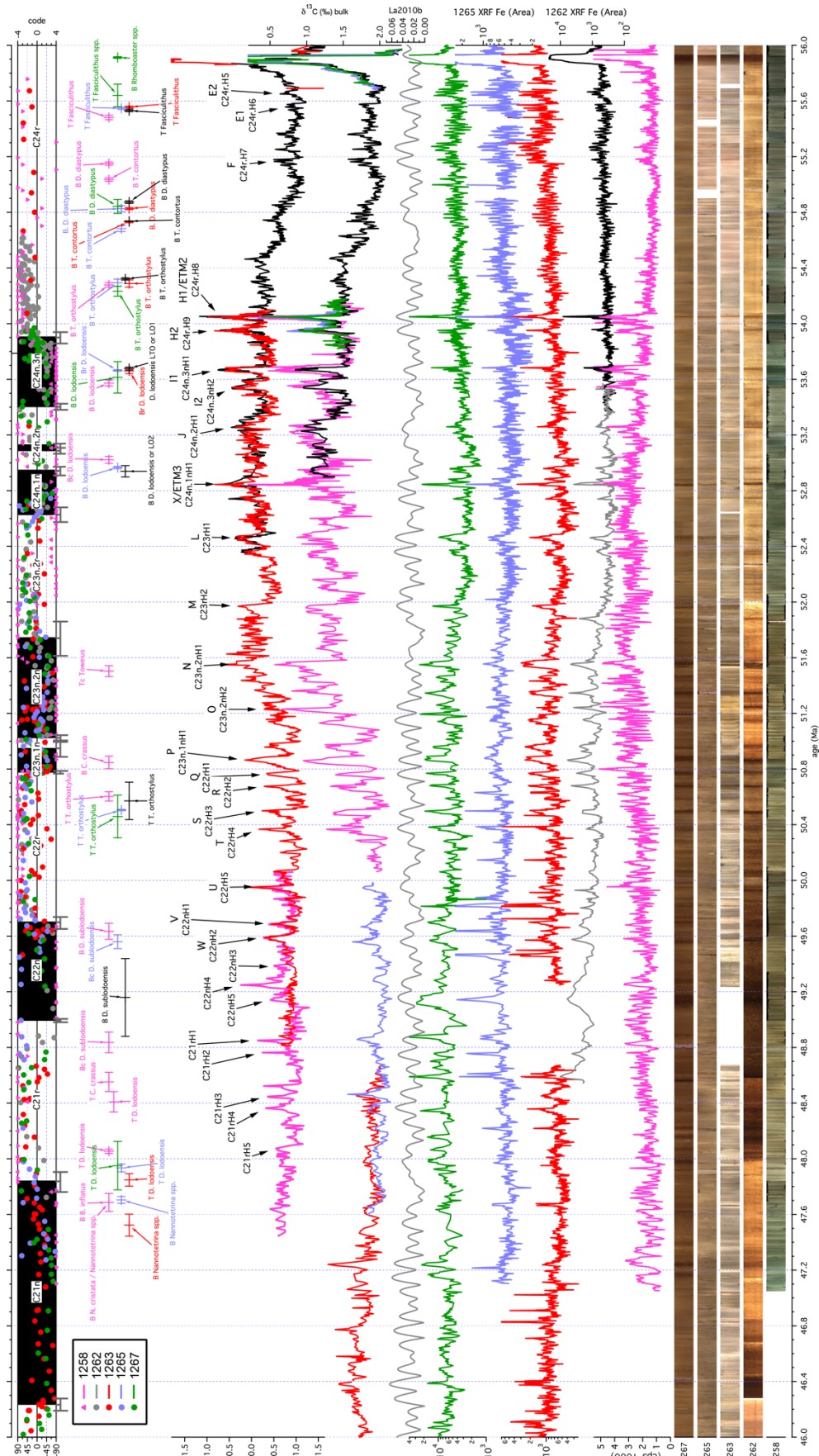

**Figure 5:** Bio- and magnetostratigraphic data, benthic and bulk stable isotopes, XRF core scanning Fe intensities and core images on the new astrochronology for the Ypresian Stage. Inclination data from ChRM analysis of each site from Leg 208 are shown, for Site 1258 the results of Suganuma and Ogg 2006 are given as a code between -4 and 4 with negative values indicating reversed polarity. The final synthesized magnetostratigraphy (Tab. S45) includes the error for the reversals (bars). Also shown is the La2010b (Laskar et al., 2011a) orbital solution by the central grey line. Note the extremely good match between orbital solution and amplitude modulation in the various XRF and stable isotope records. Hyperthermal events have been labeled according to table 3.

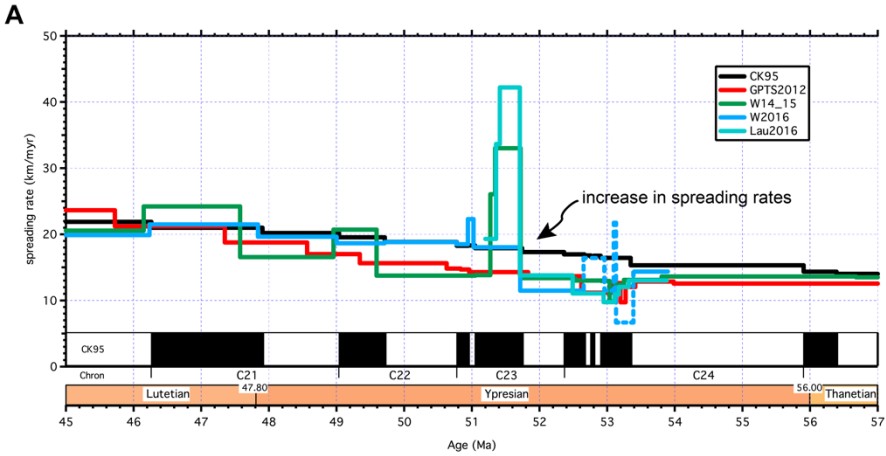

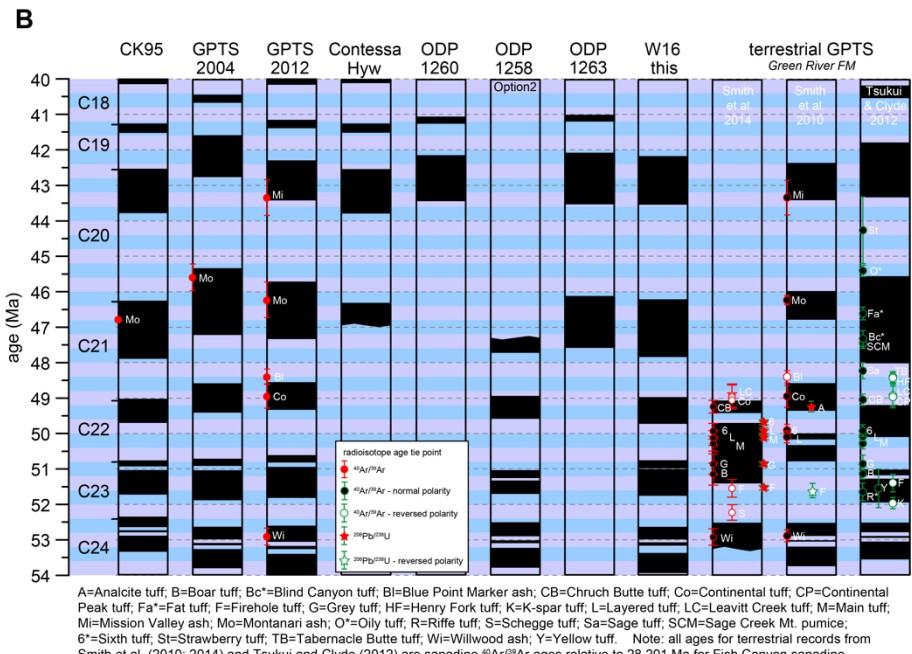

A=Analcite tuff; B=Boar tuff; Bc*=Blind Canyon tuff; Bl=Blue Point Marker ash; CB=Chruch Butte tuff; Co=Continental tuff; CP=Continental Peak tuff; Fa*=Fat tuff; F=Firehole tuff; G=Grey tuff; HF=Henry Fork tuff; K=K-spar tuff; L=Layered tuff; LC=Leavitt Creek tuff; M=Main tuff; Mi=Mission Valley ash; Mo=Montanari ash; O*=Oily tuff; R=Riffe tuff; S=Schegge tuff; Sa=Sage tuff; SCM=Sage Creek Mt. pumice; 6*=Sixth tuff; St=Strawberry tuff; TB=Tabernacle Butte tuff; Wi=Willwood ash; Y=Yellow tuff.    Note: all ages for terrestrial records from Smith et al. (2010; 2014) and Tsukui and Clyde (2012) are sanadine $^{40}Ar/^{39}Ar$ ages relative to 28.201 Ma for Fish Canyon sanadine (Kuiper et al., 2008); ashes markt by * are biotite ages; Analcite tuff is U/Pb age; U/Pb age for 6, L, M, G and F from Machlus et al. (2015)

**Figure 6: A.** Comparison of calculated spreading rates for the synthetic South Atlantic profile of Cande and Kent (1992) based on the age model of Cande and Kent (1995), GPTS2012 (Vandenberghe et al., 2012), Westerhold et al., (2015), Lauretano et al., (2016) and this study (W2016). The GPTS of CK95 is also plotted for reference. Note the increase in spreading rates in the South Atlantic at ~52 Ma based on the new GPTS presented in this study. **B.** Geomagnetic polarity timescale of CK95 (Cande and Kent, 1995), GPTS2004 (Ogg and Smith, 2004) and GPTS2012 (Ogg, 2012; Vandenberghe et al., 2012) compared to astronomical calibrations of magnetochrons from Contessa Highway (Jovane et al., 2010), Site 1260 (Westerhold and Röhl, 2013), Site 1258 (Westerhold and Röhl, 2009; Westerhold et al., 2012), Site 1263 (Westerhold et al., 2015) and the new Ypresian GPTS (this study, W16) from 40-54 Ma. Terrestrial calibration of the GPTS from the Green River Formation (Smith et al., 2010; 2014; Tsukui and Clyde, 2012) are also shown. Small red dots with error bars mark the radioisotopic calibration points used for CK95, GPTS2004, GPTS2012, and Smith et al., (2010); green circles show calibration points for the terrestrial sections used by Tsukui and Clyde (2012). The overview demonstrates the now consistent Eocene GPTS from 30 to 54 Ma from ODP stratigraphic data and the discrepancy in the terrestrial GPTS. See text for discussion.

**Tables**

**Table 1** Comparison of magnetochron boundary ages in million years

| Chron | standard GPTS | | | tuned[*] GPTS | | | | | this study[†] |
|---|---|---|---|---|---|---|---|---|---|
| | CK95 | GPTS 2004 | GPTS 2012 | PEAT Sites[#] | Contessa Hyw | ODP Site 1260 | ODP Site 1258 opt.2 | ODP Site 1263 | Leg208 Site1258 |
| C18n.2n (o) | 40.130 | 39.464 | 40.145 | 40.076 ± 0.005 | 40.120 | | | | |
| C19n (y) | 41.257 | 40.439 | 41.154 | *41.075 ± 0.007* | 41.250 | 41.061 ± 0.009 | | 41.030 ± 0.013 | |
| C19n (o) | 41.521 | 40.671 | 41.390 | *41.306 ± 0.005* | 41.510 | 41.261 ± 0.004 | | 41.180 ± 0.011 | |
| C20n (y) | 42.536 | 41.590 | 42.301 | *42.188 ± 0.015* | 42.540 | 42.152 ± 0.007 | | 42.107 ± 0.013 | 42.196 ± 0.013 |
| C20n (o) | 43.789 | 42.774 | 43.432 | | 43.790 | 43.449 ± 0.018 | | 43.517 ± 0.011 | 43.507 ± 0.011 |
| C21n (y) | 46.264 | 45.346 | 45.724 | | 46.310 | | | 46.151 ± 0.009 | 46.235 ± 0.044 |
| C21n (o) | 47.906 | 47.235 | 47.349 | | | | 47.723 ± 0.118 | 47.575 ± 0.018 | 47.834 ± 0.072 |
| C22n (y) | 49.037 | 48.599 | 48.566 | | | | 48.954 ± 0.016 | | 48.994 ± 0.012 |
| C22n (o) | 49.714 | 49.427 | 49.344 | | | | 49.593 ± 0.042 | | 49.695 ± 0.043 |
| C23n.1n (y) | 50.778 | 50.730 | 50.628 | | | | 51.051 ± 0.021 | | 50.777 ± 0.010 |
| C23n.1n (o) | 50.946 | 50.932 | 50.835 | | | | 51.273 ± 0.039 | | 50.942 ± 0.054 |
| C23n.2n (y) | 51.047 | 51.057 | 50.961 | | | | 51.344 ± 0.032 | | 51.025 ± 0.019 |
| C23n.2n (o) | 51.743 | 51.901 | 51.833 | | | | 51.721 ± 0.023 | | 51.737 ± 0.123 |
| C24n.1n (y) | 52.364 | 52.648 | 52.620 | | | | 52.525 ± 0.023 | | 52.628 ± 0.053 |
| C24n.1n (o) | 52.663 | 53.004 | 53.074 | | | | 52.915 ± 0.029 | | 52.941 ± 0.031 |
| C24n.2n (y) | 52.757 | 53.116 | 53.199 | | | | 53.037 | | 53.087 ± 0.021 |
| C24n.2n (o) | 52.801 | 53.167 | 53.274 | | | | 53.111 | | 53.123 ± 0.015 |
| C24n.3n (y) | 52.903 | 53.286 | 53.416 | | | | 53.249 ± 0.017 | | 53.403 ± 0.022 |
| C24n.3n (o) | 53.347 | 53.808 | 53.983 | | | | 53.806 ± 0.020 | | 53.899 ± 0.041 |

[*] tuned GPTS compilation from Westerhold et al., (2015)
[†] tuned to the orbital solution La2010b (Laskar et al., 2011a)
[#] combined ages based on Pacific Equatorial Age Transect Sites 1218, U1333 and U1334 (Westerhold et al., 2014)

**Table 2** Comparison of magnetochron boundary durations in million years

| Chron | standard GPTS | | | tuned[*] GPTS | | | | | this study[†] |
|---|---|---|---|---|---|---|---|---|---|
| | CK95 | GPTS 2004 | GPTS 2012 | PEAT Sites[#] | Contessa Hyw | ODP Site 1260 | ODP Site 1258 opt.2 | ODP Site 1263 | Leg208 Site1258 |
| C18n.2r | 1.127 | 0.975 | 1.009 | 0.999 ± 0.012 | | | | | |
| C19n | 0.264 | 0.232 | 0.236 | 0.231 ± 0.012 | 0.260 | 0.200 ± 0.007 | | 0.150 ± 0.024 | |
| C19r | 1.015 | 0.919 | 0.911 | 0.882 ± 0.020 | 1.030 | 0.891 ± 0.006 | | 0.927 ± 0.024 | |
| C20n | 1.253 | 1.184 | 1.131 | | 1.250 | 1.297 ± 0.013 | | 1.410 ± 0.024 | 1.311 ± 0.011 |
| C20r | 2.475 | 2.572 | 2.292 | | 2.520 | | | 2.634 ± 0.020 | 2.727 ± 0.044 |
| C21n | 1.642 | 1.889 | 1.625 | | | | | 1.424 ± 0.027 | 1.599 ± 0.072 |
| C21r | 1.131 | 1.364 | 1.217 | | | | 1.231 ± 0.134 | | 1.161 ± 0.012 |
| C22n | 0.677 | 0.828 | 0.778 | | | | 0.639 ± 0.058 | | 0.700 ± 0.043 |
| C22r | 1.064 | 1.303 | 1.284 | | | | 1.458 ± 0.063 | | 1.082 ± 0.010 |
| C23n.1n | 0.168 | 0.202 | 0.207 | | | | 0.222 ± 0.060 | | 0.166 ± 0.054 |
| C23n.1r | 0.101 | 0.125 | 0.126 | | | | 0.071 ± 0.071 | | 0.083 ± 0.019 |
| C23n.2n | 0.696 | 0.844 | 0.872 | | | | 0.377 ± 0.055 | | 0.712 ± 0.123 |
| C23n | 0.965 | 1.205 | 1.200 | | | | 0.670 ± 0.044 | | 0.961 ± 0.133 |
| C23r | 0.621 | 0.747 | 0.787 | | | | 0.804 ± 0.046 | | 0.890 ± 0.053 |
| C24n.1n | 0.299 | 0.356 | 0.454 | | | | 0.390 ± 0.052 | | 0.314 ± 0.031 |
| C24n.1r | 0.094 | 0.112 | 0.125 | | | | 0.122 | | 0.145 ± 0.021 |
| C24n.2n | 0.044 | 0.051 | 0.075 | | | | 0.074 | | 0.036 ± 0.015 |
| C24n.2r | 0.102 | 0.119 | 0.142 | | | | 0.138 | | 0.280 ± 0.022 |
| C24n.3n | 0.444 | 0.522 | 0.567 | | | | 0.557 ± 0.037 | | 0.496 ± 0.041 |
| C24n | 0.983 | 1.363 | 1.360 | | | | 1.554 ± 0.043 | | 1.271 ± 0.094 |

[*] tuned GPTS compilation from Westerhold et al., (2015)
[†] tuned to the orbital solution La2010b (Laskar et al., 2011a)
[#] combined ages based on Pacific Equatorial Age Transect Sites 1218, U1333 and U1334 (Westerhold et al., 2014)

**Table 3** Overview of naming and age of hyperthermal events

| Hyperthermal naming scheme | | | | tuned age Ma La2010b | Comment |
|---|---|---|---|---|---|
| Sexton et al., (2011) | Kirtland-Turner et al., (2014) | Cramer et al., (2003) Lauretano et al., (2016) | this study | | |
| C21rH6 | C21rH5 | - | C21rH5 | 48.075 | |
| C21rH5 | C21rH4 | - | C21rH4 | 48.365 | |
| C21rH4 | C21rH3 | - | C21rH3 | 48.450 | |
| C21rH3 | - | - | - | - | not crossing the 1σ level of Kirtland-Turner et al., 2014 |
| C21rH2 | C21rH2 | - | C21rH2 | 48.765 | |
| C21rH1 | C21rH1 | - | C21rH1 | 48.850 | |
| C22nH4 | C22nH5 | - | C22nH5 | 49.140 | |
| C22nH3 | C22nH4 | - | C22nH4 | 49.250 | |
| - | C22nH3 | - | C22nH3 | 49.385 | added by Kirtland-Turner et al., 2014 |
| C22nH2 | C22nH2 | W | C22nH2 | 49.585 | |
| C22nH1 | C22nH1 | V | C22nH1 | 49.685 | |
| C22rH3 | C22rH6 | - | - | - | no clear d13C and d18O excursion |
| C22rH2 | C22rH5 | U | C22rH5 | 49.950 | |
| C22rH1 | - | - | - | - | not crossing the 1σ level of Kirtland-Turner et al., 2014 |
| - | C22rH4 | T | C22rH4 | 50.370 | |
| - | C22rH3 | S | C22rH3 | 50.485 | |
| - | C22rH2 | R | C22rH2 | 50.670 | |
| - | C22rH1 | Q | C22rH1 | 50.760 | |
| - | C23n.1nH1 | P | C23n.1nH1 | 50.865 | |
| - | C23n.2nH1 | O | C23n.2nH2 | 51.230 | name adjusted to revised magnetostratigraphy |
| - | C23rH3 | N | C23n.2nH1 | 51.550 | name adjusted to revised magnetostratigraphy |
| - | C23rH2 | M | C23rH2 | 51.970 | |
| - | C23rH1 | L | C23rH1 | 52.460 | |
| - | C24n.1nH1 | K/ETM-3 | C24n.1nH1 | 52.845 | |
| - | - | J | C24n.2rH1 | 53.260 | not crossing the 1σ level of Kirtland-Turner et al., 2014 |
| - | C24n.3nH2 | I2 | C24n.3nH2 | 53.545 | |
| - | C24n.3nH1 | I1 | C24n.3nH1 | 53.665 | |
| - | C24rH9 | H2 | C24rH9 | 53.950 | |
| - | C24rH8 | H1/ETM-2 | C24rH8 | 54.050 | |
| - | C24rH7 | F | C24rH7 | 55.165 | |
| - | C24rH6 | E2 | C24rH6 | 55.555 | |
| - | C24rH5 | E1 | C24rH5 | 55.650 | |
| - | PETM | PETM | PETM | 55.930 | |