# Peer review of "Astronomical Calibration of the Ypresian Time Scale: Implications for Seafloor Spreading Rates and the Chaotic Behaviour of the Solar System?"

_Climate of the Past, 2017_

## Referee Comment (RC1) · F. Hilgen (Referee) · 1 Mar 2017

The ms provides an important contribution to the construction of an/the Astronomical Time Scale for the early Paleogene and its implications for seafloor spreading rates and the chaotic behavior of the Solar System, and is as such well suitable for publication in Climate of the Past. The presented integrated high-resolution stratigraphy is truly impressive and it is good to see that some naughty issues now seem to be finally solved. Nevertheless, several issues have to be addressed before the ms can be accepted for publication.

Major points

1) Statistical identification of the very long period ∼2-Myr eccentricity minima. The eccentricity nodes associated with the very long, 2-Myr, eccentricity cycle are now visually determined in their proxy records, but preferably they should also be pinpointed by means of statistical analysis, such as complex amplitude modulation or the method outlined in Meyers (2015). Such an independent statistical confirmation of the position of these nodes is critically important next to their visual determination to convince the reader of the correctness of the conclusions drawn in the ms. Otherwise the authors have to clearly state why they did not carry out the necessary and logical statistical analysis. Especially the statistical method introduced by Meyers (2015) seems very helpful to reconstruct eccentricity and capture the nodes associated with the very long term eccentricity cycle, so the question is why such statistical methods have not been applied. I guess that the authors may well have given it a try, so in that case, why was it not included in the ms (even to show that these statistical approaches do not work well in this particular case)? The authors may thus wish to discuss in some detail the (dis)advantages of a visual versus a statistical approach. This topic is discussed in some detail by Hinnov (2013) and has been presented in more detail by Steve Meyers in some of his presentations. The point here is that visual recognition of cycle patterns albeit being subjective can be considered an expert system, being able to identify distortions of the signal that are very commonly encountered in cyclostratigraphic records (see e.g. point 2) and that may cause problems when applying statistical techniques (see also Hilgen et al., 2015)

2) Potential distortion by non-linear response of the climate system. The authors have to explain that the amplitude changes they see in their proxies are related to the amplitude of the ∼100-kyr eccentricity cycle and not caused by a non-linear response of the climate system to the eccentricity forcing through associated changes in the precession amplitude. This issue might become critical when dealing with the proxy expression of early Eocene hyperthermals. Evidently, the "distortion" caused by such a non-linear

response will also have consequences for the outcome of the statistics as I guess that these usually start from linear relationships. This issue has to be addressed in the discussion.

3) Exclusion of expression of 1.0-Myr eccentricity cycle. The authors claim that they have detected the expression of the transition from libration to circulation of the very long period eccentricity cycle in the geological record. However, to be sure, they have to address the following two points. In the first place, what is the role of the relatively strong ∼1.0 Myr eccentricity component (related to g5-g1, and can also be written as a combination of ∼100-kyr components) , especially in determining the node around 53 Ma that they attribute to the ∼2-Myr cycle. They should thus make clear what the exact expression of the ∼2-Myr cycle (related g4-g3) both in the solutions and their records is.

4) Reliability of astronomical solution 1. And secondly, how certain are the authors now that the preferred solution of La2010b (or c) is reliable back to ∼56 Ma, as before they have stated (in Westerhold et al., 2015) that the solution is only reliable to 48 Ma. Indeed more and better records are now available, which seem to have led to their different appreciation of the solution. However, the pattern of the ∼100-kyr eccentricity cyclicity also needs to be reliable before the ∼2-Myr cycle can be thrusted as the latter cycle can also be written as a combination of two ∼100-kyr eccentricity components (95 and 99, and 124 and 132 kyr). One reason that Lauretano et al. (2016) had a preference for the 2 cycle age model rather than the alternative 3 cycle model for C23n was the apparently good fit of the distinct four 100-kyr maxima in the d13C records with the pattern in 400-kyr cycle no. 127 now (correctly) tuned to no. 126. However, this 400-kyr cycle (i.e., no. 126) does not show the expected 4 relatively strong ∼100-kyr maxima in its maximum in addition to less distinct ∼100-kyr maxima in d13C in the 400-kyr minima above and below. To me this suggests that the pattern of the ∼100-kyr eccentricity cyclicity might already not be fully reliable around 50.8 Ma, so this raises doubts about the reliability of the solution further back in time. This uncertainty and

lack of perfect fit should be addressed. The authors should know how careful you have to be when comparing the details observed in proxy records with the solution when its reliability becomes less certain, as they also state in the ms.

5) Reliability of astronomical solution 2. The authors discuss shortly the origins of the different La2010 and La2011 solutions. This is an important issue as their from a cyclostratigraphic perspective preferred La2010b (and c) solutions have been adjusted to the short-term INPOP08 ephemeris solution, which is considered less stable and reliable than either INPOP06 (La2010a) or INPOP10 (La2011 solution), as there is a bias in INPOP08 regarding the position of Jupiter. This point should preferably be elaborated in somewhat more detail as the authors claim that they find the best fit with the La2010b (or c) solutions which are considered less reliable from an astronomical point of view. But see also points 3 and 4.

6) Paleomagnetic data and interpretation: It would be good if a paleomagnetic expert could have a look at the data and interpretation. This is critical and I am not an expert in this field.

Minor points:

The use of the word random in l.1, p.14. This is not a correct word/term to describe the outcome of non-linear complex systems such as the Solar System, as such systems do not behave in a random way.

---

## Referee Comment (RC2) · S. Meyers (Referee) · 16 Mar 2017

The manuscript "Astronomical Calibration of the Ypresian Time Scale: Implications for Seafloor Spreading Rates and the Chaotic Behaviour of the Solar System?" presents a complete eccentricity-based astronomical tuning of the Ypresian time scale (47.8-56.0 Ma), through the synthesis of new and published data from Demerara Rise (ODP Site 1258) and the Walvis Ridge (ODP Sites 1262, 1263, 1265, 1267). The data production and assimilation campaign that is the foundation of this study is an impressive effort, and well documented in the manuscript. The results are taken to (1) resolve a

controversy about the appropriate astronomical tuning through the interval (the question of how many 405 ka cycles are in magnetochron C23, and the identification of the correct theoretical model for short eccentricity tuning), (2) provide confirmation of the chaotic behavior of the Solar System that has been proposed based on theoretical models (e.g., Laskar et al., 1989), and (3) identify an increase in ocean spreading rate between 51-52.5 Ma that is interpreted to be linked to chaotic orbital behavior, through an influence on dynamic mantle flow.

These conclusions embody a series of hypotheses that have been tested with varying degrees of rigor. Central to this study is the identification of the correct theoretical eccentricity model (Laskar et al., 2004, 2011a, 2011b), and related to this, confirmation of the chaotic behavior of the Solar System. I would like to bring the attention of the authors to a recently published study by Ma et al. (2017), which provides geologic evidence confirming the chaotic behavior of the Solar System, through the identification of a chaotic resonance transition during the Coniacian (∼85-87 Ma). Of relevance to the present CPD manuscript, Ma et al. (2017) present a framework of statistical tests that provide a rigorous basis for the identification of chaotic resonance transitions. This is achieved through quantitative evaluation of the amplitude modulation of a number of carrier signals that express the secular resonance (see Ma et al. sections "Data analysis" and "Approaches for quantifying the (s4-s3)-(g4-g3) secular resonance", and the supplementary Astrochron R-script provided in that paper; Meyers, 2014), and by leveraging available independent time control (e.g., radioisotopic data and their uncertainties, including correlation uncertainties) to eliminate potential artifacts from changes in sedimentation rate (see Table 1 of Ma et al., and the section "Anchoring the floating astrochronology with radioisotopic data").

Therefore, my major recommendation for revision of the present CPD manuscript is to follow the quantitative recipe that is outlined in Ma et al. (2017): (1) run the Astrochron R-script to test for the expected amplitude modulations in the 405 ka tuned data and (2) construct an analysis similar to that in Table 1 of Ma et al. (2017), to eliminate

the possibility that changes in sedimentation rate (including hiatus) are influencing the observed modulation patterns. In terms of testing for a chaotic resonance transition, it would be ideal to apply this approach to a floating 405 ka time scale that is not directly anchored to a theoretical astronomical solution, to avoid circular reasoning; if feasible, this can be included as a supplementary analysis. In addition to verifying the presence of a chaotic resonance transition – if present – these analyses provide more rigorous statistical grounds for selecting the appropriate theoretical model for short eccentricity tuning.

An example of the power spectrum integration approach, which is central to the Ma et al. (2017) methodology, is provided in the Astrochron R-script below. Please run this script to produce a summary figure illustrating the characteristic "grand cycles" that are expressed in the amplitude (and power) modulation of the short eccentricity terms. The resultant plots provide a fingerprint of the grand cycles associated with the different theoretical astronomical solutions, for comparison with the Walvis Ridge and Demerara Rise data. For example, note the change in the character of the grand cycles in the La2010b solution at ∼50 Ma (panel b), and also, the unusual behavior of the La2004 solution at ∼52.5 Ma (panel a).

```
**————- BEGIN Astrochron script ————-**

**compare short eccentricity modulations ("grand cycles") from La2004, # La2010b, La2010d, La2011**

library(astrochron);

la04=getLaskar("la04");

la10b=getLaskar("la10b");

la10d=getLaskar("la10d");

la11=getLaskar("la11");
```

**set plotAll=T if you want to view the progress of each function graphically # (this is slower). set plotAll=F if you simply want to display the final summary plot.**

plotAll=F;

if(!plotAll) plot2=0;

if(plotAll) plot2=3;

**extract interval from 40-60 Ma and interpolate to 2.5 ka**

la04ecc=iso(linterp(cb(la04[,1],la04[,4]),dt=2.5,genplot=plotAll),xmin=40000, xmax=60000,genplot=plotAll);

la10becc=iso(linterp(cb(la10b[,1],la10b[,2]),dt=2.5,genplot=plotAll),xmin=40000, xmax=60000,genplot=plotAll);

la10decc=iso(linterp(cb(la10d[,1],la10d[,2]),dt=2.5,genplot=plotAll),xmin=40000, xmax=60000, genplot=plotAll);

la11ecc=iso(linterp(cb(la11[,1],la11[,2]),dt=2.5,genplot=plotAll),xmin=40000, xmax=60000, genplot=plotAll);

**compute evolutive power spectra, and conduct short eccentricity integrations**

pwr_la04=eha(la04ecc,win=500,fmax=.1,output=2,pl=1,pad=5000,genplot=plot2, ydir=-1,xlab="Frequency (cycles/ka)",ylab="Age (ka)");

integrate_ecc_la04=integratePower(pwr_la04,flow=0.007,fhigh=0.012,npts=201, pad=5000,ln=T, ydir=-1,genplot=plotAll);

pwr_la10b=eha(la10becc,win=500,fmax=.1,output=2,pl=1,pad=5000,genplot=plot2, ydir=-1,xlab="Frequency (cycles/ka)",ylab="Age (ka)");

integrate_ecc_la10b=integratePower(pwr_la10b,flow=0.007,fhigh=0.012,npts=201, pad=5000,ln=T, ydir=-1,genplot=plotAll);

```
pwr_la10d=eha(la10decc,win=500,fmax=.1,output=2,pl=1,pad=5000,genplot=plot2,
ydir=-1,xlab="Frequency (cycles/ka)",ylab="Age (ka)");

integrate_ecc_la10d=integratePower(pwr_la10d,flow=0.007,fhigh=0.012,npts=201,
pad=5000,ln=T, ydir=-1,genplot=plotAll);

pwr_la11=eha(la11ecc,win=500,fmax=.1,output=2,pl=1,pad=5000,genplot=plot2,
ydir=-1,xlab="Frequency (cycles/ka)",ylab="Age (ka)");

integrate_ecc_la11=integratePower(pwr_la11,flow=0.007,fhigh=0.012,npts=201,
pad=5000,ln=T, ydir=-1,genplot=plotAll);

pl(r=1,c=4);

plot(integrate_ecc_la04[,2],integrate_ecc_la04[,1],type="l",ylim=c(60000,40000),
ylab="Time (ka)",xlab="La2004 Short Ecc. Power");

mtext("(a)",line=1);

plot(integrate_ecc_la10b[,2],integrate_ecc_la10b[,1],type="l",ylim=c(60000,40000),
ylab="Time (ka)",xlab="La2010b Short Ecc. Power");

mtext("(b)",line=1);

plot(integrate_ecc_la10d[,2],integrate_ecc_la10d[,1],type="l",ylim=c(60000,40000),
ylab="Time (ka)",xlab="La2010d Short Ecc. Power");

mtext("(c)",line=1);

plot(integrate_ecc_la11[,2],integrate_ecc_la11[,1],type="l",ylim=c(60000,40000),
ylab="Time (ka)",xlab="La2011 Short Ecc. Power");

mtext("(d)",line=1);

**————- END Astrochron script ————-**
```

The proposed link between chaotic orbital behavior and changes in ocean spreading

rate (conclusion 3 noted above) is the most speculative. If it is to be included in the manuscript in a meaningful manner, I believe it is necessary to provide a more complete description of the physical mechanism by which it is manifested, either qualitatively (how does orbital behavior impact mantle flow, and how would a chaotic transition thus be expressed as an increase in spreading rates?), or even better quantitatively through modeling. Of course, correlation is not proof of causation, but if the orbital behaviors can be reasonably demonstrated to have the appropriate order-of-magnitude effect on mantle flow and plate reorganization, this would be an important discovery.

In conclusion, I would like to reiterate that the data production and assimilation campaign that is the foundation of this study is an impressive effort, which is no doubt a tribute to the expertise of this research group, and the decades of careful work that they have conducted on the topic of Eocene astrochronology. Further, I believe that these new records will yield considerable insight into astronomical forcing during the Ypresian, a time of great interest due to the numerous hyperthermal events that are present and the overall warm climate state. It is my hope that the application of the statistical methodologies outlined in this review help to clarify and strengthen the hypothesis testing, and thus reduce the ambiguity associated with multiple plausible interpretations of the data.

Outlined below are a number of additional comments that are referenced to the pages and line numbers in the CPD manuscript.

Page 11, lines 4-5: Here is it noted that "Because of higher sedimentation rates than observed at Leg 208 sites, cyclicity in the Site 1258 XRF Fe data is mainly precession related with less pronounced modulation by eccentricity." This statement requires further explanation; as written it would suggest that sedimentation rate changes may impose amplitude modulation upon precession (and short eccentricity?) as an artifact, which could complicate the assessment of the long term "grand cycles".

Page 11, line 19; Figure S9 caption; Figure S11 caption: Please specify the details of

the detrending approach utilized, so that it can be replicated in future work. Note that Astrochron includes several functions for detrending that may be of utility here (e.g., functions 'noLow' and 'noKernel').

Page 18, line 13: Please note the study by Laurin et al. (2016), which provides additional independent confirmation of the eccentricity pacing of these hyperthermals.

Figure S3 (item 1). It is excellent to see that this study evaluates the reproducibility of the XRF Fe data, which is standard practice when presenting most geochemical results, but often ignored in XRF scanning studies.

The results from Site 1267 look excellent, but an r-squared value of 0.09 associated with the Site 1262 XRF Fe data sets indicates a surprisingly poor correlation. Please estimate the significance of the correlations (specify the p-value), and note that this should be done in a manner that accounts for autocorrelation in the data series, such as the phase randomized surrogate approach of Ebisuzaki (1997). The Ebisuzaki method has been adapted and included in Astrochron as the function 'surrogateCor' (Baddouh et al, 2016). Given that only 9% of the variance is shared between the two Site 1262 XRF Fe data sets, the implications of this for the conclusions of the study need further discussion (approximately 38% of the data comes from the first scanning study, and 62% from the second).

More generally, I would like to encourage the adoption of an XRF data reporting approach that quantifies instrument stability (e.g., see Figure A.1 of Ma et al, 2014), and reproducibility based on duplicate analyses (e.g., see Figure A.2 and Table A.1 of Ma et al., 2014).

Figure S3 (item 2): It is necessary to include a similar analysis and plot for the iron data from Site 1263.

Figure S12: While the proposed match between the theoretical astronomical solution and the benthic carbon isotope data seems plausible throughout much of the record,

the interval from 51-52 Ma shows a response that is opposite to what theory predicts. This requires some further comment in the manuscript.

References:

Baddouh, M., Meyers, S.R., Carroll, A.R., Beard, B.L., and Johnson, C.M., L. (2016). Lacustrine 87Sr/86Sr as a tracer to reconstruct Milankovitch forcing of the Eocene hydrologic cycle. Earth and Planetary Science Letters 448, p. 62-68.

Ebisuzaki, W. (1997). A Method to Estimate the Statistical Significance of a Correlation When the Data Are Serially Correlated. Journal of Climate 10, p. 2147-2153.

Laskar, J. (1989). A numerical experiment on the chaotic behavior of the Solar System. Nature 338, p. 237–238.

Laskar, J., Robutel, P., Joutel, F., Gastineau, M., Correia, A.C.M., and Levrard, B., (2004). A long-term numerical solution for the insolation quantities of the Earth. Astron. Astrophys. 428, p. 261–285.

Laskar, J., Fienga, A., Gastineau, M. & Manche, H. (2011a). La2010: a new orbital solution for the long-term motion of the Earth, Astron. Astrophys. 532, 89.

Laskar, J., Gastineau, M., Delisle, J.-B., Farres, A. & Fienga, A. (2011b). Strong chaos induced by close encounters with Ceres and Vesta. Astron. Astrophys. 532, 4.

Laurin, J., Meyers, S.R., Galeotti, S., and Lanci, L. (2016). Frequency modulation reveals the phasing of orbital eccentricity during Cretaceous Oceanic Anoxic Event II and the Eocene hyperthermals. Earth and Planetary Science Letters 442, p. 143-156.

Ma, C., Meyers, S.R., Sageman, B.B. (2017). Theory of chaotic orbital variations confirmed by Cretaceous geological evidence. Nature 542, p. 468-470.

Ma, Chao, Meyers, S.R., Sageman, B.B., and Singer, B.S., Jicha, B.R. (2014). Testing the Astronomical Time Scale for Oceanic Anoxic Event 2, and its Extension into Cenomanian Strata of the Western Interior Basin (U.S.A.). Geological Society of America

Bulletin 126, p. 974-989.

Meyers,S.R. (2014). Astrochron: An R package for astrochronology. http:// cran.r-project.org/package=astrochron.

R Core Team (2016). R: A Language and Environment for Statistical Computing, R Foundation for Statistical Computing http://www.R-project.org/.

---

## Author Comment (AC1) · 24 Mar 2017

Dear Frits Hilgen,

Thank you very much for taking the time to provide a very constructive review.

Here we reply directly to your comments.

*1) Statistical identification of the very long period ~2-Myr eccentricity minima. The eccentricity nodes associated with the very long, 2-Myr, eccentricity cycle are now visually determined in their proxy records, but preferably they should also be pinpointed by means of statistical analysis, such as complex amplitude modulation or the method outlined in Meyers (2015). Such an independent statistical confirmation of the position of these nodes is critically important next to their visual determination to convince the reader of the correctness of the conclusions drawn in the ms. Otherwise the authors have to clearly state why they did not carry out the necessary and logical statistical analysis. Especially the statistical method introduced by Meyers (2015) seems very helpful to reconstruct eccentricity and capture the nodes associated with the very long term eccentricity cycle, so the question is why such statistical methods have not been applied. I guess that the authors may well have given it a try, so in that case, why was it not included in the ms (even to show that these statistical approaches do not work well in this particular case)? The authors may thus wish to discuss in some detail the (dis)advantages of a visual versus a statistical approach. This topic is discussed in some detail by Hinnov (2013) and has been presented in more detail by Steve Meyers in some of his presentations. The point here is that visual recognition of cycle patterns albeit being subjective can be considered an expert system, being able to identify distortions of the signal that are very commonly encountered in cyclostratigraphic records (see e.g. point 2) and that may cause problems when applying statistical techniques (see also Hilgen et al., 2015)*

This is a very important point. Extraction of the amplitude modulation (AM) using statistical methods like those implemented in the *Astrochron* package (Meyers 2014) or the ENVELOPE (Schulz et al. 1999) routine are important for independently testing the visual recognition of cycle patterns. AM analysis on XRF core data using the ENVELOPE routine was applied at ODP Site 1262 (52-60 Ma) and ODP Site 1258 (47-54 Ma) records in Westerhold et al. (2012) in order to search for the very long eccentricity minima. Meyers (2015) used a* values (red over green ratio from [shipboard] color core scanning) from ODP Site 1262 between PETM and ETM2 (Elmo) to test the existing astrochronologies (Lourens et al. 2005, Westerhold et al. 2007). Both methods (Astrochron, ENVELOPE) already provide sound statistical testing of chronologies at ODP Site 1258 and Leg 208 sites.

In our manuscript we have not explicitly included results of these statistical analysis. But based on the reviewer´s recommendation we will now provide the statistical analysis on XRF Fe intensity data from ODP 1258, 1262, and 1263 in the revised ms: these data and analyses have already proven their great potential in testing astronomical solutions (Lourens et al. 2005, Westerhold et al. 2007, Westerhold et al. 2012, Meyers 2015). Following the approach of Zeeden et al. (2015) we filtered out the short eccentricity cycle (100-kyr) and applied a broad bandpass filter (0.004 to 0.016 cycles/kyr; 250-62.5 kyr per cycle; Tukey window) and subsequently made a Hilbert transform to extract the AM using the *Astrochron* software package (Meyers 2015) for Site 1258 and 1263 data. As a basic age model we used the 405-kyr age model as given in Table 46 of the submitted dataset. The resulting 405-kyr

AM of the XRF Fe intensity data are plotted against the La2004, La2010, and La2011 orbital solutions (Fig. 1 of this reply). The AM of the orbital solutions were extracted as described in Westerhold et al. 2012. For ODP Site 1262 we plotted the 405-kyr AM of XRF Fe intensity data using the Option2 age model of Westerhold et al. 2012 (the Option2 W12 405-kyr age model is at Site 1262 almost identical with the updated 405-kyr age model in the submitted manuscript for the 53 to 58 Ma interval). We followed this procedure to demonstrate that similar results can be obtained with different approaches (Astrochron vs ENVELOPE).

The position of the very long eccentricity minima in the AM of XRF Fe intensity data in the interval from 46 to 59 Ma (blue bars in Fig. 1 below) best fits with minima in the La2010b and La2010c orbital solutions. In contrast, the minima do not match minima in the La2004 solution suggesting that this solution is not appropriate for testing geological data. The La2010a-d and La2011 solutions fit to geological data back to 50 Ma. Beyond 50 Ma these solutions diverge (as discussed in detail in Westerhold et al. 2012). Only the La2010b and La2010c solutions exhibit very long eccentricity minima at ~53.3 and ~54.5 Ma and thus were chosen for detailed astronomical tuning of the records presented in our manuscript. The minimum at ~54.5 Ma is a very prominent feature in the data of the Leg 208 sites that has been intensively discussed (Lourens et al. 2005, Westerhold et al. 2007, Meyers 2015). The minimum at ~53.3 Ma is also detectable using the statistical methods but can be much better seen by visual inspection of the data (see manuscript Fig. 2b, 3, and 4).

As pointed out by the reviewer the latter leads to the discussion of the (dis)advantages of a visual versus a statistical approach. With respect to our study for the revised manuscript we will have both a statistical and visual approach which lead to the same results.

We do not want to repeat what already was discussed by Hinnov (2013) and Hilgen et al. (2015). But one important aspect complicating the analysis of the data is the strong influence of the hyperthermal layers on sedimentary features and data. We will further comment on this aspect in the reply to issue 2 below.

Clean AM can be extracted from eccentricity modulated precession dominated records. This is the case for the Leg 208 sections from 56 to 54 Ma, as already shown in Lourens et al. (2005), Westerhold et al. (2007) and Meyers (2015). Due to a drop in sedimentation rates around the ETM2 event at Leg 208 sites (54.050 Ma), precession cycles are not the dominant frequency over the 52 to 47 Ma interval of Leg 208 sites. Instead, the lower sedimentation rates resulted in the recording of more pronounced eccentricity cycles. Isolating the AM of the 100 and 405 eccentricity cycles in the sediments should be easy, and as mentioned by Laskar (1999): *"… in the climatic precession, the two terms p+g4 and p+g3 should induce also a modulation of frequency g4 – g3 … (period ca. 2.475 Ma) in the 19 ka term of the climatic precession, as well as in the 95 and 125 ka terms in the eccentricity. For these two last terms, it should be noted that even if the resolution of the data does not make it possible to discriminate between the 95 and 125 ka terms, the modulation of the amplitude of these terms is the same, and thus could still be discernible in the geological record"*. Indeed, Site 1263, which is characterized by the highest sedimentation rate of the Leg 208 sites, still preserves the best AM record of the sites in the transect (Fig. 1 below).

We agree that visual recognition of cycle patterns in data benefits from many years of experience. Aberrations in the data by additional effects (e.g. dissolution during hyperthermal events) will surely cause issues when applying statistical techniques as also already discussed in Hilgen et al. (2015).

For the revised manuscript we will add a section on the issues mentioned above and add a new figure (presented as Fig. 1 below).

[Figure]

**Figure 1** – Comparison of the amplitude modulation (AM) of the short eccentricity cycle between the La2004, La2010, and La2011 orbital solutions and Fe intensity data from ODP Sites 1258 (red), 1262 (orange) and 1263 (blue). For the orbital solutions we also plotted the 405-kyr AM. The short eccentricity AM of Sites 1258, 1262 and 1263 Fe intensity data are plotted on the 405-kyr scale model (Table 46 of the submitted manuscript). The very long eccentricity minima are highlighted by light blue bars in the orbital solutions and the Fe intensity data. Statistical and visual recognition of cycle pattern suggest that the La2010b and La2010c solutions are most consistent with the geological data.

*2) Potential distortion by non-linear response of the climate system. The authors have to explain that the amplitude changes they see in their proxies are related to the amplitude of the ~100-kyr eccentricity cycle and not caused by a non-linear response of the climate system to the eccentricity forcing through associated changes in the precession amplitude. This issue might become critical when dealing with the proxy expression of early Eocene hyperthermals. Evidently, the "distortion" caused by such a non-linear response will also have consequences for the outcome of the statistics as I guess that these usually start from linear relationships. This issue has to be addressed in the discussion.*

It is not fully clear what the referee exactly refers to in the first part of his comment. Because the precession amplitude is modulated by eccentricity, it would be nearly impossible in most records to rule out *"a non-linear response of the climate system to the eccentricity forcing through associated changes in the precession amplitude"*.

Non-linear response of climate is critical in the Ypresian, as also pointed out by the referee in the second part of his comment. Multiple carbon cycle perturbations are documented in the stable isotope records (CIEs). These hyperthermal events are likely caused by massive releases of carbon to the ocean-atmosphere system including the dissolution of carbonates at the seafloor. Because the extend of the CIE's are scaled to the amount of carbon injected (Pagani et al., 2006) and the residence time of carbon is in the order of 100 kyr (Broecker and Peng, 1982), the events will influence the amplitude of the bulk and benthic stable carbon isotope data and thus any AM analysis of early Eocene records. On top of this, the added carbon leads also to dissolution of carbonates at the seafloor increasing the relative amount of non-carbonate material in the sediment (as detected by higher XRF Fe values). This will influence the statistical and visual recognition of cyclicity. Modeling suggests that hyperthermals, except for the PETM, seem to be paced by eccentricity forcing of the carbon cycle with the amplitude of the events being partly driven by the eccentricity amplitude itself (Kirtland-Turner et al. 2014). The carbon isotope data (see Figure 4 of the submitted manuscript) do show a good correspondence to the short eccentricity AM. In particular, the very long eccentricity minima are expressed as an interval of very low AM in the benthic carbon isotope data. Almost all hypothermal events occur outside the very long eccentricity minima. Only slight excursions at C21R5, C22r5, and C23n.2nH1 coincide with these nodes, but with comparatively reduced CIE than hyperthermals suggesting these are not hyperthermals. Hyperthermal layers are very well documented in the XRF data by prominent peaks due to dissolution of carbonate. Larger CIEs show higher XRF Fe peaks. This tends to exaggerate the AM in the statistical analysis (see Fig. 1). Hyperthermal events could be interpreted as amplifiers of the eccentricity amplitude with a bias toward higher amplitudes. As our focus lies on the very long eccentricity minima, this ensures that the distortion by hyperthermal events is not significantly altering the results of our study.

In the revised manuscript we will add a chapter carefully addressing this issue.

*3) Exclusion of expression of 1.0-Myr eccentricity cycle. The authors claim that they have detected the expression of the transition from libration to circulation of the very long period eccentricity cycle in the geological record. However, to be sure, they have to address the following two points. In the first place, what is the role of the relatively strong ~1.0 Myr eccentricity component (related to g5-g1, and can also be written as a combination of ~100-*

*kyr components) , especially in determining the node around 53 Ma that they attribute to the ~2-Myr cycle. They should thus make clear what the exact expression of the ~2-Myr cycle (related g4-g3) both in the solutions and their records is.*

As stated by Laskar (1999) one needs to extract the AM of both obliquity and precession in a geological dataset in order to detect the libration to circulation. This is almost impossible as mentioned by Laskar et al (2011), because this would require a record that is both influenced (or driven) by high latitude and low latitude processes. Obliquity AM could be extracted from benthic d18O records if the deep sea temperature is continuously affected by obliquity. This is not the case for the Paleocene and early Eocene (see for example Littler et al., 2014), thus investigation of the AM of obliquity is difficult with the currently available records. Laskar et al. 2011 recommended: "It may be more direct to search only for a modulation of the g4 – g3 (or s4 – s3 ) period, as it appears in Fig. 11 (of Laskar et al. 2011)."; g4 – g3 is the ~2.4 myr eccentricity modulation.

It is not clear what the reviewer is referring to with the g5-g1 argument. This roughly 1-myr cycle was never mentioned by Laskar. Clarification would be very welcome.

The transition from libration to circulation should be visible according to Laskar (1999), Laskar et al. (2004), and Laskar et al. (2011) (see also Pälike et al. 2004) by a switch from a ~2.4 myr period to a ~1.2 myr period in the modulation of eccentricity and climatic precession. And very importantly, it can switch back to ~2.4 myr shortly after (see Laskar 1999, Figure 9 therein and in the response to reviewer Stephen Meyers). Comparing orbital solutions with geological data indicates such a switch occurred between 52 and 55 Ma. The AM minima in the data from 47 to 52 Ma are 2 to 2.4 myr spaced (Fig. 1 above), from 52 to 55 Ma they are roughly 1.2 myr spaced, and after 55 Ma the spacing is ~2.4 myr.

The transition we have found is the first of the Cenozoic. Earlier transitions are likely as proposed by Laskar (1999). Indeed, one transition has recently been identified in the Cretaceous, ~86 Ma ago in a paper that was just published after our initial submission (Ma et al. 2017). We will take these findings into account in the revised version (please also see reply to reviewer Stephen Meyers). Both new observations of transitions from the 1:2 to the 1:1 resonance "*could make it possible to obtain very precise information on the initial conditions and (or) parameters of the model. One could even dream that if the succession of the transitions from the 1:2 to the 1:1 resonance were found and dated over an interval of 200 Ma that this could be the ultimate test for the gravitational model. It would make it possible, for example, to obtain the J2 value of the Sun with high accuracy, or to test the model of general relativity.*" (Laskar 1999)

In the revised manuscript we will elaborate more on the fingerprint of the transition from libration to circulation.

*4) Reliability of astronomical solution 1. And secondly, how certain are the authors now that the preferred solution of La2010b (or c) is reliable back to ~56 Ma, as before they have stated (in Westerhold et al., 2015) that the solution is only reliable to 48 Ma. Indeed more and better records are now available, which seem to have led to their different appreciation of the solution. However, the pattern of the ~100-kyr eccentricity cyclicity also needs to be reliable before the ~2-Myr cycle can be thrusted as the latter cycle can also be written as a*

*combination of two ~100-kyr eccentricity components (95 and 99, and 124 and 132 kyr). One reason that Lauretano et al. (2016) had a preference for the 2 cycle age model rather than the alternative 3 cycle model for C23n was the apparently good fit of the distinct four 100-kyr maxima in the d13C records with the pattern in 400-kyr cycle no. 127 now (correctly) tuned to no. 126. However, this 400-kyr cycle (i.e., no. 126) does not show the expected 4 relatively strong ~100-kyr maxima in its maximum in addition to less distinct ~100-kyr maxima in d13C in the 400-kyr minima above and below. To me this suggests that the pattern of the ~100-kyr eccentricity cyclicity might already not be fully reliable around 50.8 Ma, so this raises doubts about the reliability of the solution further back in time. This uncertainty and lack of perfect fit should be addressed. The authors should know how careful you have to be when comparing the details observed in proxy records with the solution when its reliability becomes less certain, as they also state in the ms.*

This is a valid concern.  None of the available orbital solutions perfectly fit the geological data. However, it is important that we isolated the transition in the data, which is also present in the La2010b and La2010c solutions. The short eccentricity cycle pattern both in the solutions and the geological data will not match perfectly beyond 50 Ma where the uncertainty in the solutions increase (as discussed in Westerhold et al. 2012). Surprisingly the geological data and La2010b/c solutions are very similar from 53.5 Ma to the PETM. The Westerhold et al. (2012) paper did not include stable isotope data and new XRF core scanning data from Leg 208, and relied on Site 1258 XRF data in the interval from 47 to 53 Ma only.

The 51 to 52 Ma interval mentioned by the reviewer is the most difficult part to tune in the Ypresian, in large part because of the  multiple hyperthermal events and the shift in carbon isotope data (discussed in detail in the manuscript). We agree that the eccentricity solutions from La2010b/c might not be completely reliable in this interval. We will detail this and the potential limits in the revised manuscript.

Despite the uncertainties we decided to provide a tuned age model to La2010b/c because the match in the 53.5 Ma to the PETM interval is good enough to do so. If in doubt, readers can still make use the 405-kyr age model, also given in the submitted manuscript.

*5) Reliability of astronomical solution 2. The authors discuss shortly the origins of the different La2010 and La2011 solutions. This is an important issue as their from a cyclostratigraphic perspective preferred La2010b (and c) solutions have been adjusted to the short-term INPOP08 ephemeris solution, which is considered less stable and reliable than either INPOP06 (La2010a) or INPOP10 (La2011 solution), as there is a bias in INPOP08 regarding the position of Jupiter. This point should preferably be elaborated in somewhat more detail as the authors claim that they find the best fit with the La2010b or c) solutions which are considered less reliable from an astronomical point of view. But see also points 3 and 4.*

We present a data-compliant manuscript including a revised 405-kyr stable eccentricity cyclostratigraphy for the Ypresian. Comparing the geological data with astronomical solutions exhibits the consistent transitions from the 1:2 to the 1:1 resonance present in the La2010b and La2010c solution. We are fully aware that the La2010b and La2010c solution from an astronomer's point of view are considered less reliable because they used a less

stable ephemeris (Fienga et al. 2011, Laskar et al. 2011). It should be the task of the astronomers to explore the consequences of our findings. It is by far beyond the scope of our manuscript to evaluate the ephemerides in detail, an exercise which by the way can only be undertaken by experts in the field (e.g. Agnes Fienga, Jacques Laskar, etc.).

We were surprised by our results because previous publications suggested that the La2010d and La2011 solutions would fit better to the then available geological data (see Westerhold et al. 2012, Lauretano et al. 2016). But our new results offer an alternative, if not controversial, perspective. Please be aware that the La2010 solutions are all very similar up to 54 Ma (see Westerhold et al. 2012 Figure 2b), but only the La2010b/c solutions show the AM minimum at 53.2 Ma, as in the geological data from Leg 208 (see manuscript Fig. 4 and Fig. 2 of this reply). The La2010d and La2011 solutions, deemed most reliable in the Westerhold et al. (2012) paper, also exhibit no clear minimum around 54.5 Ma but the geological data suggest low amplitude variability from 54.2 to 55.0 Ma in this interval (as in La2010b/c).

In the revised version we will add a comment on the orbital solutions to make this more clear.

[Figure]

**Figure 2** – Amplitude modulation (AM) in the La2010b,c,d and La2011 orbital solutions for eccentricity from 47 to 57 Ma. Note that La2010b/c are so similar that the individual eccentricity curves are difficult to separate.

*Minor points:*
*The use of the word random in l.1, p.14. This is not a correct word/term to describe the outcome of non-linear complex systems such as the Solar System, as such systems do not behave in a random way.*

The full sentence referred to is: "Older than 50 Ma the location of the very long eccentricity nodes in available orbital solutions (La2004 – Laskar et al., 2004; La2010 – Laskar et al., 2011a; La2011 – Laskar et al., 2011b) are considered to appear randomly (Westerhold et al., 2015)". Of course the Solar System is not behaving randomly. The sentence says that the very long eccentricity nodes we want to look for in the geological data appear randomly in different numerical solutions. This means their position in any of the solutions cannot be used for direct anchoring of astrochronologies for older than 50 Ma until the solution is identified which best fits to geological data.

We will now rephrase the sentence as follows: "Older than 50 Ma the location of the very long eccentricity nodes in available orbital solutions (La2004 – Laskar et al., 2004; La2010 – Laskar et al., 2011a; La2011 – Laskar et al., 2011b) are much more uncertain."

**Referenzes for the reply:**

Broecker, W. S., and Peng, T. H.: Tracers in the Sea, edited by: University, C., Lamont Doherty Geol. Obs. Publications, New York, 689 pp., 1982.

Fienga, A., Laskar, J., Kuchynka, P., Manche, H., Desvignes, G., Gastineau, M., Cognard, I., and Theureau, G.: The INPOP10a planetary ephemeris and its applications in fundamental physics Celestial Mechanics and Dynamical Astronomy, 111, 363-385, 10.1007/s10569-011-9377-8, 2011.

Hilgen, F. J., Hinnov, L. A., Abdul Aziz, H., Abels, H. A., Batenburg, S., Bosmans, J. H. C., de Boer, B., Hüsing, S. K., Kuiper, K. F., Lourens, L. J., Rivera, T., Tuenter, E., Van de Wal, R. S. W., Wotzlaw, J.-F., and Zeeden, C.: Stratigraphic continuity and fragmentary sedimentation: the success of cyclostratigraphy as part of integrated stratigraphy, Geological Society, London, Special Publications, 404, 10.1144/sp404.12, 2014.

Hinnov, L. A.: Cyclostratigraphy and its revolutionizing applications in the earth and planetary sciences, Geological Society of America Bulletin, 125, 1703-1734, 10.1130/b30934.1, 2013.

Kirtland Turner, S., Sexton, P. F., Charles, C. D., and Norris, R. D.: Persistence of carbon release events through the peak of early Eocene global warmth, Nature Geosci, 7, 10.1038/ngeo2240, 2014.

Laskar, J.: The limits of Earth orbital calculations for geological time-scale use, in: Phil. Trans. R. Soc. Lond. A, edited by: Shackleton, N. J., McCave, I. N., and Weedon, G. P., 1735-1759, 1999.

Laskar, J., Robutel, P., Joutel, F., Gastineau, M., Correia, A., and Levrard, B.: A long-term numerical solution for the insolation quantities of the Earth, Astronomy and Astrophysics, 428, 261-285, 10.1051/0004-6361:20041335, 2004.

Laskar, J., Fienga, A., Gastineau, M., and Manche, H.: La2010: a new orbital solution for the long-term motion of the Earth, Astronomy and Astrophysics, 532, 15, 10.1051/0004-6361/201116836, 2011a.

Laskar, J., Gastineau, M., Delisle, J. B., Farrés, A., and Fienga, A.: Strong chaos induced by close encounters with Ceres and Vesta, Astronomy and Astrophysics, 532, 4, 10.1051/0004-6361/201117504, 2011b.

Lourens, L. J., Sluijs, A., Kroon, D., Zachos, J. C., Thomas, E., Röhl, U., Bowles, J., and Raffi, I.: Astronomical pacing of late Palaeocene to early Eocene global warming events, Nature, 435, 1083-1087, 10.1038/nature03814, 2005.

Meyers, S.R., 2014 - Astrochron: An R Package for Astrochronology. http://cran.r-project.org/package=astrochron

Pagani, M., Caldeira, K., Archer, D., and Zachos, J. C.: An Ancient Carbon Mystery, Science, 314, 1556-1557, 10.1126/science.1136110, 2006.

Schulz, M., W. H. Berger, M. Sarnthein, and P. M. Grootes ,1999: Amplitude variations of 1470-year climate oscillations during the last 100,000 years linked to fluctuations of continental ice mass, Geophys. Res. Lett., 26(22), 3385–3388, doi:10.1029/1999GL006069

Westerhold, T., Röhl, U., Laskar, J., Bowles, J., Raffi, I., Lourens, L. J., and Zachos, J. C.: On the duration of magnetochrons C24r and C25n and the timing of early Eocene global warming events: Implications from the Ocean Drilling Program Leg 208 Walvis Ridge depth transect, Paleoceanography, 22, 10.1029/2006PA001322, 2007.

Westerhold, T., Röhl, U., and Laskar, J.: Time scale controversy: Accurate orbital calibration of the early Paleogene, Geochem. Geophys. Geosyst., 13, Q06015, 10.1029/2012gc004096, 2012.

Zeeden, C., Meyers, S. R., Lourens, L. J., and Hilgen, F. J.: Testing astronomically tuned age models, Paleoceanography, 30, 369-383, 10.1002/2014PA002762, 2015.

---

## Author Comment (AC2) · 24 Mar 2017

Dear Stephen Meyers,

Thank you very much for your detailed review.

Below we will reply to the comments.

*… I would like to bring the attention of the authors to a recently published study by Ma et al. (2017), which provides geologic evidence confirming the chaotic behavior of the Solar System, through the identification of a chaotic resonance transition during the Coniacian (~85-87 Ma) …*

    The Ma et al. (2017) manuscript was not published at the time when this manuscript was submitted but will be considered in the revised version.

*My major recommendation for revision of the present CPD manuscript is to follow the quantitative recipe that is outlined in Ma et al. (2017): (1) run the Astrochron R-script to test for the expected amplitude modulations in the 405 ka tuned data and (2) construct an analysis similar to that in Table 1 of Ma et al. (2017), to eliminate the possibility that changes in sedimentation rate (including hiatus) are influencing the observed modulation patterns.*

    Regarding 1) - We have compiled a new figure (Fig. 1 below, also see reply to Hilgen review) with the statistical analysis on the XRF Fe intensity data from ODP 1258, 1262, and 1263. We follow the approach of Zeeden et al. (2015) which is similar to the Ma et al. 2017 recipe: filtering out the short eccentricity cycle (100-kyr) using a broad bandpass filter (0.004 to 0.016 cycles/kyr; 250-62.5 kyr per cycle; Tukey window) and subsequently making a Hilbert transform to extract the AM using the *Astrochron* software package (Meyers 2015) for Site 1258 and 1263 data. As a basic age model we used the 405-kyr age model as given in table 46 of the submitted dataset. The resulting 405-kyr AM of the XRF Fe intensity data are the plotted against the La2004, La2010, and La2011 orbital solutions (Fig. 1 above, also see reply to Hilgen review).

    Regarding 2) – Table 1 of Ma et al. (2017) is about radioisotopic anchors used for the Libsack astrochronology. Our record does not include ash layers for which we could do a similar analysis. Therefore, it is not clear to us why the reviewer is suggesting this approach. A similar table for calcareous nannofossil events or magnetostratigraphy would be rather complex and would in no way be helpful in identifying hiatuses or jumps in sedimentation rate also as these zones do not resolve time in suitable resolution if compared to cyclostratigraphy. Please be aware that we are using multiple records too from two different regions to ensure we are dealing with a complete record. We made clear in the manuscript that a single site or region record potentially could include gaps and/or condensed sections that could only be detected if compared to another record. Deriving errors from calcareous nannofossil datums at different site is difficult because they are not perfectly synchronous between Site 1258 and Leg 208 sites. Even between Leg 208 sites some events are not accurately synchronous probably due to sampling and / or depth related (dissolution etc.) issues. It is also in the nature of magnetostratigraphy with each site exhibiting slightly different results that error analysis is not straight. We could for example

imagine that a second Libsack core (Ma et al. 2017) would also yield slightly different ages for ash layers. Thus we refrain from compiling a table like in Ma et al. 2017 as it will not provide better constraints on the data as already in the extensive dataset from multiple records presented in the manuscript.

[Figure]

**Figure 1** – Comparison of the amplitude modulation (AM) of the short eccentricity cycle between the La2004, La2010, and La2011 orbital solutions and Fe intensity data from ODP Sites 1258 (red), 1262 (orange) and 1263 (blue). For the orbital solutions we also plotted the 405-kyr AM. The short eccentricity AM of Sites 1258, 1262 and 1263 Fe intensity data are plotted on the 405-kyr scale model (Table 46 of the submitted manuscript). The very long eccentricity minima are highlighted by light blue bars in the orbital solutions and the Fe intensity data. Statistical and visual recognition of cycle pattern suggest that the La2010b and La2010c solutions are most consistent with the geological data.

*In terms of testing for a chaotic resonance transition, it would be ideal to apply this approach to a floating 405 ka time scale that is not directly anchored to a theoretical astronomical solution, to avoid circular reasoning; if feasible, this can be included as a supplementary analysis. In addition to verifying the presence of a chaotic resonance transition – if present – these analyses provide more rigorous statistical grounds for selecting the appropriate theoretical model for short eccentricity tuning.*

We have done so, as described in the reply to Hilgen review, and will include this in the revised manuscript.

*An example of the power spectrum integration approach, which is central to the Ma et al. (2017) methodology, is provided in the Astrochron R-script below. Please run this script to produce a summary figure illustrating the characteristic "grand cycles" that are expressed in the amplitude (and power) modulation of the short eccentricity terms. The resultant plots provide a fingerprint of the grand cycles associated with the different theoretical astronomical solutions, for comparison with the Walvis Ridge and Demerara Rise data. For example, note the change in the character of the grand cycles in the La2010b solution at ~50 Ma (panel b), and also, the unusual behavior of the La2004 solution at ~52.5 Ma (panel a).*

We applied the script and will add the following figure (Fig. 2 in this reply) to the revised manuscript supplement.

[Figure]

**Figure 2** - Short eccentricity band power for La2004, La2010b, La2010d and La2011 extracted using the *Astrochron* software (Meyers 2014) according to Ma et al. (2017) from 40 to 60 Ma. The results are similar to those plotted in Figure 1 above (from Westerhold et al. 2012). The La2010b solution clearly shows the transition from libration to circulation and back between 52 and 55 Ma. In contrast La2010d and La2011 solutions do not show the transition. La2004 solution also does not show the transition, but an unusual behavior from 53 to 54 Ma. Please note that the AM of La2010 and La2011 solutions are very similar to 50 Ma but diverge thereafter. The La2004 solution AM is similar to the La2010 and La2011 up to 45 Ma, in times older that than 45 Ma the AM significantly diverge (as discussed in Westerhold et al. 2012).

*The proposed link between chaotic orbital behavior and changes in ocean spreading rate (conclusion 3 noted above) is the most speculative. If it is to be included in the manuscript in a meaningful manner, I believe it is necessary to provide a more complete description of the physical mechanism by which it is manifested, either qualitatively (how does orbital behavior impact mantle flow, and how would a chaotic transition thus be expressed as an increase in spreading rates?), or even better quantitatively through modeling. Of course, correlation is not proof of causation, but if the orbital behaviors can be reasonably demonstrated to have the appropriate order-of-magnitude effect on mantle flow and plate reorganization, this would be an important discovery.*

Our manuscript is data rich. And includes surprisingly new results which cannot all be extensively presented in a single manuscript. The reviewer asks: *how does orbital behavior impact mantle flow, and how would a chaotic transition thus be expressed as an increase in spreading rates?, or even better quantitatively through modeling.*

We argue here that these are perfect questions for future research projects and modeling studies that should address and test our new findings. We definitely agree that correlation is not a proof of causation.

The Ma et al. 2017 paper argues that Ocean Anoxic Event 3 (OAE 3) might be mechanistically related to the transition. They speculate that "*Such a resonance transition would permit positive reinforcement of eccentricity- and obliquity-modulated seasonality, allowing for a more pronounced impact of astronomical forcing on palaeoceanography.*" We think this is also very speculative and would require modeling to be rigorously tested.

Setting up a model for the complex and chaotic mantle flow is highly sophisticated and should be done by exerts in that field, not us. We hope to give some inspiration to the modeling community to test our hypothesis. This will require a new kind of collaboration between dynamic mantle flow modeling and astronomy, something to our knowledge not undertaken before.

*In conclusion, I would like to reiterate that the data production and assimilation campaign that is the foundation of this study is an impressive effort, which is no doubt a tribute to the expertise of this research group, and the decades of careful work that they have conducted on the topic of Eocene astrochronology. Further, I believe that these new records will yield considerable insight into astronomical forcing during the Ypresian, a time of great interest due to the numerous hyperthermal events that are present and the overall warm climate state. It is my hope that the application of the statistical methodologies outlined in this review help to clarify and strengthen the hypothesis testing, and thus reduce the ambiguity associated with multiple plausible interpretations of the data.*

We agree that the focus of the manuscript is the very complex data synthesis. In the revised version we will include more rigorous statistical testing as outline above and in the reply to reviewer Frits Hilgen. We hope that our compilation of published and new data will be basis for insightful research in the Ypresian to understand climate dynamics in a warm world with elevated $pCO_2$.

***Additional comments***

*Page 11, lines 4-5: Here is it noted that "Because of higher sedimentation rates than observed at Leg 208 sites, cyclicity in the Site 1258 XRF Fe data is mainly precession related with less pronounced modulation by eccentricity." This statement requires further explanation; as written it would suggest that sedimentation rate changes may impose amplitude modulation upon precession (and short eccentricity?) as an artifact, which could complicate the assessment of the long term "grand cycles".*

The sentence is not referring to sedimentation rate changes but varying sedimentation rates at different sites. Higher sedimentation rates in the order of 3 to 5 cm / kyr lead to pronounced precession cycle recordings. Slower sedimentation rates tend to amplify the modulation of precession cycles, thus eccentricity. Secondly, as stated in Westerhold & Röhl (2009), the XRF data from 1258 show strong eccentricity-modulated precession cycles, meaning that precession cycles dominate (due to the relatively high sedimentation rates), but these cycles are clearly modulated by eccentricity. Compared to lower sedimentation rate sites with 1 to 2 cm /kyr the modulation of eccentricity is less pronounced in the XRF data. We will clarify this in the revised version modifying Page 11, lines 4-5.

*Page 11, line 19; Figure S9 caption; Figure S11 caption: Please specify the details of the detrending approach utilized, so that it can be replicated in future work. Note that Astrochron includes several functions for detrending that may be of utility here (e.g., functions 'noLow' and 'noKernel').*

The full sentence is as follows "*Published benthic and bulk stable isotope data were combined for Leg 208 and Site 1258 (Fig. 2 and S7), plotted on the 1263 rmcd and detrended for long term trends (Fig. S9)*".

Figure S9 caption says "*Second, a long-term average (thick grey) was defined graphically to avoid removing the apparent 405-kyr cycle*". The bulk and benthic isotope data have been linearly interpolated at 2 cm spacing. Then the data were smoothed using the IGOR Pro smooth operation using binomial (Gaussian) smoothing and 30001 points in the smoothing window. We smooth the data using a Gaussian filter and then subtract the smoothed curve from the original data to form a residual curve. We chose a Gaussian filter because it weights the center of the smoothing window more than the flanks (as opposed to a Boxcar filter which weights all values within the smoothing window equally). The smoothing factor (e.g., 1000, 10000, 30000) dictates how wide the effective window is. The wider the window, the less weighting is applied to the center and the significant contributions to the smoothed value extend further out from the center. The choice of smoothing number is subjective; selected by the operator through multiple trials to best eliminate one-off data shifts while including cyclic signals. Low smoothing numbers tend to accentuate high frequency signals in the residual while larger smoothing numbers include more low frequency power.

We will make sure to add these details to the revised manuscript.

*Page 18, line 13: Please note the study by Laurin et al. (2016), which provides additional independent confirmation of the eccentricity pacing of these hyperthermals.*

We will add the reference to the revised version (Laurin, J., Meyers, S.R., Galeotti, S., and Lanci, L. (2016). Frequency modulation reveals the phasing of orbital eccentricity during Cretaceous Oceanic Anoxic Event II and the Eocene hyperthermals. Earth and Planetary Science Letters 442, p. 143-156)

*Figure S3 (item 1). It is excellent to see that this study evaluates the reproducibility of the XRF Fe data, which is standard practice when presenting most geochemical results, but often ignored in XRF scanning studies….*

There might be a misunderstanding here. Figure S3 shows the intercalibration for Fe intensity data obtained from different generations of XRF core scanners and their distinct hardware for Site 1262 and 1267. NOT reproducibility. This was done to be able to plot all data on the same y-axis. Thus the extended comment in this paragraph is not really relevant for the study.

*Figure S3 (item 2): It is necessary to include a similar analysis and plot for the iron data from Site 1263.*

See above. The intercalibration is only needed to plot the data on the same y-axis and be able to analyze the full data set for time series analysis. It will not change or improve the results of the manuscript.

*Figure S12: While the proposed match between the theoretical astronomical solution and the benthic carbon isotope data seems plausible throughout much of the record, the interval from 51-52 Ma shows a response that is opposite to what theory predicts. This requires some further comment in the manuscript.*

The interval mentioned is around ~260 rmcd at Site 1263 where due to a major shift in $\delta^{13}C$ two tuning options identifying two or three 405 kyr cycles were proposed by Lauretano et al. (2016). In chapter *4.1 How many 405-kyr cycles represent Chron C23?* this particular interval is discussed in detail. The data shown in Figure S12 are detrended compiled benthic stable isotope data. The major carbon shift at ~260 rmcd at Site 1263 complicates the tuning procedure. But taking other data into account (bulk carbon isotopes, XRF Fe intensities; see Figure 3 of the submitted manuscript) we are confident about the quality of the tuning in this interval.

In the revised manuscript we will add a critical comment to chapter 4.1 dealing with the unusual carbon isotope data pattern in the 51 to 52 Ma interval.

---

## Referee Comment (RC3) · Anonymous Referee #3 · 6 Apr 2017

The manuscript describes an impressive effort to establish a coherent astronomically calibrated Ypresian time scale using new and previously published XRF, isotope, nannofossil and magnetic data from Walvis Ridge cores. I will restrict my comments to the paleomagnetic data and analysis, as this is my area of expertise. The processing of samples and data at the individual specimen level appears to be well done with no major issues. As previous workers have found, the interpretation of the data in terms of a magnetostratigraphy is more challenging. I don't necessarily object to the final interpretation, but my concerns lie in the lack of clarity in how the error bars were assigned and how the final magnetostratigraphy was selected. The authors are very

vague on how this was done. There is no description of how error bars were placed on the beginning and end of each polarity interval (Fig. 2, Fig. 4). In some cases, the placement appears to be highly subjective. As one extreme example, it is stated in the text that there is no interpretable signal below 260 rmcd at site 1263, yet they identify polarity boundaries along with error bars within this interval (Fig. 2a). It is traditional to use a gray bar (instead of black or white) to denote intervals of ambiguous polarity. This might be helpful here to make it clear which parts of the record are truly unresolvable in the authors' opinion. It is also unclear how errors from each individual site were propagated into the final magnetostratigraphy and/or how this stratigraphy was decided upon. They merely say that it was "based on the integration of all data and evaluation of errors." I think you need to be more explicit.

I was unable to find any of the supplementary tables, so I can't evaluate what's reported there. Perhaps the tables clarify how the final stratigraphy was selected?

In (main text) tables 1 and 2, I think there is an issue with the age and time units. The ages are reported in millions of years, but the uncertainty is reported in what? I assume that 47.723 Ma +/- 118 Myr is not accurate?

---

## Author Comment (AC3) · 10 Apr 2017

Dear anonymous referee #3,

Thank you very much for your review and specific comments on the magnetostratigraphy part of our study.

Below we will reply to the comments.

*The manuscript describes an impressive effort to establish a coherent astronomically calibrated Ypresian time scale using new and previously published XRF, isotope, nannofossil and magnetic data from Walvis Ridge cores. I will restrict my comments to the paleomagnetic data and analysis, as this is my area of expertise. The processing of samples and data at the individual specimen level appears to be well done with no major issues. As previous workers have found, the interpretation of the data in terms of a magnetostratigraphy is more challenging. I don't necessarily object to the final interpretation, but my concerns lie in the lack of clarity in how the error bars were assigned and how the final magnetostratigraphy was selected. The authors are very vague on how this was done. There is no description of how error bars were placed on the beginning and end of each polarity interval (Fig. 2, Fig. 4). In some cases, the placement appears to be highly subjective. As one extreme example, it is stated in the text that there is no interpretable signal below 260 rmcd at site 1263, yet they identify polarity boundaries along with error bars within this interval (Fig. 2a). It is traditional to use a gray bar (instead of black or white) to denote intervals of ambiguous polarity. This might be helpful here to make it clear which parts of the record are truly unresolvable in the authors' opinion. It is also unclear how errors from each individual site were propagated into the final magnetostratigraphy and/or how this stratigraphy was decided upon. They merely say that it was "based on the integration of all data and evaluation of errors." I think you need to be more explicit.*

We are pleased to hear that referee 3 has no objection of the final interpretation. However, clarification in how the error bars were assigned and how the final magneto-stratigraphy was selected is requested.

Unfortunately referee 3 did not request the supplementary data from the editor. All data used for this study will be available open access at http://doi.pangaea.de/10.1594/ PANGAEA.871246 by the time the manuscript will be accepted for publication. Simultaneously with the submission of the manuscript the dataset was also submitted and can be requested by the referees from the editor. Thus, we kindly ask the editor to confidentially provide the dataset to referee 3 to provide the chance for clarification how the final stratigraphy was selected. For example, Table S27 "Magnetostratigraphy ODP 1263" clearly expresses that the C23n.2n/C23r reversal below 260 rmcd is uncertain. Similar for ambiguous intervals at Site 1267. For a revised manuscript version, we will mark intervals of ambiguous polarity with a gray bar (instead of black or white) as recommended by referee 3. In doing so it will make clear which parts of the record are unresolvable.

The assignment of error bars, as with all magnetostratigraphic data, is a quite subjective endeavor. We identified reversed or normal polarity from carefully evaluating the inclination data as explained in the main text. The error bar for Leg 208 data marks the interval where the inclination shifts from clearly reversed to clearly normal polarity or vice versa. Poorer sample resolution and/or ambiguous or transitional inclination values across a reversal thus will increase the error bar. We did not apply an inclination threshold value to mark a shift in

polarity because the reversals occur in different seafloor depth at all sites. Drilling depth and compaction difference between sites might have affected the inclination at each site differently. A much sharper-defined length of error bars could be derived from higher-resolution data (e.g., by analyzing u-channels). However, this could be focus of a follow up investigation but is not the accomplishment in the context of our already data rich and complex manuscript.

The final combined magnetostratigraphy for the Ypresian at Sites 1258, 1262, 1263, 1265 and 1267 is given in Table S45 of the supplementary dataset. The individual reversals were all identified at one site and thus the error bars assigned in the individual site (as in tables S26 to S30) are transferred into Table S45. The definition for each reversal is described in the main text in chapter *3.3 Magnetostratigraphic results and interpretation*. In combination with the detailed supplementary figures and the dataset tables we are confident that our interpretation can be followed appropriately. For a revised version we will add more details on the error evaluation and final magnetostratigraphy to chapter *3.3 Magnetostratigraphic results and interpretation*.

*In (main text) tables 1 and 2, I think there is an issue with the age and time units. The ages are reported in millions of years, but the uncertainty is reported in what? I assume that 47.723 Ma +/- 118 Myr is not accurate?*

Thanks for pointing this out. We will add a note to the tables on that the error is given in kyr.

---

## Author Response (AR1)

✉ Universität Bremen **I MARUM I** 28359 Bremen

Dr.
**Thomas Westerhold**
Research Scientist

Leobener Strasse 8
MARUM building, Room 0220
28359 Bremen – Germany

Telefon +49 421 218 – 65672
E-Mail twesterhold@marum.de
www www.marum.de

To the Editors of

**Climate of the Past**

.

27/May/2017

Dear Professor Sluijs,

Please accept our submission of the revised manuscript entitled "Astronomical Calibration of the Ypresian Time Scale: Implications for Seafloor Spreading Rates and the Chaotic Behaviour of the Solar System?" for consideration by *Climate of the Past*.

We thank you and the referees for your time and effort to evaluate our manuscript. We appreciate the comments and constructive criticism of the three referees that encouraged us to carefully revise the manuscript accordingly.

Detailed answers to the comments of the referees were already posted on the *Climate of the Past Discussions* web page. Below we describe our approach on modifying the manuscript for the revised version.

We also would like to reiterate that the aim of the manuscript is a very complex data synthesis. Nevertheless, in the revised version we included a more rigorous statistical testing. The application of the statistical methodologies as proposed by the referees definitely helped to clarify and strengthened the manuscripts conclusions. We hope that our data compilation will provide the base for insightful research in the Ypresian toward an improved understanding of climate dynamics in a high $pCO_2$ world.

We hope that the revised manuscript now meets the requirements to be published in *Climate of the Past*.

Sincerely,

Thomas Westerhold, Ursula Röhl, Thomas Frederichs, Claudia Agnini,
Isabella Raffi, James C. Zachos, and Roy H. Wilkens

**Modifications to the initial submission**

**Changes according to the comments by referee #1 Frits Hilgen**

Referee #1 raised five major issues. The first and most important one was the missing statistical identification of the very long period ~2-Myr eccentricity minima. In the initial submission we only had included the visual determination of the minima being extremely important to evaluate the match between different orbital solutions and the geological data.

In the revised manuscript (as also outlined in the discussion reply to referee #1) we subdivided initial chapter *4.2 Which orbital solution applies best for astronomical tuning?* into three new subchapters. The new subchapter 4.2.2 *Statistical evaluation and determination* now includes the independent statistical testing using ASTROCHRON and ENVELOPE. A new figure (Fig. 4) comparing the amplitude modulation (AM) of the short eccentricity cycle between the La2004, La2010, and La2011 orbital solutions and Fe intensity data from ODP Sites 1258, 1262 and 1263 was added for clarification. Subchapter *4.2.3 Potential distortion by non-linear response of the climate system* addresses the 2$^{nd}$ major issue of referee #1 on influences of the carbon cycle perturbations during hyperthermal events on the records used to extract orbital cycles.

The 3$^{rd}$ issue of missing a more detailed discussion regarding the fingerprint of the transition from libration to circulation was tackled by adding a section (as outlined in the reply to referee #1) to chapter *5.1 Geological evidence for chaotic behaviour of the solar system in the Ypresian?* for clarification on the expression of the 2-Myr cycle (related g4-g3) both in the solutions and our records in detail.

The 4$^{th}$ and 5$^{th}$ issues deal with an apparent lack of discussion regarding the reliability of astronomical solutions. We added two more text intervals to chapter *5.1* (as mentioned in the reply to referee #1) – see marked changes in the revised version.

A minor point in the first line of p.14 of the initial submission was corrected; we rephrased the sentence as follows: "Older than 50 Ma the location of the very long eccentricity nodes in available orbital solutions (La2004 – Laskar et al., 2004; La2010 – Laskar et al., 2011a; La2011 – Laskar et al., 2011b) are much more uncertain."

All five issues have been addressed and incorporated in the revised manuscript that clearly benefited from the referee´s suggestions.
* * *
**Changes according to the comments by referee #2 Stephen Meyers**

Referee #2 mentioned the new study by Ma et al. 2017, which provides geologic evidence confirming the chaotic behavior of the Solar System, through the identification of a chaotic resonance transition during the Coniacian (~85-87 Ma). The study was not published at the time when the initial manuscript was submitted. In the revised version we have taken this into account.

Referee #2 recommended two additional approaches: 1) run the Astrochron R-script to test for the expected amplitude modulations in the 405 ka tuned data, and (2) construct an analysis similar to Table 1 of Ma et al. (2017), to eliminate the possibility that changes in sedimentation rate (including hiatus) are influencing the observed modulation patterns.

Indeed referee #1 had asked for a similar type of analysis. Therefore, the revised manuscript now includes an independent statistical test extracting the amplitude modulations in the data in a new subchapter 4.2.2 *Statistical evaluation and determination*. We did not follow the second recommendation (as explained in the reply to referee #2), because it would not provide better data constraints as already present in the extensive data compilation set from multiple records.

Referee #2 suggested to run an Astrochron R-script he provided to produce a summary figure illustrating the characteristic "grand cycles" that are expressed in the amplitude (and power) modulation of the short eccentricity terms. We ran the script on the La2004, La2010b and La2010d and La2011 orbital solutions and added the new Figure S14 to the supplement. The analysis shows that in fact all eccentricity amplitude modulations from the different theoretical astronomical solutions show a different and to some extend unusual behavior between 52-54 Ma (new Fig. S14).

Regarding the referee´s suggestion to elaborate on the proposed link between chaotic orbital behavior and changes in ocean spreading rate: we argue that the questions raised by referee #2 are good questions for future research projects and modeling studies that should be address to test our new findings, but were not elaborated in our study (see reply to referee #2) as this would be far beyond the scope of our study. Setting up a model for the complex and chaotic mantel flow is highly sophisticated and should be undertaken by exerts in the field.

The additional comments of referee #2 we have addressed as follows:
1) Page 11, lines 4-5 (initial manuscript): clarified in *4. Astronomical Calibration of the Ypresian*.
2) Page 11, line 19; Figure S9 caption; Figure S11 caption: added detail of the detrending applied in *4. Astronomical Calibration of the Ypresian.*
3) Page 18, line 13: added the reference to the revised version Laurin, J., Meyers, S.R., Galeotti, S., and Lanci, L.: Frequency modulation reveals the phasing of orbital eccentricity during Cretaceous Oceanic Anoxic Event II and the Eocene hyperthermals. Earth and Planetary Science Letters, 442, 143-156, 2016.
4) Figure S12, Referee #2: "*While the proposed match between the theoretical astronomical solution and the benthic carbon isotope data seems plausible throughout much of the record, the interval from 51-52 Ma shows a response that is opposite to what theory predicts*." We added critical comments in *5.1 Geological evidence for chaotic behaviour of the solar system in the Ypresian?*: "None of the available orbital solutions perfectly fit the geological data. However, it is important that we isolated the transition in the data, which is also present in the La2010b and La2010c solutions. The short eccentricity cycle pattern both in the solutions and the geological data will not match perfectly beyond 50 Ma where the uncertainty in the solutions increase (as discussed in Westerhold et al. 2012). Still the geological data and La2010b/c solutions are very similar from 53.5 Ma to the PETM. In the interval from 51 to 52 Ma, the most difficult part to tune in the Ypresian, multiple hyperthermal events and the shift in carbon isotope data make a direct comparison much more difficult. It has to be noted that the eccentricity solutions from La2010b/c might not be completely reliable in this interval. Despite the uncertainties we provide a tuned age model to La2010b/c because the match in the 53.5 Ma to the PETM interval is good enough to do so. If in doubt, the provided 405-kyr age model can still be used."

**Changes according to the comments by anonymous referee #3**

Referee #3 listed specific comments on the magnetostratigraphy part of our study. He has no objection of the final interpretation but asked for some clarification in how the error bars were assigned and how the final magnetostratigraphic interpretation was done.
We hope that the manuscript´s dataset was forwarded to referee 3 in order to provide a chance for direct insight on how the final stratigraphy was constructed (see reply to referee #3). As requested by referee #3 we marked all ambiguous polarity interpretations with gray bars (Figures 2a, 2b, S1a, S1b, S4a, and S4b). Additionally we added more details on the error evaluation and final magnetostratigraphy to chapter *3.4 Magnetostratigraphic results and interpretation*. We also corrected the errors given in Tables 1 and 2 from kyr to myr.

[revised manuscript text omitted]

---

## Referee Report (RR1)

**Comments on the revised Climates of the Past Discussion Paper:**
*Astronomical Calibration of the Ypresian Time Scale: Implications for Seafloor Spreading Rates and the Chaotic Behaviour of the Solar System?*

**Referee: Stephen R. Meyers, University of Wisconsin-Madison**

This manuscript seeks to address several related objectives: (1) to resolve a controversy about appropriate astronomical tuning within magnetochron C23 (how many 405 ka cycles are present?), (2) to solve the so-called "50 Ma discrepancy", (3) to identify the correct theoretical model(s) for short eccentricity tuning during the Yepresian, (4) to test for the chaotic behavior of the Solar System, and (5) to provide a complete Yepresian Astronomical Time Scale (YATS), yielding time-calibrated biostratigraphic, magnetostratigraphic, and chemostratigraphic data. The study proposes to achieve these goals, and in addition, suggests that an increase in ocean spreading rate between 51-52.5 Ma is linked to chaotic orbital behavior, through an influence on dynamic mantle flow.

The fundamental questions that this study seeks to address are well established in the literature, and they constitute important issues worthy of discussion in Climates of the Past. Therefore, in reviewing this manuscript I have considered two essential questions: is the data synthesis conducted appropriately, and are the interpretations robust? As mentioned in my prior review, the data production and assimilation campaign that is the foundation of this study is an impressive effort. Given the wide range of topics this study covers, however, there are varying degrees to which each hypothesis has been tested, and in my opinion, improvements are needed to more clearly describe the hypothesis testing and the robustness of the results. I provide more information on these recommendations below. Once these issues are addressed, I believe the manuscript will be suitable for acceptance, and it will be an important and valuable contribution to Climates of the Past.

Before proceeding however, I would like to note that there is an important aspect of this study that is not presently emphasized, but has broad relevance to the field of chronostratigraphy. This work provides an excellent case study into the challenges, uncertainties and potential of chemostratigraphy, magnetostratigraphy and biostratigraphy, made possible by the integration of comprehensive data sets spanning two ODP campaigns (Leg 207 & 208). There are valuable lessons from this study that apply more broadly to the science of interpreting magnetostratigraphic, biostratigraphic and chemostratigraphic data. I recommend emphasizing this important aspect in the abstract and introduction. It is already discussed nicely in the main body of the paper.

**1. Secular Resonances and Chaos**
**1.1 Statistical Tests for Chaos**
As outlined in Ma et al. (2017, Nature 542, 468-470), there exist several tests for verifying the emergence of the (s4 − s3) − (g4 − g3) secular resonance in the geologic record. Emergence of this secular resonance underlies the chaotic behavior predicted by the theoretical models.

A. For quantification of g4-g3:
   • Amplitude modulation of the 405 kyr long eccentricity cycle
   • Amplitude modulation of short eccentricity
   • Amplitude modulation of precession

B. For quantification of s4-s3:
   • Amplitude modulation of ~40 kyr obliquity term

C. Phase relationships:
   • For example, the anti-phased relationship between long-eccentricity "grand cycles" and short eccentricity "grand cycles"

D. Radioisotopic time control, or biostratigraphy-magnetostratigraphy-chemostratigraphy calibrated to radioisotopic dates, to guard against missing portions of "grand cycles" that could be erroneously interpreted as indicating a resonance transition.

Note that, for parts A-C, there are numerous ways to go about evaluating the modulations (e.g., complex demodulation, power spectrum integration). An essential revision to the manuscript is the explicit discussion of these 6 tests, an evaluation of which ones the Yepresian data passes, which it fails, and which are not possible. This will make it more obvious to the non-specialist where ambiguity exists, and where it doesn't.

In the prior review of this study, I recommended the authors construct an analysis similar to that in Table 1 of Ma et al. (2017), to evaluate the possibility that changes in sedimentation rate (including hiatus) are influencing the observed modulation patterns (test D above). It was not my intention to imply that the authors require high-precision radioisotopic data for the success of their study. Rather, I hope that the authors provide some quantification of how much time could be missing, and where in the stratigraphy it is most likely absent, based on independent constraints. To some degree, suggestions of this are woven into the existing text (e.g., pg. 12, lines 1-2), but it is not presented in a comprehensive and linear manner. For the sake of transparency and evaluation by colleagues, this type of information is essential; the resulting uncertainties may be very large, but that is important to explicitly acknowledge. And compared to the tremendous amount of work that has gone into generating and interpreting the impressive records from Leg 207 and 208, this is a relatively small investment of time.

It is also worth emphasizing in the manuscript that, in absence of high-precision independent time control, it is essential to combine multiple records from multiple regions, to help safeguard against incompleteness that is otherwise difficult to assess qualitatively and/or quantitatively.

Finally, given that the approach used in this manuscript is to develop an astronomical time scale "anchored" to a theoretical solution, the 6 statistical tests outlined above also provide a critical assessment of the veracity of the hypothesized YATS.

**1.2 Additional Comments on Node Identification**

One very interesting result of this study is the reinterpretation of cycles in the interval spanning 68-92 rmcd at Site 1258, suggesting the emergence of obliquity forcing.  In precession dominated settings such as those investigated here, the emergence of obliquity can occur during eccentricity nodes.  So, this provides some independent evidence for the existence of an eccentricity "grand cycle" node at ~80 rmcd (Site 1258).

Is there any indication of obliquity cycles during the other proposed eccentricity nodes?  If so, this would provide quantitative evidence supporting correct eccentricity node identification (this is essentially another way of at getting at $g4-g3$, in settings that are almost exclusively precession forced).

**2. XRF Data**

As noted in the prior review, the intercalibration results from Site 1267, comparing XRF scanning campaigns conducted on two different instruments, look excellent.  But an $r^2$ value of 0.09 associated with the Site 1262 XRF Fe data sets indicates a surprisingly poor correlation. In their response, the authors argue:

> "There might be a misunderstanding here. Figure S3 shows the intercalibration for Fe intensity data obtained from different generations of XRF core scanners and their distinct hardware for Site 1262 and 1267. NOT reproducibility."

I understand the practical reason for conducting this intercalibration, but this exercise is equivalent to evaluating inter-instrument reproducibility, and the results are not reassuring. The good news is that Site 1262 doesn't appear to be really essential to the conclusions. So rather than using problematic data, my recommendation is to do one of the following: (1) rescan the critical interval, or (2) eliminate the suspect data from the manuscript.  But if the authors choose to keep the data in the manuscript, at a bare minimum, they should explicitly acknowledge the problematic nature of this data in the main text, and indicate that their primary conclusions do not rely on it.

Again, I think it would be worthwhile, as a general practice, to adopt an XRF data reporting approach that quantifies instrument stability (e.g., see Figure A.1 of Ma et al, 2014), and reproducibility based on duplicate analyses (e.g., see Figure A.2 and Table A.1 of Ma et al., 2014). It seems inconsistent that there is no discussion of data quality in section 2.1 ("XRF core scanner data"), in contrast to the other geochemical data ("2.2 Bulk stable isotope data").

**3. Chaotic orbital influence on dynamic mantle flow**
As mentioned in my prior review, I am concerned about the proposed link between chaotic orbital behavior and changes in ocean spreading rate, because no description is made of the physical mechanism by which it is manifested, either qualitatively (how does orbital behavior impact mantle flow, and how would a chaotic transition thus be expressed as an increase in spreading rates?), or quantitatively through modeling. Of course, some level of speculation is important and productive in science; if instead the authors were proposing a new link between chaotic orbital behavior and climate/oceanographic change, this would not be too difficult to justify as plausible, considering that there exist good conceptual and quantitative models for how astronomical cycles influence insolation and climate.

The critical difference with the present study is its speculation that astronomical perturbations are sufficient to influence mantle flow, without discussion of an existing theory to support it, or elaboration on how this could plausibly happen. If the authors were correct, however, this would be an important discovery. I encourage the authors to further develop the underlying theory in this manuscript (at least in a qualitative sense), or to reserve discussion of this speculative hypothesis for a future manuscript where it can be treated in greater detail.

**4. Additional comments**
pg., 17, Lines 10-11: note that mechanisms have been proposed that account for low-latitude expression of obliquity forcing (see STIG; Bosmans et al., 2015, Climates of the Past 11, 1335–1346).

For individuals not familiar with the site numbers, it would be helpful to explicitly distinguish the Walvis Ridge data (Leg 208; Sites 1261, 1263, 1265, 1267) from the Demerara Rise data (Leg 207, Site 1258), when feasible, throughout the manuscript.

Figure S14. It would also be helpful to put "kyr" next to the numbers in the 2 and 3 cycle model plots (similar to the lower plot labels, "cm").

---

## Author Response (AR2)

✉ Universität Bremen I **MARUM** I 28359 Bremen

Dr.
**Thomas Westerhold**
Research Scientist

Leobener Strasse 8
MARUM building, Room 0220
28359 Bremen – Germany

Telefon   +49 421 218 – 65672
E-Mail    twesterhold@marum.de
www       www.marum.de

**Prof. Dr. Appy Sluijs**

**Editor**

*Climate of the Past*

.

24/July/2017

Dear Professor Sluijs,

   Please accept our submission of the second revision of the manuscript entitled "Astronomical Calibration of the Ypresian Time Scale: Implications for Seafloor Spreading Rates and the Chaotic Behaviour of the Solar System?" for consideration by *Climate of the Past*.

   We thank you and referee Stephen Meyers for taking the time to review the revised manuscript in detail. The comments and constructive criticism encouraged us to carefully address all issues raised to improve the manuscript.

   Detailed answers to the comments of referee Stephen Meyers are given below including the modifications to the manuscript for the 2nd revised version.

   First, we would like to address your comments in the decision letter from 26th Jun 2017. We are pleased that you consider our manuscript of fundamental importance. In particular, you point out that "*a good manuscript can become an important one only if the complete set of statistical tests is included. So please do consider his advice in his section 1.1. In addition, please also consider his point 2, regarding XRF data quality carefully*".

   As pointed out in the reply to referee Stephen Meyers we believe that most if not all of the statistical tests are already in the manuscript. Any aim of trying to restructure the current manuscript along the recommended point-by-point tests would be very complicated and corrupt the current approach of presenting the logical step-by-step development of the timescale. The only test not present in the first revised manuscript is the anti-phased relationship between long-eccentricity "grand cycles" and short eccentricity (the fifth point, "C" in the referee's comments). We now added the analysis to supplementary Figure S15, which is fully in accordance to Ma et al. (2017)'s approach. A detailed discussion on radio-isotopic time control to guard against missing portions of "grand cycles" (sixth point in the referee's comments) is as already detailed in the rebuttal to the first revised ms. not possible due to missing radio-isotopic dates in the investigated ODP cores. Therefore, the subsequently by referee Stephen Meyers requested quantification of missing time cannot be realized (or would just contribute a lengthy, but very speculative discussion on potential missing intervals). This could only be realized by retrieving new Ypresian records in other regions to test our YATS. Nevertheless, we have successfully revised the XRF data quality issue as recommended in the reply to referee Stephen Meyers.

Regarding the issue "…*obliquity during eccentricity nodes (1.2)…*" on the eccentricity node identification we added appropriate text to the *4.2.2 Statistical evaluation and determination chapter*.

Finally, "…*speculations regarding the influence of astronomical perturbations influencing mantle flow (3)…*" have been suggested to be added to a revised 2[nd] version of the manuscript. We have to reiterate that the aim of the manuscript is a very complex data synthesis to establish a robust Ypresian chronostratigraphy. Considering the maybe random coincidence of orbital variations and changes in mantle dynamics we decided to remove the still highly speculative hypothesis from the revised 2[nd] version of the manuscript. All text passages thus have been removed accordingly. Although being surprised and thrilled to discover that there at least seems to be a temporal correlation of the chaotic transition, the benthic carbon isotope shift, and the change in deep-sea spreading rates we think that prior to publication of the hypothesis a simple modeling approach is require to test basic ideas.

Please also consider to contact referee Frits Hilgen again for a second view on the points raised by referee Stephen Meyers and our here submitted rebuttal.

We hope that the 2[nd] revision of the manuscript including the comment to the referee now meets the requirements to be published in *Climate of the Past*.

Sincerely,

Thomas Westerhold, Ursula Röhl, Thomas Frederichs, Claudia Agnini,
Isabella Raffi, James C. Zachos, and Roy H. Wilkens

**Modifications to the 1[st] revised submission**

**Changes according to the comments by referee Stephen R. Meyers**

We thank referee Stephen R. Meyers for his detailed and constructive review of our revised manuscript. Several major points have been raised to improve the manuscript that we address here one by one.

1) Referee Stephen R. Meyers suggested: "This *work provides an excellent case study into the challenges, uncertainties and potential of chemostratigraphy, magnetostratigraphy and biostratigraphy, made possible by the integration of comprehensive data sets spanning two ODP campaigns (Leg 207 & 208). There are valuable lessons from this study that apply more broadly to the science of interpreting magnetostratigraphic, biostratigraphic and chemostratigraphic data. I recommend emphasizing this important aspect in the abstract and introduction. It is already discussed nicely in the main body of the paper*".

In the 2[nd] revised version of the manuscript we have added text at the end of the abstract and introduction pointing to the broader significance of our work for chronostratigraphy in general.

2) Six Statistical Tests for Chaos. Referee Stephen R. Meyers suggested: "*An essential revision to the manuscript is the explicit discussion of these 6 tests, an evaluation of which ones the Ypresian data passes, which it fails, and which are not possible. This will make it more obvious to the non-specialist where ambiguity exists, and where it doesn't.*"

The main point of criticism is that the manuscript does not follow the six tests for verifying the emergence of the (s4 – s3) – (g4 – g3) secular resonance in the geologic record as outlined in Ma et al. (2017) (Nature 542, 468-470). Based on the initial review of Stephen R. Meyers we already added crucial points regarding this issue to chapter *5.1 Geological evidence for chaotic behaviour of the solar system in the Ypresian?* in the discussion section of the manuscript. However, obviously more details are asked for.

We think that most of the points made are already incorporated in the revised manuscript as outlined in the rebuttal and tracked-changes-version of the 1[st] revision which benefited from the detailed three constructive reviews. The first three points ("A" in the referee's comments) aim at the quantification of g4-g3 modulation changes. All of this is indeed already discussed in detail in chapter *4.2.2 Statistical evaluation and determination* including independent statistical testing using ASTROCHRON and ENVELOPE, plus adding a new figure (Fig. 4) comparing the amplitude modulation (AM) of the short eccentricity cycle to the revised ms.

The fourth point ("B" in the referee's comments), quantification of s4-s3 modulation changes refers to the amplitude modulation of the ~40 kyr obliquity term. This is not possible with our data as detailed in chapter *5.1 Geological evidence for chaotic behaviour of the solar system in the Ypresian? …*". One would need to extract the AM of both obliquity and precession in a geological dataset in order to detect the transition from libration to circulation (Laskar 1999), which is almost impossible requiring a record that is both driven (or at least influenced) by both high and low latitude processes (Laskar et al., 2011). Obliquity AM could be extracted from benthic $\delta^{18}O$ records, for example, if deep-sea temperature variations are continuously affected by obliquity. This is **not** the case for the Paleocene and early Eocene (Littler et al., 2014, Zeebe et al., 2017), thus investigation of the AM of obliquity is difficult with the currently available records. Therefore Laskar et al. (2011) recommended to search for a modulation of the g4 – g3 period, the ~2.4 myr eccentricity modulation.

We have doubts that it is at all possible to extract both precession and obliquity modulation from the same record (in that case the FMI record) as done by Ma et al. 2017. This issue was already discussed by Jacques Laskar at various occasions. A reliable modulation pattern is only possible if the data are equally influenced by precession and obliquity over the entire investigated interval. This means strong obliquity cycles, clearly reflecting the modulation, as in the 41-kyr world of the Plio-Pleistocene combined with strong influence of eccentricity modulated precession at the same time. This record would look a bit like the 65°N solar insolation curve calculated from the Laskar solutions, and the only record that looks like this is from Ceara Rise ODP Leg 154 over longer intervals (e.g., Pälike et al. 2004). In a perfect situation, as written in the manuscript, two synchronized records are available with one recording the obliquity modulation (e.g. $\delta^{18}$O of benthic forams) and the other recording precession modulation (e.g. lithology changes, XRF Fe data). However, this kind of records still have to be identified for the entire Ypresian to further examine on this fourth criterion. As far as we know from published high-resolution benthic isotope and other geochemical or lithological records there is no evidence for a strong contribution of obliquity in the signal that reliably allows the extraction of the obliquity AM for the Ypresian. This was mentioned in the revised manuscript already, as given above.

The fifth point ("C" in the referee's comments) phase relationships refers to the anti-phased relationship between long-eccentricity "grand cycles" and short eccentricity "grand cycles". We added the long eccentricity cycle bandpass filter to supplementary Figure S13 of the 2$^{nd}$ revised version where the grand cycles using the Meyers script have been plotted. This figure shows the anti-phased relationship as in Ma et al. 2017 for the interval 40 to 60 Ma for different orbital solutions.

The sixth point ("D" in the referee's comments) refers to radio-isotopic time control to guard against missing portions of "grand cycles". As mentioned in the rebuttal to the revised version of the manuscript this is not possible due to missing radio-isotopic dates in the ODP cores. We elaborate more on this in the following comment three of the referee.

3) Referee Stephen R. Meyers comments on age uncertainties: "*I hope that the authors provide some quantification of how much time could be missing, and where in the stratigraphy it is most likely absent, based on independent constraints. To some degree, suggestions of this are woven into the existing text (e.g., pg. 12, lines 1-2), but it is not presented in a comprehensive and linear manner. For the sake of transparency and evaluation by colleagues, this type of information is essential; the resulting uncertainties may be very large, but that is important to explicitly acknowledge.*"

This is a very difficult task. We could also reword the issue asking: How do we know something is missing? The simple answer is, we do not know until another section would be available indicating that something is missing or alternatively validating that the Ypresian record is complete. Without this information it is very hard to predict if there are gaps in the current records. We made clear in the manuscript that a single site or region record potentially could include gaps and/or condensed sections that could only be detected if correlated to another record. Deriving errors from calcareous nannofossil datums at (a) different site(s) is not recommended, because these are not perfectly synchronous between Site 1258 and Leg 208 sites also considering the potential overall error in defining biozones. Even between Leg 208 sites some events are not accurately synchronous probably due to sampling and/or depth related (dissolution etc.) issues. It is also the nature of magnetostratigraphy that error analysis is not straight as different site exhibit slightly different results related to similar issues as biostratigraphy. How to quantify errors using independent constraints that we can hardly grasp? We also do not have independent constraints like radio-isotopic age datums in the ODP Sites. Please note that we included a discussion on potentially missing or extra 405-kyr cycles in chapter *4.1 How many 405-kyr cycles represent Chron C23?*).

Therefore, we refrain from adding a speculative discussion on potential gaps. But have now emphasized "*in the manuscript that, in absence of high-precision independent time control, it is essential to combine multiple records from multiple regions, to help safeguard against incompleteness that is otherwise difficult to assess qualitatively and/or quantitatively*", as recommended by the referee.

4) Referee Stephen R. Meyers: "*Finally, given that the approach used in this manuscript is to develop an astronomical time scale "anchored" to a theoretical solution, the 6 statistical tests outlined above also provide a critical assessment of the veracity of the hypothesized YATS*"

We are confident that our manuscript now addresses all of the issues dealing with the six tests. We refrain from totally rewriting the manuscript while strictly following the recommended six test. Please be also aware that most of the study was developed before Ma et al. (2017) was published. In addition, as mentioned already on the Ma et al. (2017) approach of extracting both precession and obliquity AM from the same record (that is clearly dominated by precession at first glance) is questionable. Extended statistical testing of our data will not change the results, interpretation and implication of our study, and in case of still existing doubts we recommend the editor to forward the 2[nd] revised version and rebuttal to referee Frits Hilgen for an alternate opinion on this matter.

5) Additional Comments on Node Identification

Referee Stephen R. Meyers: "*One very interesting result of this study is the reinterpretation of cycles in the interval spanning 68-92 rmcd at Site 1258, suggesting the emergence of obliquity forcing. In precession dominated settings such as those investigated here, the emergence of obliquity can occur during eccentricity nodes. So, this provides some independent evidence for the existence of an eccentricity "grand cycle" node at ~80 rmcd (Site 1258).*

*Is there any indication of obliquity cycles during the other proposed eccentricity nodes? If so, this would provide quantitative evidence supporting correct eccentricity node identification (this is essentially another way of at getting at g4- g3, in settings that are almost exclusively precession forced).*"

To our opinion this is not really a way of looking at the g4-g3 resonance. It would just show that the signal in the XRF data is probably generated by processes originating from high latitudes (e.g. polar ice sheets; deep water formation). As mentioned in the text in chapter 5.1 "Obliquity AM could be extracted from benthic $\delta^{18}O$ records, for example, if deep-sea temperature variations are continuously affected by obliquity. This is **not** the case for the Paleocene and early Eocene (Littler et al. 2014, Zeebe et al. 2017), thus investigation of the AM of obliquity is difficult with the currently available records." Thus, there is no obliquity component in the early Eocene, also not in data published in Littler et al. (2014).

In Chapter 4.1 *How many 405-kyr cycles represent Chron C23?* we discussed the first appearance of some obliquity cycles (4-5 cycles) at around 80 rmcd in the Site 1258 record (also see supplement Figure S14). The appearance of additional obliquity cycles in ODP Site 1258 in the interval from 55 to 60 rmcd was already discussed in Westerhold and Röhl (2009) and Westerhold et al. (2012). These cycles are located at the beginning of the very long eccentricity node at 50 Ma, the few obliquity cycles at 80 rmcd are located at the end of the very long eccentricity node at 52 Ma. In the interval of another node at 48 Ma Site 1258 Fe data do not clearly show obliquity cycles but low amplitude modulations of precession. All of this has been already detailed in Westerhold and Röhl (2009) (see Figure 9 therein) and Westerhold et al. (2012).

We also refrain from reiterating what has been discussed in great detail in Westerhold and Röhl (2009) and Westerhold et al. (2012) regarding the appearance of obliquity cycles in nodes of the very long eccentricity cycles. Reference to these papers are included at the appropriate locations throughout the text. In the revised second version of the manuscript we added the following text towards the end of chapter *4.2.2 Statistical evaluation and determination*: "Quantitative evidence supporting the correct eccentricity node identification can also be derived from the emergence of obliquity cycles in the data at the nodes. Obliquity is not present in the Paleocene and early Eocene (Littler et al. 2014, Zeebe et al. 2017) of the investigated records. But Site 1258 Fe intensity data show some obliquity related cycles at around 80 rmcd (also see supplement Figure S14), and from 55 to 60 rmcd corresponding to the end of the very long eccentricity node at 52 Ma and the beginning of the very long eccentricity node at 50 Ma. At another potential node (48 Ma) Site 1258 Fe data do not clearly exhibit obliquity cycles, but low amplitude modulations of precession related cyclicity (Westerhold and Röhl (2009), see Figure 9 therein). Considering all these observations they provide some independent evidence for the existence of eccentricity nodes at 50 and 52 Ma. The nodes at ~53.3 and ~54.5 Ma show no prominent obliquity cycles in the Fe records as already discussed above and in Littler et al. (2014)."

6) XRF Data

Referee Stephen R. Meyers' comments on the poor correlation of XRF data (Figure S3) for the intercalibration for Fe intensity data obtained from different generations of XRF core scanners and their distinct hardware for Site 1262 and 1267.

We replicated the statistical analysis now using the IGOR Pro software package and identified two mistakes in Figure S3. The $R^2$ values in the submission was 0.09 for 1262A-11H-1 and 0.87 for 1267A-26X-3 and -4. The repeated statistical analysis results into $R^2$ of 0.50 for 1262A-11H-1 and 0.91 for 1267A-26X-3 and -4. The regression formulas are the same. We apologize for the mistaken R2 values for both plots and have corrected this in the revised version for the 2$^{nd}$ submission.

However, a $R^2$ value of 0.50 is still not very good and the result from calibrating the records in a carbonate-rich interval with general relatively low XRF Fe intensities but more noise. Referee Stephen R. Meyers proposed to rescan the critical interval or alternatively to eliminate the noisy data. A rescan would not make a difference independent from the scanner and hardware generations. It is also not required to remove the relatively noisier data. We would like to emphasize that the intercalibration was undertaken only for allowing to plot all data on the same y-axis, cyclic variations in the data utilized for further cyclostratigraphic interpretation do not depend on the intercalibration.

7) Chaotic orbital influence on dynamic mantle flow

Referee Stephen R. Meyers comments: "*As mentioned in my prior review, I am concerned about the proposed link between chaotic orbital behavior and changes in ocean spreading rate, because no description is made of the physical mechanism by which it is manifested, either qualitatively (how does orbital behavior impact mantle flow, and how would a chaotic transition thus be expressed as an increase in spreading rates?), or quantitatively through modeling. Of course, some level of speculation is important and productive in science; if instead the authors were proposing a new link between chaotic orbital behavior and climate/oceanographic change, this would not be too difficult to justify as plausible, considering that there exist good conceptual and quantitative models for how astronomical cycles influence insolation and climate.*"

"*The critical difference with the present study is its speculation that astronomical perturbations are sufficient to influence mantle flow, without discussion of an existing theory to support it, or elaboration on how this could plausibly happen. If the authors were correct, however, this would be an important discovery. I encourage the authors to further develop the underlying theory in this manuscript (at least in a qualitative sense), or to reserve discussion of this speculative hypothesis for a future manuscript where it can be treated in greater detail.*"

We follow the advice of Stephen R. Meyers reserving the discussion of the speculative coupling of the orbital variations and changes in mantle dynamics for a future manuscript. All text passages regarding this hypothesis have been removed from the manuscript. We really appreciate the referee's enthusiastic suggestion to elaborate more on the possible influence of the chaotic transition on mantle flow dynamics. We now think that any investigation on the mechanism probably should be accompanied at least by a simple modeling approach for testing some general approaches. This is by far not the scope of our submitted manuscript on the Ypresian time scale. We were surprised and thrilled to discover that there at least seems to be a temporal correlation of the chaotic transition, the benthic carbon isotope shift, and the change in deep-sea spreading rates. To maybe address this further in the future first contacts have already been made with experts on complex and chaotic mantel flow models.

8) Additional comments a) referee Stephen R. Meyers commented: "*pg., 17, Lines 10-11: note that mechanisms have been proposed that account for low-latitude expression of obliquity forcing (see STIG; Bosmans et al., 2015, Climates of the Past 11, 1335–1346).*"

We thank referee Stephen R. Meyers for this hint. The paper mentioned deals with the summer inter-tropical insolation gradient (SITIG) showing by a sophisticated coupled ocean–atmosphere global climate model that obliquity-induced changes in tropical climate can occur without high-latitude ice sheet fluctuations. Mainly that study intends to explain the observation that obliquity seems to modulate the thickness pattern of sapropel deposits in the Mediterranean.

In the context of pg., 17, Lines 10-11 we wrote: "One needs to extract the AM of both obliquity and precession in a geological dataset in order to detect the transition from libration to circulation (Laskar 1999), which is almost impossible requiring a record that is both influenced (or driven) by high latitude and low latitude processes (Laskar et al (2011)". The mechanism of Bosmans et al. (2015) cannot explain a dominant obliquity driven record over millions of years, because the SITIG curve (as in Bosman et al. 2015 Fig. 2) shows little influence of obliquity or obliquity cycles in intervals with high precession amplitude modulations. But to be able to extract the AM of obliquity over millions of years a record is needed that is dominated by obliquity throughout, including intervals with higher AM of precession by eccentricity. Thus we leave this citation out as is does not add to the point of the discussion at pg., 17, Lines 10-11.

b) referee Stephen R. Meyers commented: "*For individuals not familiar with the site numbers, it would be helpful to explicitly distinguish the Walvis Ridge data (Leg 208; Sites 1261, 1263, 1265, 1267) from the Demerara Rise data (Leg 207, Site 1258), when feasible, throughout the manuscript.*"

This is a general issue. To add the full ODP IDs each time throughout the text will make it harder to read. Therefore, we would like to keep the wording as is. We are confident that most readers after the initial (full) definition of the sites at the beginning (introduction) will be confident in following.

b) referee Stephen R. Meyers commented: "*Figure S14. It would also be helpful to put "kyr" next to the numbers in the 2 and 3 cycle model plots (similar to the lower plot labels, "cm").*"

Thanks, we have corrected this accordingly.

[revised manuscript text omitted]

---

## Author Response (AR3)

✉ Universität Bremen **I MARUM I** 28359 Bremen

Dr.
**Thomas Westerhold**
Research Scientist

Leobener Strasse 8
MARUM building, Room 0220
28359 Bremen – Germany

Telefon   +49 421 218 – 65672
E-Mail    twesterhold@marum.de
www       www.marum.de

**Prof. Dr. Appy Sluijs**

**Editor**

*Climate of the Past*

.

08/August/2017

Dear Professor Sluijs,

   We are very pleased that our submitted manuscript entitled "Astronomical Calibration of the Ypresian Time Scale: Implications for Seafloor Spreading Rates and the Chaotic Behaviour of the Solar System?" now has been accepted for publication in *Climate of the Past*.

   We thank you and all three referees for taking the time to review the manuscript versions in great detail.

   Please find below the comments to your suggested technical corrections

Sincerely,

   Thomas Westerhold, Ursula Röhl, Thomas Frederichs, Claudia Agnini,
   Isabella Raffi, James C. Zachos, and Roy H. Wilkens

**Technical corrections to the 2ⁿᵈ revised manuscript**

We thank the editor Appy for his great effort to overview and coordinate the reviewing process Here we address the technical corrections requested by the editor for the final version of the manuscript.

P3. 23-27. Unclear sentence; please rewrite and/or split in two
We have rewritten the sentence as follows: *"The favored age model in Lauretano et al. (2016) proposes two 405-kyr cycles in this complex interval which provided absolute age estimates for early Eocene hyperthermal events. The two 405-kyr cycle age model is favoured because it assumes constant sedimentation rates. However, it did not solve the "50 Ma discrepancy" because the duration for C23n.2n is much too short (295 kyr) compared to CK95 (696 kyr) in ODP 1258."*

P5. 8-11. Put years of publication between brackets, separate by commas instead of semicolons and include 'and' before final reference.
Corrected as requested.

P5. 16. Faculty 5 Geosciences? Should this be Faculty of Geosciences?
In german this is called: Fachbereich 5 – Geowissenschaften, we rewrote to "Faculty 5 – Geosciences".

P5. 21-22 / P6 10-11. Is the raw paleomag and nannofossil data also available in an online database? Specifically mentioned for the isotope data but not here.
The raw inclination, declination, and intensity data for each measurement step are given in online data table S17 to S20. Calcareous nannofossil tables are in online data table S31 to S35.

P6 3. Delete comma after Perch-Nielsen
Corrected.

P7 34. Should be 'regionally' instead of 'regional' (adverb)
Corrected.

P8. 14. 'can be identified at the Leg 208 sites' (or similar)
Sentence was corrected to: *"The larger MADs that can be identified at Leg 208 sites in a few samples are not simply related to the intensity of their remanent magnetization."*

P10. 3. 'proven' instead of 'proved'
Corrected.

P10. 15 'the Tethyan realm' (or similar) instead of 'Tethyan data'
Corrected.

P13. 27. Please avoid starting sentence with a number; perhaps start sentence with 'Due to the evolution…'
Sentence was changes to :" Due to the evolution of the precession frequency p, the periods have been estimated to be ~22.5 and ~18.6 kyr (Laskar et al., 2004) 50 million years ago."

P16. 15-17. Please rewrite to: "Multiple carbon cycle perturbations are documented as negative carbon isotope excursions (CIEs) and the dissolution of carbonates at the seafloor, both pointing to massive releases of 13C-depleted carbon to the ocean-atmosphere system. Associated warming has led to the term 'hyperthermals' for these events."
Rewrote.

P16. 18. Add "If the 13C-depleted carbon caused the warming and all events were triggered by carbon from the same reservoir, the magnitude (instead of extent?) of the CIE's are scaled to the amount of carbon injected (Pagani et al. 2006)…" These are the implicit assumptions Pagani's paper made, and these are both under discussion.

Added. Thanks for the clarification, this is indeed important to the reader.

P16. 28. 'hyperthermal' instead of 'hypothermal'

Corrected.

P17. 19. Correct reference

Corrected.

P19. 6-7. To my recollection, the PE boundary is defined on the clay layer that corresponds to the steepest slope of the CIE at the GSSP (not the base if you will and also not formally defined on the CIE but the layer)

Page 19, lines 6-7 read "*Combining the new astrochronology with the revised magnetostratigraphy for the Ypresian allows us to consider the significance of the abrupt global increase in spreading rates in Chron C23n.2n, which is also known as the "50 Ma discrepancy" in the Paleogene Time Scale (Vandenberghe et al., 2012).*"

It is not obvious what the comment is aiming at.

However, Page 20, line 6-7 read "*The GSSP of the Ypresian, that also marks the Paleocene/Eocene boundary, is defined at the base of the onset of the Paleocene-Eocene Thermal Maximum (PETM) carbon isotope excursion (CIE) (Aubry et al., 2007) about two thirds of the way down in magnetochron C24r (Westerhold et al., 2007) at base of Zone CNE1 where the Top of the calcareous nannofossil Fasciculithus richardii group and the Base of Calcareous nannofossil excursion taxa (CNET) occur (Westerhold et al., 2007; Agnini et al., 2014; Westerhold et al., 2015).*"

We corrected to "*…, is defined at the basal inflection of the carbon isotope excursion (CIE) of the Paleocene-Eocene Thermal Maximum (PETM) (Aubry et al., 2007), …*"

P20. 25. Diachrony, consider to use diachronicity instead

Changed to diachronicity.

P21. 10. The community probably agrees that the Cramer et al 2003 paper was the first to try, but shown to be quite wrong due to incomplete records; ok to omit from citation for this particular statement.

Citation is omitted.

[revised manuscript text omitted]